



# The role of thermokarst evolution in debris flow initiation (Hüttekar Rock Glacier, Austrian Alps)

Simon Seelig[1], Thomas Wagner[1], Karl Krainer[2], Michael Avian[3], Marc Olefs[3], Klaus Haslinger[3], and Gerfried Winkler[1]

[1]Institute of Earth Sciences, NAWI Graz Geocenter, University of Graz, Heinrichstrasse 26, 8010 Graz, Austria
[2]Institute of Geology, University of Innsbruck, Innrain 52, 6020 Innsbruck, Austria
[3]Central Institute for Meteorology and Geodynamics, Hohe Warte 38, 1190 Vienna, Austria

**Correspondence:** Simon Seelig (simon.seelig@uni-graz.at)

**Abstract.** A rapid sequence of cascading events involving thermokarst lake outburst, rock glacier front failure, debris flow development and river blockage hit Radurschl Valley (Ötztal Alps, Tyrol) on 13 August 2019. Compounding effects from multivariate permafrost degradation and drainage network development initiated the complex process chain. The debris flow dammed the main river of the valley, impounding a water volume of 120,000 m³ that was partly drained by excavation to prevent

a potentially catastrophic outburst flood. Since the environmental forces inducing the debris flow evolved under ambiguous conditions, potentially destabilizing factors were analyzed systematically to deduce the failure mechanism and establish a basis for multi hazard assessment in similar settings. Identification and evaluation of individual factors revealed a critical combination of topographical and sedimentological disposition, climate, and weather patterns driving the evolution of thermokarst and debris flow. Progressively changing groundwater flow and storage patterns characterizing the hydraulic configuration within

the frozen sediment accumulation governed the slope stability of the rock glacier front. The large amount of mobilizable sediment, dynamically changing internal structure, and substantial water flow along a rapidly evolving channel network eroded into the permafrost body, render active rock glaciers complex multi hazard elements in periglacial, mountainous environments.

## 1 Introduction

Climate change and its adverse effects on slope stability rapidly alter patterns of landslide occurrence and the balance between

destabilizing and resisting forces (Crozier, 2010; Gariano and Guzzetti, 2016; Adler et al., 2022). Landslide engineers and decision makers face a delicate challenge since data-driven hazard assessment methods are at risk of losing predictive power as boundary conditions change at timescales that are short compared to typically available landslide records (Haeberli and Whiteman, 2015; Patton et al., 2019). In this context, a process-based understanding of potential failure mechanisms is indispensable for identifying landslide hazard at sites considered stable under past conditions (Evans and Delaney, 2015; Schauwecker et al.,

2019). This is especially true for debris flows initiating in permafrost affected terrain, due to the rapid alteration of slope stability in their initiation zones, their ability to traverse great distances at high velocities, and their high impact forces (Jakob and Hungr, 2005; Jakob et al., 2012; Petley, 2012; Dowling and Santi, 2014; Nikolopoulos et al., 2014). Hence failure in remote



areas which are typically characterized by data scarcity and incomplete monitoring has the potential to dramatically affect people and infrastructure far from debris flow initiation (Haque et al., 2016, 2019).

The European Alps are affected by severe increases in surface air temperature (∼0.3 °C per decade, exceeding global warming rates), altering snow cover dynamics and driving permafrost thaw (Beniston et al., 2018; Hock et al., 2019; Patton et al., 2019; Olefs et al., 2020; Matiu et al., 2021; Fox-Kemper et al., 2021). The increasing frequency of high-intensity precipitation events amplifies the likelihood of landslides (Giorgi et al., 2016; Rajczak and Schär, 2017; Schlögl and Matulla, 2018; Hock et al., 2019; Patton et al., 2019; Ranasinghe et al., 2021). Slope stability decreases as subsurface liquid water

content, ice ductility, and permeability increase in response to permafrost degradation (Patton et al., 2019; Hock et al., 2019). Moreover, glacier retreat enhances the accumulation of loose, unstable sediment at high elevations (Stoffel and Huggel, 2012; Deline et al., 2015; Buckel et al., 2018; Hock et al., 2019; Mölg et al., 2021). These changing environmental conditions alter the geotechnical and hydraulic ground properties, and modify the characteristics and seasonality of landslides accordingly (Stoffel and Huggel, 2012; Gariano and Guzzetti, 2016; Beniston et al., 2018; Patton et al., 2019). The consecutive variation in process

chains and cascading events makes them especially difficult to predict under a changing climate, with potentially catastrophic consequences downstream (Deline et al., 2015).

In terms of landslide initiation, the steep slopes and large sediment accumulations provided by active rock glacier fronts pose a serious hazard. Active rock glaciers, i. e. distinct sediment accumulations consisting of ice-debris mixtures slowly creeping downhill, constitute important periglacial landforms regarding sediment dynamics and hydrology due to their large sediment

transfer capability, pronounced water storage capacity, and wide-spread occurrence in mountainous terrain (Kummert and Delaloye, 2018; Winkler et al., 2018; Jones et al., 2019b; Hayashi, 2020; Wagner et al., 2020, 2021a). Common deformation rates are on the order of decimeters to meters per year, accelerating across the European Alps in response to climate change (Kääb et al., 2007; Roer et al., 2008; Beniston et al., 2018; Groh and Blöthe, 2019; Hock et al., 2019; Kenner et al., 2020; Fleischer et al., 2021; Fox-Kemper et al., 2021; Marcer et al., 2021). Thermo–hydro–mechanical processes determine the

deformation characteristics of rock glaciers, with plastic deformation governed by permafrost ice content and temperature, while discrete shear failure is commonly initiated by elevated pore-water pressures along the failure surface (Arenson and Springman, 2005; Ikeda et al., 2008; Buchli et al., 2013, 2018; Krainer et al., 2015; Cicoira et al., 2019; Kofler et al., 2021). Active rock glaciers exhibit complex drainage patterns indicating a dual groundwater flow system, where large amounts of water are rapidly transported along a network of convoluted meltwater channels eroded into the frozen rock glacier core

(Wahrhaftig and Cox, 1959; Potter, 1972; White, 1971; Johnson, 1978; Giardino et al., 1991; Burger et al., 1999; Krainer and Mostler, 2002; Vonder Mühll et al., 2003; Arenson et al., 2010; Springman et al., 2012; Winkler et al., 2018; Jones et al., 2019b; Wagner et al., 2021b; Kainz, 2022).

Active rock glaciers constitute multi hazard elements (Kappes et al., 2012; Gallina et al., 2016) in that they induce a spectrum of mass movement processes ranging from occasional rockfall to debris flow events, potentially involving complex chain

processes (Burger et al., 1999; Kummert and Delaloye, 2018; Kummert et al., 2018). Destabilizing rock glaciers frequently experience significant acceleration, often accompanied by the appearance of morphological discontinuities such as cracks and crevasses (Kääb et al., 2007; Roer et al., 2008; Stoffel and Huggel, 2012; Delaloye et al., 2013; Marcer et al., 2019).





Debris flows initiated by destabilizing rock glacier fronts occur most frequently in response to heavy rainfall. However, intense snowmelt, rain-on-snow events, and exceptionally warm periods have also been discerned as triggering factors (Rebetez et al., 1997; Lugon and Stoffel, 2010; Krainer et al., 2012; Bodin et al., 2017; Kummert et al., 2018; Marcer et al., 2020; Kofler et al., 2021). Regardless of the detailed initiation process, the mechanics of debris flows require excessive amounts of water, capable of transporting the mobilized debris down the flow path (Soeters and van Westen, 1996). Thus, assessing the hazard potential of rock glaciers requires an integrated approach combining hydrogeological, meteorological, thermal, geomorphological, and mechanical aspects in a coherent framework.

The aim of this paper is to explore the destabilizing factors leading to failure of an active rock glacier front in a high mountain cirque, and the actuated cascading processes involving thermokarst lake outburst flood, debris flow and river blockage. We evaluate a set of potentially contributing factors, assess critical combinations, and develop a consistent conception explaining debris flow initiation and evolution. Similarities and differences with respect to documented rock glacier front failures and glacial lake outburst floods (GLOFs) are analyzed, and conclusions drawn regarding hazard potential.

## 2   Study site

The study site comprises the highest areas of Radurschl Valley, a valley in the Western Ötztal Alps constituting a headwater tributary to the Inn River (Fig. 1). The landscape is characterized by rugged terrain of glacial and periglacial origin, including steep slopes and crests, interrupted by relatively flat areas covered by talus, moraines, and periglacial sediments. The area shows an exceptionally high rock glacier density (56 % areal coverage), as well as small remnants of cirque glaciers in its uppermost parts (Kerschner, 1982; Krainer and Ribis, 2012; Wagner et al., 2020, 2021a).

The specific cirque under consideration is the Hüttekar-cirque, a small (2.8 km²) headwater catchment encircled by ragged rock walls except to the west, where it steeply descends to Radurschl Valley (Fig. 2; 46°54´ N, 10°39´ E). The altitudinal range in the study area varies from 2,387 to 3,353 m a. s. l. (mean altitude 2,870 m a. s. l.), including a relatively flat valley bottom between 2,600 and 2,700 m a. s. l. The bedrock lithology is composed of metamorphic rocks of the Ötztal–Stubai–Complex (Ötztal–Bundschuh nappe system; Hoinkes and Thöni (1993); Schmid et al. (2004)), as illustrated in Fig. 3. The mountain ridges bordering the cirque to the south and to the east are dominated by orthogneiss (augen- and flasergneiss), the ridge to the north and northwest exhibits muscovite-granite-orthogneiss. At Rotschragenjoch and south of Bruchkopf, paragneiss and micaschist with thin intercalations of amphibolite are exposed. The terrain is composed of bedrock (38 %) mainly exposed in the highest parts, talus, and debris slopes (17 %) along the valley sides, while the lower parts are covered by moraine deposits (12 %) and four rock glaciers (27 %). Two small, north facing cirque glaciers (Glockturmferner, Hüttekarferner; 6 %) are situated at 2,853 and 3,029 m a. s. l., respectively.

The local climate is influenced by the dry, inner alpine conditions prevailing in the Ötztal Alps (Frei and Schär, 1998; Isotta et al., 2014). Moderate annual precipitation (1042 mm) and low mean annual air temperature (-2.5 °C) reflect the high altitude of Hüttekar-cirque and its central position close to the main chain of the Alps. Monthly precipitation reaches its maximum in August and its minimum in February, with the respective months corresponding to the highest and lowest mean air temperature





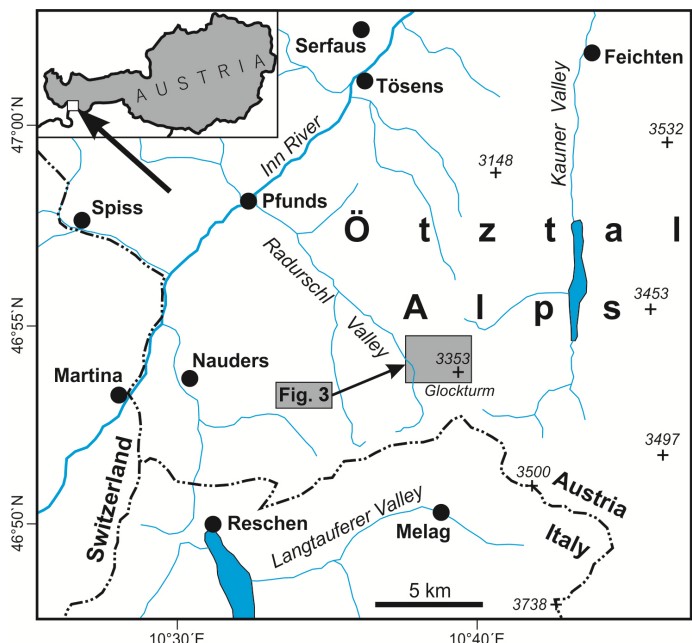

**Figure 1.** Location map of Hüttekar in the Ötztal Alps, Tyrol (Austria).

**Table 1.** Long-term (1976–2019) mean monthly air temperature and precipitation in Hüttekar-cirque (based on Hiebl and Frei (2016, 2018)).

|  | J | F | M | A | M | J | J | A | S | O | N | D |
|---|---|---|---|---|---|---|---|---|---|---|---|---|
| Air temperature (°C) | -9.7 | -9.8 | -7.7 | -4.9 | -0.3 | 3.2 | 5.4 | 5.4 | 2.3 | -0.5 | -5.5 | -8.4 |
| Precipitation (mm) | 64 | 45 | 61 | 57 | 99 | 125 | 151 | 156 | 90 | 71 | 61 | 62 |

(Table 1). The local altitude distribution and dry climatic conditions promote the development of rock glaciers. The equilibrium line altitude of glaciers, rising from ∼3,100 to ∼3,300 m a. s. l. during the 20$^{th}$ century, is located within the steep summit region (Žebre et al., 2021). These unfavorable conditions for cirque glacier development allow rock glaciers to cover the extensive flat terrain above the lower permafrost boundary, ranging from ∼2,400 to ∼2,600 m a. s. l. (Kerschner, 1982; Boeckli et al., 2012a; Ribis, 2017).


The coevolution of cirque glaciers and rock glaciers is highlighted by comparing a set of glacier inventories ranging from the little ice age (about 1850 CE) to 2015 CE (Fig. S1; Fischer et al. (2015); Buckel et al. (2018)), documenting the development of Glockturmferner from a pure ice glacier to a largely debris covered glacier, presumably transitioning into the ice-cored rock glacier below (Anderson et al., 2018; Jones et al., 2019a; Knight et al., 2019). Along the surface of Hüttekar Rock

Glacier, distinct ridges and furrows are visible in the northernmost part, while the central and southern parts show a smooth and flat surface morphology. Rock glacier debris is composed of orthogneiss derived from the Glockturm massif. Poorly sorted boulders, arranged in a loose, clast-supported structure form a heterogeneous surface layer of variable thickness. The blocky




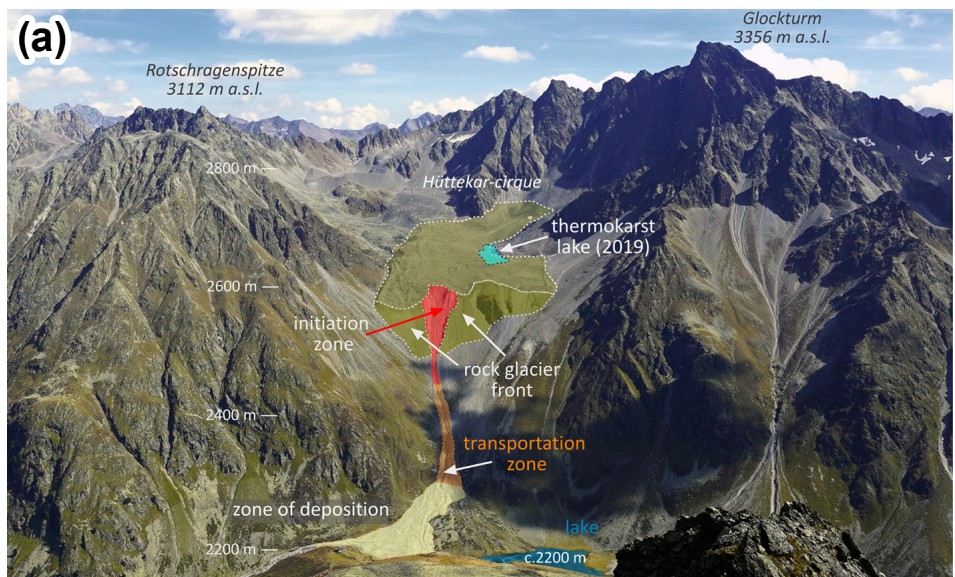

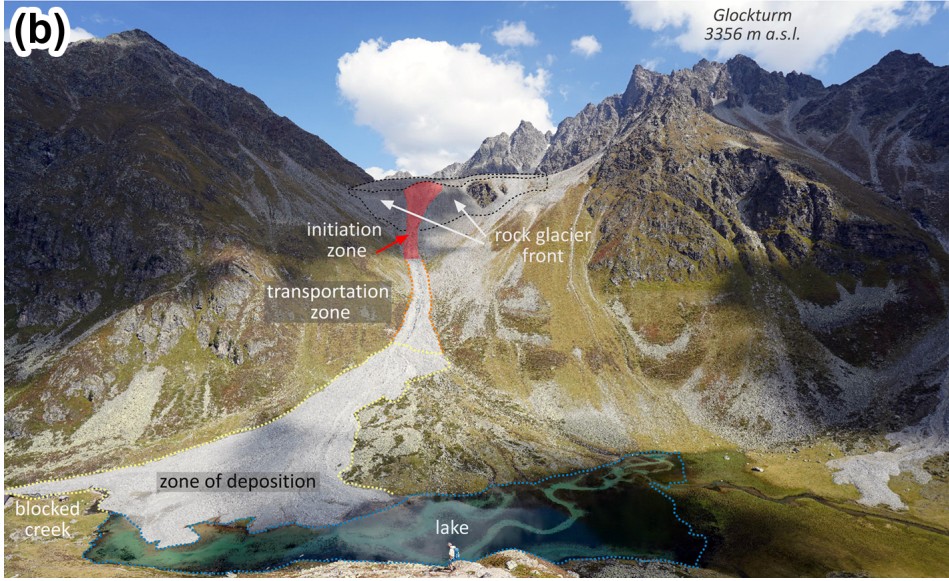

**Figure 2.** (a) Illustration of Hüttekar-cirque, Hüttekar Rock Glacier, the debris flow and the impounded lake (view is towards the east). The debris flow was initiated at the steep front of Hüttekar Rock Glacier (highlighted), caused by rapid drainage of a thermokarst lake that existed between 01 June and 13 August 2019 (former position indicated). The debris flow path sections are outlined and labelled according to Hungr (2005). (b) Debris flow morphology two years after its initiation. The progressively enlarging initiation zone eroded already significant parts of the steep rock glacier front. The transport zone is characterized by a set of levees along a narrow channel. The former flow path of the blocked river is still clearly visible below the lake surface. The channel draining the lake was excavated to prevent a potentially catastrophic outburst following river blockage in August 2019 (photographs: Rudolf Philippitsch, 14 September 2021).



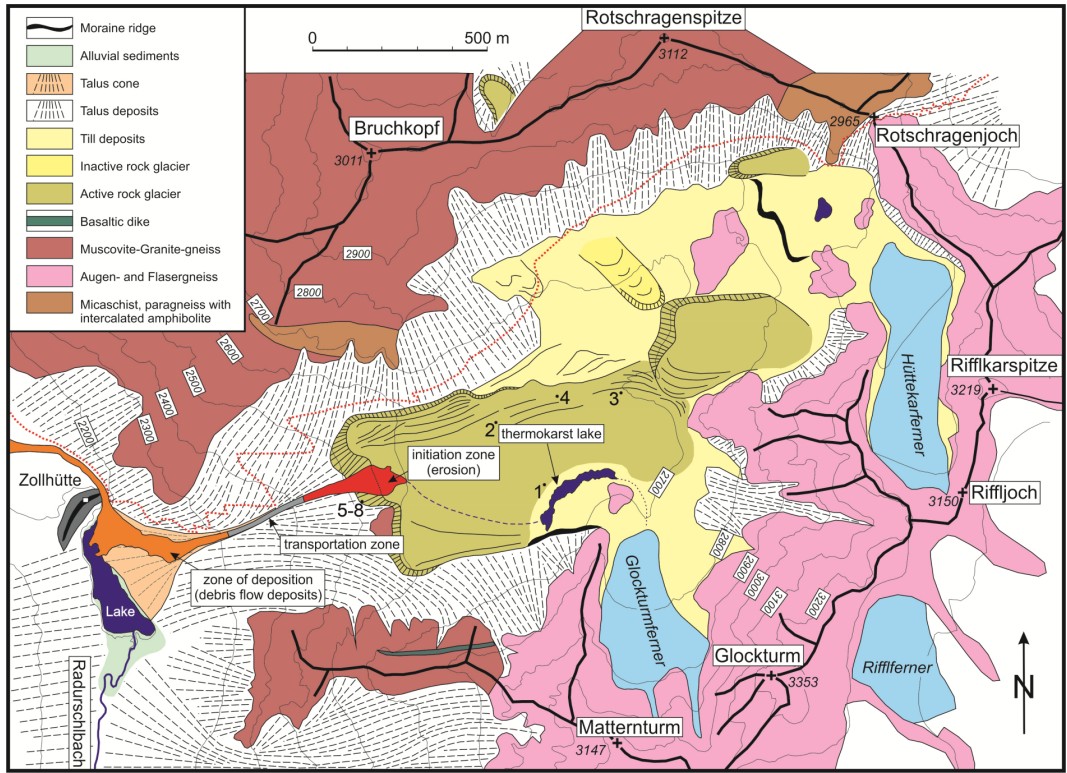

**Figure 3.** Geological-geomorphological map, compiled using the most recent provisional geological map provided by the Geological Survey of Austria (Moser, 2012). It is complemented by ortho-images and a high-resolution digital terrain model (DTM) derived from airborne laser scanning data (Government of the Province of Tyrol, 2021a). The preliminary map is based on comprehensive field mapping (2019–2021). Numbers indicate grain size analysis sampling locations.

surface layer covers a finer-grained, frozen layer dominated by well-graded gravel and sand that is exposed along the rock glacier front. Water flowing along the permafrost table is visible and audible between boulders at several places in the southern part of the rock glacier, following distinct channels eroded into the ice core. Despite its considerable catchment area, Hüttekar-cirque is drained exclusively by subsurface flow, emerging as a group of small springs at the toe of the slope descending to Radurschltal.

## 3 Debris flow event

An active, retrogressive debris flow erodes the steep slope bordering Hüttekar-cirque to the west (Fig. 2; classification according to Varnes (1978); Cruden and Varnes (1996); Hungr et al. (2014)). Following a moderate precipitation event, destabilization initiated on 13 August 2019, mobilizing a volume of several thousand m$^3$ from the steep rock glacier front (Fig. 4a). The event description is based on in situ observations of Josef Waldner, staff of the nearby hut Hohenzollernhaus (pers. comm.).



The debris flow started at 03:00 AM (Central European Summer Time, UTC+2), the main event lasted until about 12:00 PM, followed by reduced debris flow activity that persisted until the next day. Slope failure initiated along an irregularly shaped

rupture in ice-cemented debris, exposed at the main scarp (Fig. 4b). Accelerating and disintegrating, the transported mass evolved into a debris flow following a narrow channel down the steep slope below the rock glacier front (Fig. 2, Fig. 3). About 200 m below the initiation zone, the material spread out and formed a deposition fan of 33,000 m$^2$ and an estimated volume of 40,000 – 50,000 m$^3$, thereby damming the river Radurschlbach at 2,200 m a. s. l. (Fig. 4c, Fig. 4d). Consequently, a lake covering an area of ∼60,000 m$^2$ developed in Radurschl Valley, causing the downstream riverbed to fall dry temporarily (Fig.

4c). Excavation of a drainage channel lowered the mean water depth from 2 m to 1 m during the following days to prevent a potentially catastrophic outburst. Subsequently, a dam was constructed on the debris fan to restrain future debris flows from damming Radurschlbach again.

Concurrently with the debris flow initiation, a thermokarst lake on top of Hüttekar Rock Glacier (∼350 m behind the initiation zone; Fig. 3, Fig. 4e, Fig. 4f) drained almost completely within one day. The thermokarst lake had started to develop

coincidently with the onset of snowmelt in early June 2019 within a shallow depression where massive ice within the rock glacier prevented drainage (Fig. 4g). During the last decades, a comparable feature had never been observed before in Hüttekar-cirque, despite frequent visits by hikers, hunters, shepherds, and staff of Hohenzollernhaus. In the stage of its largest extent, the thermokarst lake was approximately 300 m long, up to about 150 m wide and 4–5 m deep, comprising an estimated water volume of ∼150,000 m$^3$. Effective drainage through a large crevasse (width ∼1.5 m, height ∼2 m) caused initiation of the

debris flow at the front of the rock glacier (Fig. 4e, Fig. 4f, Fig. 4h). Large amounts of water were rapidly transferred to the debris flow initiation zone and torrent beneath. Retrogressive linear erosion visible in the initiation zone indicates that concentrated water flow emerged at the main scarp (Fig. 4b), in good agreement with earlier observations of rock glacier front failures (Kummert et al., 2018; Marcer et al., 2020; Kofler et al., 2021). Since then, the depression never filled again but still shows distinct morphology.

**4    Methods**

The inherent stability of the slope under consideration determines its response to changes affecting the balance of driving and resisting forces (Glade and Crozier, 2005; Crozier, 2010). Systematic identification of factors promoting slope instability is conducted by differentiating (Glade and Crozier, 2005):

(1) predisposing factors (inducing a static setting capable of enhancing the destabilizing impact of dynamic forces),

(2) preparatory factors (dynamic forces shifting the slope towards a state susceptible to failure), and

(3) triggering factors (dynamic forces initiating failure).

Acknowledging that debris flow hazard analysis requires consideration of multiple destabilizing factors, their impact on slope stability is characterized individually to establish a basis for reconstructing the failure mechanism. The chosen methods aim at maximizing comparability to earlier studies on rock glacier front failures and debris flows in periglacial regions.



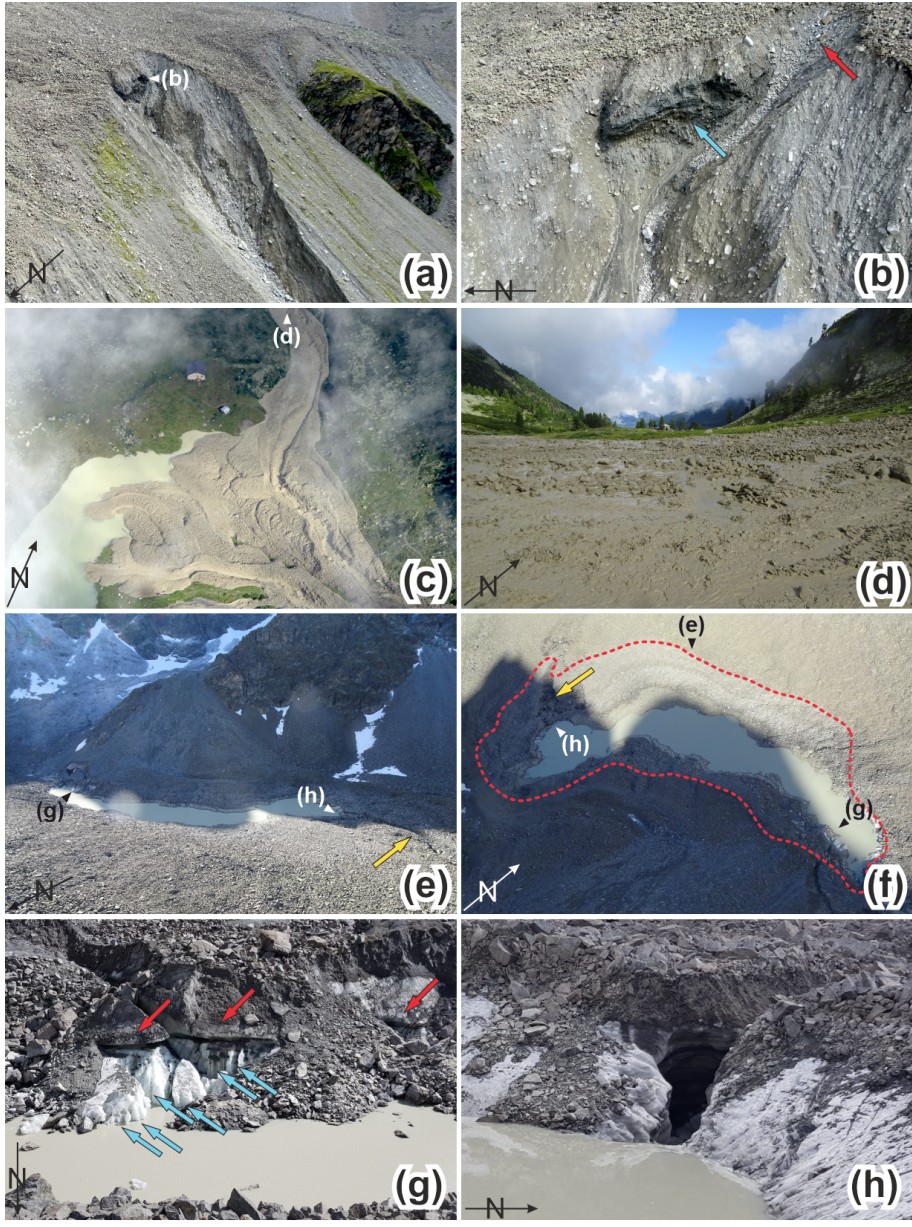

**Figure 4.** Images documenting the debris flow event that initiated on 13 August 2019. (a) Debris flow initiation zone at the steep front of Hüttekar Rock Glacier. (b) Ice-cemented debris (blue arrow) exposed in the main scarp. Linear retrogressive erosion indicates concentrated water flow (red arrow). (c) Debris flow deposits blocking Radurschlbach. (d) Debris flow material. (e) Thermokarst lake and drainage channel (yellow arrow) on Hüttekar Rock Glacier. (f) Extent of thermokarst lake on 14 August 2019 and maximum extent one day earlier (dashed line). The channel connecting the lake to the debris flow initiation zone is indicated by the yellow arrow. (g) Impermeable ice underlying the thermokarst lake. Blue arrows indicate vertical convexities attributed to thermal convection of water during the lake development. Red arrows indicate undercutting of the ice along the lake shoreline, promoted by thermal convection. (h) Thermokarst channel eroded into the frozen rock glacier core, facilitating rapid drainage of the thermokarst lake. Photographs were taken on 14 August 2019 (c–h) and 26 August 2019 (a–b), respectively, by Roman Außerlechner, Thomas Figl, Werner Thöny, and Josef Waldner.





### 4.1 Predisposing factor analysis


A topographical setting promoting rock glacier destabilization is a necessary (but not sufficient) precondition for rock glacier front failure. Attributes significantly affecting slope stability include morphometric parameters such as length, slope angle, and curvature (Chowdhury et al., 2010; Reichenbach et al., 2018; Marcer et al., 2019). Sediment erosion, transport, and deposition depend on local energy gradients, available transport vectors, and material properties (Bracken et al., 2015). Collectively, these

variables control local loading stresses and the availability of material for subsequent mobilization (Kummert et al., 2018; Kofler et al., 2021). The respective impact of these factors on slope stability is evaluated using a high-resolution (1 × 1 m) DTM based on airborne laserscanning data acquired two years prior to debris flow occurrence (Government of the Province of Tyrol, 2021a), applying SAGA GIS 2.1.4 for morphometric analyses (Conrad et al., 2015). Slope angle and downslope curvature calculations (Freeman, 1991) are performed using a smoothed (cubic convolution resampling to 10 × 10 m) DTM to

detect the fundamental topographical features.

Thermal ground conditions and the presence of subsurface ice affect the shear strength of involved materials, alter groundwater flow patterns, and impact the storage and release of water by thermokarst evolution. The spatial permafrost distribution is estimated by taking recourse to the Alpine Permafrost Index Map, providing a static proxy for potential permafrost occurrence and its spatial coherence (Boeckli et al., 2012b, a). While the dynamic effect of climate change is not considered explicitly by

the index map, permafrost degradation is expected to intensify close to the lower permafrost boundary (Marcer et al., 2019). The local influence of topography on the energy available for permafrost degradation (Scherler et al., 2014; Marcer et al., 2019) is assessed by calculating the potential incoming solar radiation based on the DTM (Hock, 2005; Cuffey and Paterson, 2010). Calculations of mean annual solar radiation under clear-sky conditions are performed employing GRASS GIS 8.0 (GRASS Development Team, 2022) module r.sun (Hofierka et al., 2007), assuming an average surface albedo of 0.2.

The mechanics of debris flows are largely determined by solid-fluid interactions, rendering grain size distribution and water content major variables driving flow behavior (Iverson, 1997). The rate of water supply is determined by the permeability of sediments above the initiation zone. Geotechnical and hydrogeological characteristics of the materials involved are inspected by analyzing the grain size distribution of the rock glacier surface layer at one coarse-grained and three relatively fine-grained domains (sampling locations indicated in Fig. 3). At each location the maximum diameter of 200 clasts lying side by side is

measured in an area of approximately 2 × 2 m (fine grained domains) to 5 × 5 m (coarse grained domain). The grain size distribution of the poorly sorted sediment layer below the blocky surface is analyzed by manual wet sieving of a sample taken at the steep rock glacier front.

### 4.2 Preparatory factor analysis

The geometry of the active rock glacier front depends on the dynamic balance between rock glacier kinematics and erosion

rates (Kummert and Delaloye, 2018). Time series of ortho-images are used to quantify rock glacier surface displacement rates and the evolution of surface features indicating destabilization (including cracks, crevasses, and scarps; Avian et al. (2007); Roer et al. (2008); Delaloye et al. (2013); Marcer et al. (2019); ortho-images provided by the Government of the Province



of Tyrol (2021b)). In the observation period 1970–2020, 200 prominent blocks, geometrically well distributed on the rock glacier surface, are visually identified at each ortho-image epoch to approximate the horizontal surface displacement rate at the

respective location. Additionally, the position of the rock glacier front line (top of the erosional slope) is mapped at every epoch. Both analyses provide the basis for assessing the kinetic patterns on the rock glacier surface (Avian et al., 2009; Kummert and Delaloye, 2018).

Climatic factors and their progressive change exert a key control on slope stability (Gariano and Guzzetti, 2016; Patton et al., 2019). Alterations in volume and intensity of rainfall, snowmelt and ice melt affect subsurface water content, pore-water

pressure and seepage forces (Ikeda et al., 2008; Cicoira et al., 2019; Patton et al., 2019). The long-term evolution of air temperature and precipitation in Hüttekar-cirque is evaluated using the gridded (1 × 1 km) SPARTACUS data set (Spatiotemporal Reanalysis Dataset for Climate in Austria; Hiebl and Frei (2016, 2018)). Daily precipitation totals and 24 hour mean air temperature data observed from 1976 to 2019 are extracted and averaged across Hüttekar-cirque. Specific aspects of the local climate are explored by calculating the annual positive degree day sum (daily mean air temperature > 0 °C) as proxy for avail-

able melting energy, as well as precipitation due to very wet days (> 95th percentile) reflecting the annual magnitude of heavy precipitation events (Klein Tank et al., 2009; Cuffey and Paterson, 2010). Considering that the hydrometeorological conditions preceding the debris flow determine the critical amount of water necessary to initiate failure (Crozier, 2010), the impact of these climate indices is assessed by comparing their respective values during summer 2019 (01 June–13 August) to previous years (1976–2019). To estimate changes in their central tendency, trend direction and rate of change are evaluated using the

nonparametric seasonal Mann Kendall test (Theil, 1950; Sen, 1968; Hirsch et al., 1982; Hirsch and Slack, 1984). Calculations are performed employing R packages 'rkt' 1.6 (Marchetto, 2021) and 'climdex.pcic' 1.1 (Bronaugh, 2020), implemented in R 4.2.0 (R Core Team, 2022).

In the Alps, individual storms exhibit strong intensity gradients at length scales < 5 km (i. e. below the basic length-to-crest scale; Haiden et al. (2011); Nikolopoulos et al. (2014, 2015a); Marra et al. (2016); Destro et al. (2017)). The combination

of weather station data and remote sensing information allows the detection of individual event characteristics (Haiden et al., 2011; Borga et al., 2014; Marra et al., 2014, 2016; Destro et al., 2017). Detailed (∼hourly) temporal resolution of precipitation data is necessary to avoid biased estimates of rainfall intensity and duration (Marra, 2019). Meteorological analyses at time intervals from hours to months are thus based on the INCA system (Integrated Nowcasting through Comprehensive Analysis; Haiden et al. (2011)), providing gridded (1 × 1 km) data sets at hourly temporal resolution. Inclusion of 12 weather stations at

a distance < 25 km from Hüttekar-cirque, in combination with C-band radar measurements and Meteosat Second Generation Satellite Products, allows assessing the spatial and temporal patterns of meteorological conditions at the study site.

Snow cover development is assessed using the spatially distributed, physically based snow cover model SNOWGRID (Olefs et al., 2013). It employs a simple two-layer scheme, considering settling, the heat and liquid water content of the snow cover, snowline depression effects and the energy added by rain. For every time step and layer, the state variables snow density,

snow water equivalent, snow temperature, liquid water content, bottom liquid water flux, and surface albedo are calculated. The primary focus of the model is to obtain a high-resolution representation of their spatial distribution and to provide fast calculations on a large grid. The model employs a 100 × 100 m bilinear interpolation of the INCA data set in combination with



schemes for radiation and cloudiness developed at the Central Institute for Meteorology and Geodynamics (Austria). These are based on ground measurements, satellite products, and high quality solar and terrestrial radiation data (Olefs et al., 2013, 2016).

For each winter season, the cumulated snowmelt volume in Hüttekar-cirque is calculated and compared to the 2018/19 season. The rate of snowmelt is approximated by the respective seasonal average during the time span between maximal snow volume and complete ablation of the winter snow cover (Kofler et al., 2021).

Glacial meltwater infiltrates directly from Glockturmferner and indirectly from Hüttekarferner into the rock glacier (Fig. 3), thus the total volume and average rate of ice melt is calculated for each year between ablation of the snow cover and 13

August, based on the surface energy balance for the respective glacier (Hock, 2005; Cuffey and Paterson, 2010). Ice melt rates are calculated at hourly intervals by evaluating radiative, turbulent and advective energy fluxes per glacier derived from INCA, assuming a constant glacier surface temperature of 0 °C (considered a reasonable approximation for the ablation zones of temperate glaciers during the summer season; Cuffey and Paterson (2010)). Shortwave net radiation is calculated by evaluating global radiation, accounting for the reflective and shadowing influences of topography as outlined in Sect. 4.1. Atmospheric

conditions influencing longwave radiation are considered following Greuell et al. (1997) and Oerlemans (2000). Sensible and latent heat fluxes are calculated by parameterization of the respective transport processes based on turbulence similarity (Cuffey and Paterson, 2010). The rainfall heat flux is estimated assuming that rainfall temperature approaches the near-surface air temperature. Physical specifications and plausibility evaluation are provided in Appendix A.

### 4.3 Triggering factor analysis

Rainfall-induced debris flows are triggered by infiltration rates exceeding subsurface drainage rates and the corresponding alteration of pore-water pressures (Sidle, 1984; Johnson and Sitar, 1990; Anderson and Sitar, 1995; Wieczorek, 1996; Wieczorek and Glade, 2005; Crozier, 2010). Infiltrating water adversely affects slope stability by adding additional weight while simultaneously decreasing shear strength, either by lowering effective stresses in response to increasing positive pore-water pressures or by eliminating suction due to declining negative pore-water pressures (Johnson and Sitar, 1990; Anderson and Sitar, 1995;

Chowdhury et al., 2010; Crozier, 2010). In order to evaluate the impact of the storm immediately preceding the debris flow on 13 August 2019, rainfall time series are extracted from INCA, averaged across the potentially contributing area (Hüttekar-cirque), and discretized into single events by defining a minimum of 24 hours between individual events (Nikolopoulos et al., 2015b; Marra et al., 2016). Calculations are performed employing the R package 'IETD' 1.0 (Duque, 2020), implemented in R 4.2.0 (R Core Team, 2022). The resulting ensemble of rainfall event duration, volume and average intensity is analyzed by

conducting a frequency analysis. The severity of individual events is assessed employing the frequentist approach developed by Brunetti et al. (2010) and Peruccacci et al. (2012). The rainfall event characteristics immediately preceding the slope failure are compared to regional critical rainfall thresholds for debris flow initiation (Nikolopoulos et al., 2015b; Marra et al., 2016).

Glacial lake outbursts evolving into debris flows are commonly initiated by mechanical failure of the dam or by rapid expansion of the lake drainage system through thermal erosion of ground ice. The volume and geometry of the impounded reservoir,

dam characteristics, failure mechanism, downstream topography and sediment availability control the hazard potential (Clague and Evans, 1994). The development of thermokarst features, including meltwater lakes and channels, is highly sensitive to



thermal ground conditions and their response to climate change (Kääb and Haeberli, 2001). The spatiotemporal evolution of the thermokarst lake on Hüttekar Rock Glacier is deduced by combining Sentinel-2 multi-spectral satellite data and the latest (2017/18) available DTM, characterizing the lake surface area dynamics and corresponding water volume development (mod-
ified Copernicus data 2020 processed by Sentinel Hub; Sentinel Hub (2020); Government of the Province of Tyrol (2021a)). Due to the dynamic nature of the rock glacier surface, volume estimates are considered rough estimates, failing to account for water stored in the pore space and potential intra-permafrost channels. Breakthrough of thermokarst channels is facilitated by a positive feedback mechanism of energy transfer along the channel system inducing ice melt and channel enlargement, which in turn increases discharge and accelerates channel growth (Nye, 1976; Clarke, 1982, 2003; Spring and Hutter, 1981a, b; Clague
and Evans, 1994; Walder and Costa, 1996; Huggel et al., 2004; Cuffey and Paterson, 2010; Clague and O'Connor, 2015). Dimensional analysis indicates that the energy provided by the upstream water body governs expansion rates of the drainage system (Clarke, 1982, 2003; Cuffey and Paterson, 2010). The prediction of the specific channel system evolution is impossible without exact knowledge of its geometry and hydraulic properties. However, the estimation of the meltwater lake energy budget provides a measure of the total energy available for thermokarst development. Undercutting and vertical pipe structures eroded
into the ice along the lake shoreline provide field evidence for thermal convection (Fig. 4g), promoting energy turnover and melting of ice beneath the lake while constraining the surface water temperature to < 4 °C due to negative thermal expansion (Haeberli et al., 2001; Kääb and Haeberli, 2001; Werder et al., 2010). Rapid transfer of energy exchanged at the lake surface to the ice sealing the lake bottom keeps the water at roughly constant temperature, thus allowing for an assessment of the lake energy balance based on INCA data (physical specifications are provided in Appendix A).

## 5 Results

A systematic evaluation of destabilizing factors and adverse combinations thereof requires an individual evaluation of every potentially contributing factor.

### 5.1 Predisposing factors

Morphometric analysis of the study area reveals several features promoting debris flow development. The slope (900 m long
and 400 m in elevation difference) bordering Hüttekar-cirque to the west shows a very steep, convex top and a steep, concave main part (Fig. 5a, Fig. 5b). Prior to landslide initiation, the prospective initiation zone was characterized by steep slope angle and convex downslope curvature (Table 2). The channel below the rock glacier front was less steep and concave, while the debris fan was already comparatively flat before the 2019 debris flow (Table 2). Irregular micromorphology, depositional levees, and boulder trains document earlier debris flows (Fig. 2, Fig. 5; Soeters and van Westen (1996); Pack (2005); Hungr
et al. (2014)). Material eroded at the rock glacier front is available for subsequent mobilization further downslope, inhibiting the formation of a stabilizing debris accumulation at the toe of the rock glacier front (Kummert and Delaloye, 2018; Kummert et al., 2018; Kofler et al., 2021).





**Figure 5.** Spatial distribution of predisposing factors across Hüttekar-cirque prior to debris flow initiation. (a) Slope angle, indicating that the debris flow initiated in steep terrain (33°). (b) Downslope curvature, demonstrating that convex topography prevailed in the initiation zone before the debris flow event (+0.05). (c) Potential incoming solar radiation, illustrating the high-energy environment characterizing the debris flow initiation zone (2135 kWh m$^{-2}$ a$^{-1}$). (d) Permafrost index, indicating that the debris flow initiated close to the lower permafrost boundary (Boeckli et al., 2012a). Terrain analyses are based on a high-resolution DTM acquired in the period 2017/2018 by the Government of the Province of Tyrol (2021a). Perceptually uniform and color-vision-deficiency friendly color maps provided by Crameri (2018).





**Table 2.** Morphometric analysis based on a high-resolution DTM acquired in the period 2017/2018 by the Government of the Province of Tyrol (2021a). Results are given as averages across the debris flow initiation zone, transportation zone, and zone of deposition, respectively (indicated in Fig. 3)

| Morphometric parameter | Initiation zone | Transportation zone | Zone of deposition |
|---|---|---|---|
| Mean altitude (m a.s.l.) | 2,541 | 2,362 | 2,227 |
| Mean slope gradient (°) | 33.0 | 25.0 | 14.2 |
| Mean downslope curvature (-) | +0.05 | -0.08 | -0.08 |

Assessment of thermal ground conditions indicates that the debris flow initiation site is subjected to a high energy environment. Potential incoming solar radiation reaches 2,135 kWh m$^{-2}$ a$^{-1}$ (Fig. 5c). The Alpine Permafrost Index Map indicates that the initiation zone is located at the lower permafrost boundary, with permafrost preserved only in very favorable conditions (Fig. 5d). Collectively, the convex morphology and exposed position, strong potential radiation and unfavorable permafrost index are conducive to an advanced stage of permafrost degradation, increased water contents, and large amounts of unfrozen, loose sediment susceptible to mobilization (Marcer et al., 2019).

Grain size analyses indicate that the blocky surface layer of Hüttekar Rock Glacier is composed of poorly sorted, coarse-grained sediment. The finer-grained parts, measured on three locations (1–3; positions given in Fig. 3), show an average grain size of 11.6, 12.8, and 16.0 cm, respectively. The coarse-grained site (4) exhibits an average grain size of 65.6 cm, documenting the heterogeneous structure of the blocky surface layer. Figure 6a displays the respective grain-size distributions. Due to the subordinate presence of components smaller than gravel (largely matrix-free surface layer), the rock glacier classifies as bouldery rock glacier sensu Ikeda and Matsuoka (2006). These characteristics indicate a permeable surface layer exhibiting high infiltration capacity and low hydraulic resistance to water flowing along the permafrost table.

Sieve analyses of samples 5–8 taken at the rock glacier front (Fig. 6b, positions indicated in Fig. 3) show that gravel (64 %) and sand (25 %) are the dominating grain sizes, while the amount of clay and silt is very low (< 1 %). The samples are extremely poorly sorted (4.4 after Folk and Ward (1957)). Loose deposits of sediments exhibiting similar grain size distributions are well known to respond to shearing in a contractive manner and constitute characteristic debris flow material (Iverson, 1997; Savage and Baum, 2005). Experimental investigations of similarly composed source material confirm these observations (dashed lines in Fig. 6b), identifying contractive response to shearing and pore pressure diffusion timescales exceeding the duration of debris flow motion as major prerequisites for mobilization (Major, 1996; Iverson, 1997). Provided sufficient water is available to keep the rock glacier debris saturated, these characteristics promote the development of undrained loading conditions, high sensitivity of effective stresses to sediment compaction and perpetually high excess pore-water pressures in response to deformation (Major, 1996; Iverson, 1997; Major et al., 1997). These features document the compositional propensity of the rock glacier front to debris flow mobilization.




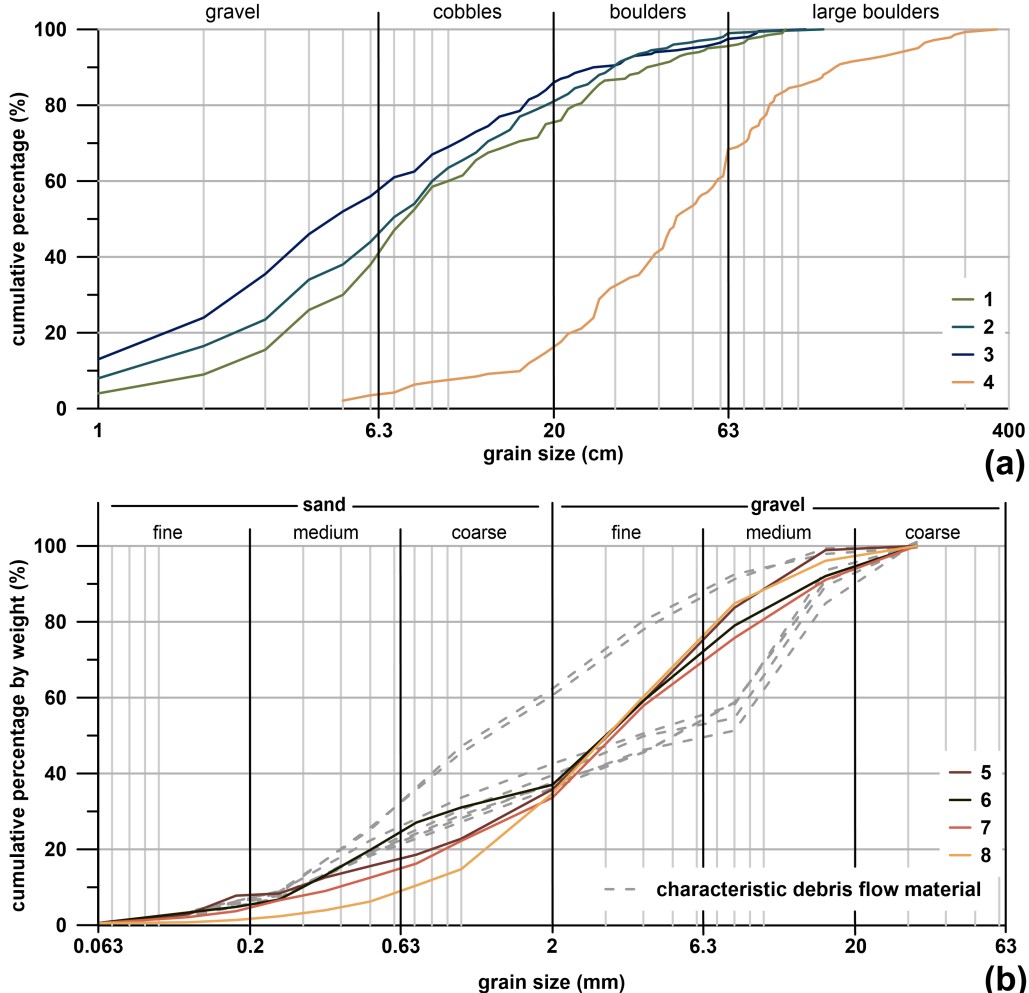

**Figure 6.** Grain-size distributions of Hüttekar Rock Glacier surface layer (a) and front material (b). The corresponding sampling locations are indicated in Fig. 3. Dashed grey lines represent source compositions of a set of experimentally investigated debris flows showing contractive shear response and undrained failure (USGS debris flow flume; Major (1996); Iverson (1997); Iverson et al. (1997)). Classification according to ISO 14688-1 (International Organization for Standardization, 2017).

## 5.2 Preparatory factors

Analysis of multi-temporal ortho-images indicates constant, moderate surface displacement rates. The kinematics of the rock glacier surface show mean annual surface displacement rates of the 200 prominent blocks ranging from 1–50 cm a$^{-1}$ (mean: 14 cm a$^{-1}$) in the observation period of 1970–2020. The position of the rock glacier front fluctuates non-directionally (amplitude 3–7 m), indicating that steady erosion of the front approximately balances rock glacier movement, thus ensuring an invariant long-term front geometry (Kummert and Delaloye, 2018). While destabilization of rock glaciers is frequently characterized






**Table 3.** Climate analysis highlighting the warm and moderately dry conditions prevailing during summer 2019, preceding the debris flow. In addition, air temperature and positive degree day sum show a strong long-term trend, promoting permafrost degradation in Hüttekar-cirque. Angle brackets indicate average values, insignificant trends (at the $\alpha = 0.05$ level) are given in rounded brackets.

| Parameter | Air temperature | Positive degree day sum | Precipitation | Precipitation due to very wet days |
|---|---|---|---|---|
| Summer 2019 | 7.0 °C | 519 °C | 329 mm | 102 mm |
| Summer <1976–2019> | 4.6 °C | 358 °C | 346 mm | 97 mm |
| <2019> | -1.5 °C | 842 °C | 1,348 mm | 359 mm |
| <1976–2019> | -2.5 °C | 653 °C | 1,042 mm | 245 mm |
| Trend magnitude | +0.05 °C a⁻¹ | +7.2 °C a⁻¹ | (+0.13 mm a⁻¹) | (+0.05 mm a⁻¹) |
| Trend significance (*p*-value) | $2.4 \times 10^{-13}$ | $3.3 \times 10^{-7}$ | $(3.7 \times 10^{-1})$ | $(9.9 \times 10^{-1})$ |

by accelerating surface deformation rates and the development of surface discontinuities such as cracks and crevasses, the absence of these features on Hüttekar Rock Glacier does not point towards general destabilization of the rock glacier before
the 2019 debris flow. However, after the debris flow, a cluster of collapse structures connecting the shallow depression hosting the thermokarst lake and the debris flow initiation zone is clearly recognizable (Fig. S2; Government of the Province of Tyrol (2021b)).

Analyzing the long-term (1976–2019) climate signal highlights the distinct hydrometeorological characteristics of the months preceding the failure and the respective long-term trends in Hüttekar-cirque (Table 3). In 2019, the summer months
show exceptionally high air temperatures compared to the long-term average (+2.4 °C). The high positive degree day sum (+161 °C) reflects the large amount of energy available for melting (Table 3). Total precipitation is moderately low (-17 mm) and slightly concentrated on very wet days (+5 mm) with respect to the long-term average (Table 3). The climate data record record shows significantly increasing air temperature (+0.05 °C a⁻¹, exceeding corresponding trends at the global and European Alps scale (+0.02 (±0.01) and +0.03 (±0.02) °C a⁻¹, respectively; Hock et al. (2019))) and positive degree day sum (+7.2 °C
a⁻¹). In contrast, neither total precipitation nor precipitation due to very wet days increase significantly during the 43 years preceding the debris flow (at the $\alpha = 0.05$ level). Recapitulating, the months preceding the debris flow are characterized by warm and dry conditions, on top of an exceptionally strong long-term trend of increasing air temperature in Hüttekar-cirque.

Figure 7 depicts the inter-annual comparison of snowmelt, ice melt and rainfall (normalized to catchment area), indicating moderately wet conditions anteceding the debris flow. Intense snowmelt and ice melt (11.2 and 2.7 mm d⁻¹, respectively)
reflect the high energy available in 2019, in agreement with the high positive degree day sum reported above. The winter season preceding the debris flow is characterized by a prominent snow cover, almost monotonically gaining volume until 29 May 2019, roughly restricting meltwater production to June. The long-lasting snow cover and rapid snowmelt (Fig. 7b) is in excellent agreement with observations throughout Tyrol (Hydrological Service of Tyrol, 2019). Comparing total volumes indicates inconspicuous overall conditions in 2019 (Fig. 7c), acknowledging that the relatively short record precludes statistically
substantiated conclusions.





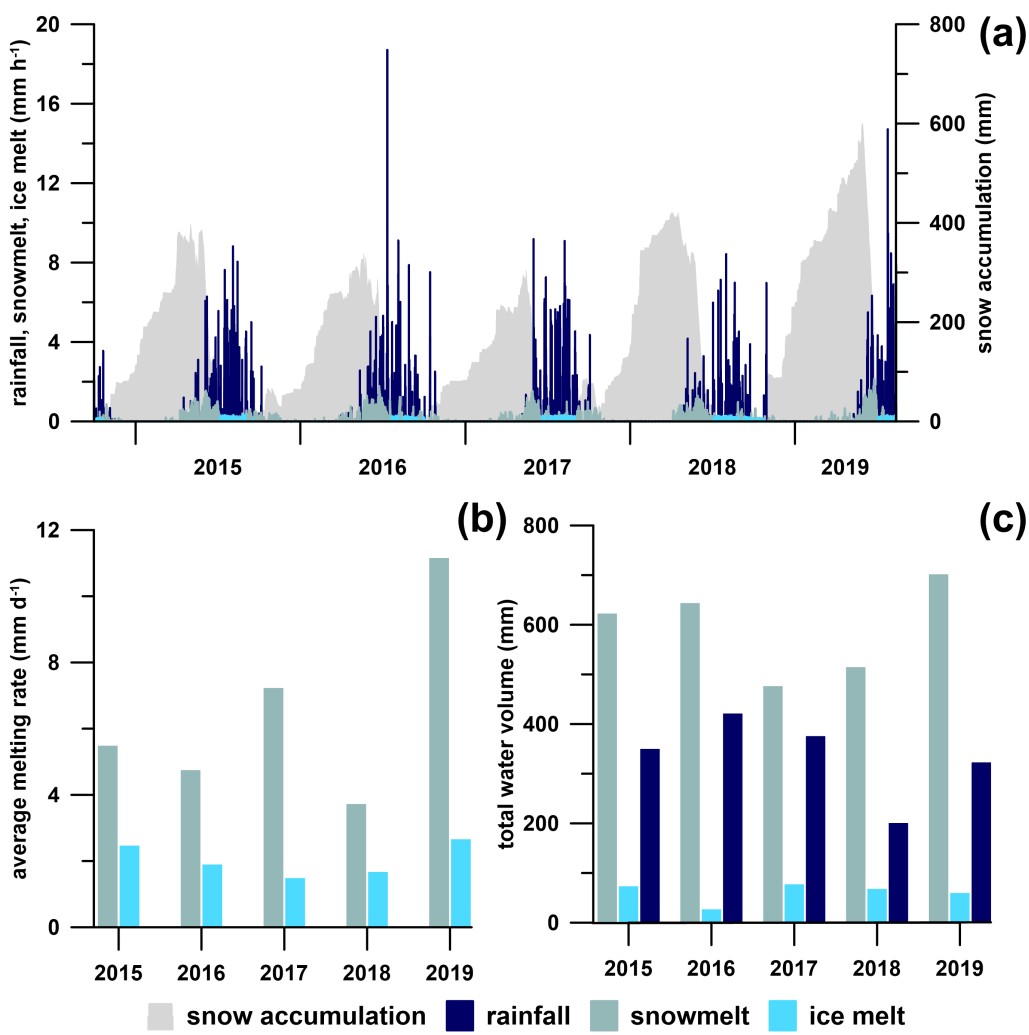

**Figure 7.** (a) Time series of rainfall, snowmelt and ice melt in Hüttekar based on INCA (Haiden et al., 2011) and SNOWGRID (Olefs et al., 2013). The snow cover dynamics in spring 2019 differ from the preceding years by the late and rapid snowmelt. (b) Average snowmelt and ice melt rates for individual years (linear approximation). Spring 2019 is characterized by exceptionally high snowmelt rates. (c) Total water volumes (between 01 January and 13 August) per year, attributed to rainfall, snowmelt and ice melt, respectively. Despite the different snow cover dynamics in spring 2019, the total meltwater volume exceeds the respective volume of the preceding years only slightly.

## 5.3 Triggering factors

The rainfall event preceding the debris flow lasted for 74 h at an average intensity of 0.54 mm h$^{-1}$ (total rainfall volume ~40 mm). The frequency analysis shows that 12 % of earlier rainfall events hitting Hüttekar-cirque exceeded it in severity (Fig. 8). The event does not exceed the regional critical rainfall thresholds for debris flow initiation during the summer season with





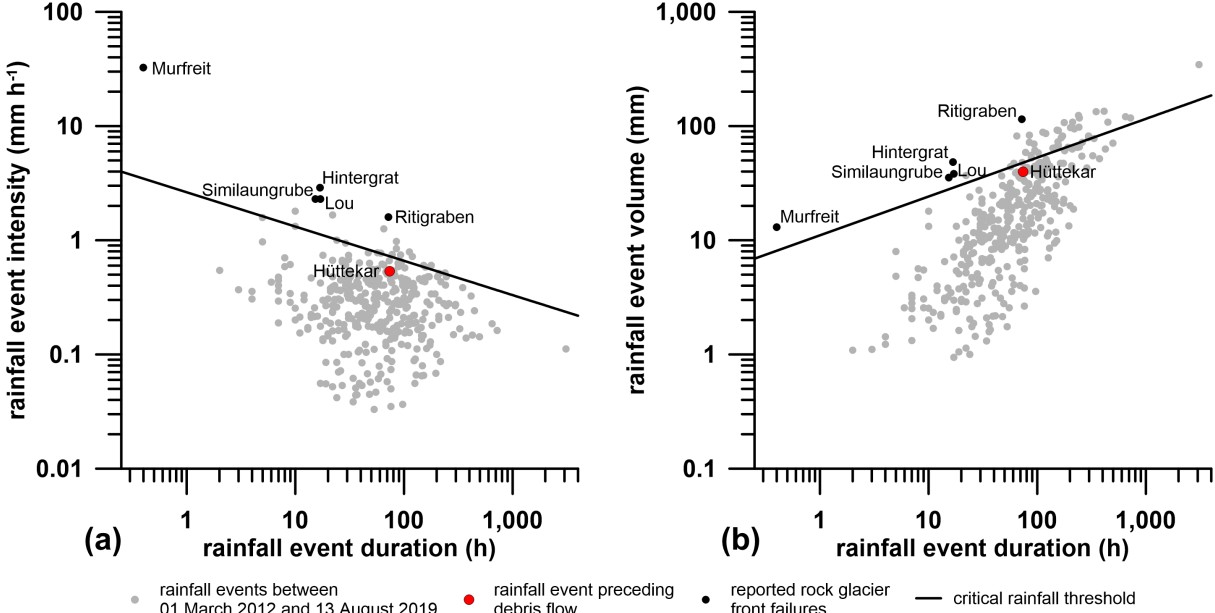

**Figure 8.** Rainfall event frequency analysis with respect to (a) rainfall intensity-duration and (b) rainfall volume-duration based on INCA (Haiden et al., 2011). The rainfall event preceding the Hüttekar debris flow (red dot) was neither especially intense nor especially persistent with respect to earlier rainfall events hitting Hüttekar-cirque (grey dots), critical rainfall thresholds (black lines) and rainfall events triggering documented rock glacier front failures in the European Alps (black dots) (note logarithmic scale). Regional thresholds describing minimum conditions for rainfall-induced debris flows are based on Nikolopoulos et al. (2015b) and Marra et al. (2016). Characteristics of documented triggering events are based on Lugon and Stoffel (2010); Krainer et al. (2012); Kofler et al. (2021); Marcer et al. (2020).

respect to intensity-duration (Fig. 8a; Nikolopoulos et al. (2015b)) and volume-duration (Fig. 8b; Marra et al. (2016)), although these are regarded conservative estimates (Wieczorek and Guzzetti, 2000; Guzzetti et al., 2007). These characteristics distinctly contrast with the rainfall events preceding well documented rock glacier front failures in the European Alps that occurred in response to heavy rainfall (Fig. 8): the Ritigraben event on 24 September 1993 (Lugon and Stoffel, 2010), the Murfreit event on 02 July 2003 (Krainer et al., 2012), the Hintergrat and Similaungrube events, both on 13 August 2014 (Kofler et al., 2021),

as well as the Lou event on 14 August 2015 (Marcer et al., 2020).

Field observations as well as satellite data analysis detect the rapid evolution and drainage of the thermokarst lake that was formed on the rock glacier surface. Sentinel-2 satellite imagery indicates water pooling in the depression on 03 June 2019, when snow still covered large parts of the rock glacier catchment (520 mm average snow water equivalent based on SNOWGRID). The thermokarst lake evolved over a period of 10 weeks, but drained within hours through a newly formed

channel, indicating that the characteristic time scale of thermokarst breakthrough is shorter than the summer season (Fig. 4e-h, Fig. S2). Comparing snow cover and lake development shows that the latter gained most of its volume during the intense snowmelt period in June (Fig. 8a). Linear interpolation of the lake surface area between the cloud-free satellite imagery dates





**Table 4.** Thermokarst lake development based on INCA (Haiden et al., 2011), Sentinel-2 satellite data (Sentinel Hub, 2020) and the most recent (2017) high-resolution DTM of the rock glacier surface (Government of the Province of Tyrol, 2021a).

| Date | Surface area (m²) | Water volume (m³)[a] | Mean water depth (m) | Total energy input (J)[b] | Total ice melt (m³)[b] |
|---|---|---|---|---|---|
| 01 June 2019 | 0 | 0 | 0 | 0 | 0 |
| 03 June 2019 | 1,520 | 1,320 | 0.9 | $2.9 \times 10^{10}$ | 100 |
| 13 June 2019 | 21,370 | 68,340 | 3.2 | $1.6 \times 10^{12}$ | 5,410 |
| 16 June 2019 | 26,380 | 97,020 | 3.7 | $2.7 \times 10^{12}$ | 8,850 |
| 18 June 2019 | 28,570 | 110,760 | 3.9 | $3.4 \times 10^{12}$ | 11,180 |
| 26 June 2019 | 31,670 | 131,870 | 4.2 | $6.3 \times 10^{12}$ | 20,940 |
| 28 June 2019 | 34,010 | 148,290 | 4.4 | $7.3 \times 10^{12}$ | 24,220 |
| 01 July 2019 | 35,650 | 158,770 | 4.5 | $8.8 \times 10^{12}$ | 29,380 |
| 06 July 2019 | 36,400 | 162,370 | 4.5 | $1.1 \times 10^{13}$ | 36,280 |
| 16 July 2019 | 36,960 | 166,040 | 4.5 | $1.4 \times 10^{13}$ | 47,050 |
| 23 July 2019 | 36,960 | 166,040 | 4.5 | $1.7 \times 10^{13}$ | 57,550 |
| 26 July 2019 | 36,960 | 166,040 | 4.5 | $1.9 \times 10^{13}$ | 63,090 |
| 13 August 2019 | 36,960 | 166,040 | 4.5 | $2.6 \times 10^{13}$ | 87,480 |

[a]estimate based on DTM (2017) [b]since 01 June 2019

enables the calculation of the total energy input to the lake (Table 4). While the area gain slowed down at the beginning of July, the energy input was considerable thereafter, allowing for extensive ice melt beneath the surface. The total amount of ice melt preceding the failure sums to ∼87,000 m³ (corresponding energy input $2.6 \times 10^{13}$ J). The impact of energy available for thermokarst evolution is emphasized by dividing this value by the distance to the debris flow initiation zone (∼350 m), and assuming a channel of circular cross section, resulting in an upper bound estimate of 18 m for the channel diameter. While only a fraction of this energy is actually involved in channel development, the resulting order of magnitude clearly demonstrates the potential of thermokarst evolution for triggering the lake outburst during the course of a single summer season. Dividing the latest lake water volume estimate (∼166,000 m³) by the drainage time (∼1 day) indicates a drainage rate on the order of ∼m³ s⁻¹, supplying a substantial amount of water to the debris flow initiation zone. In contrast to many hazardous glacial lakes evolving over time scales of years to decades (Haeberli, 1983; Mölg et al., 2021), the thermokarst lake in Hüttekar-cirque unfolded its destructive power already two months after its formation.

## 6   Discussion

While the absence of instrumentation in Hüttekar-cirque during the failure inhibits a detailed reconstruction of the debris flow initiation mechanism, the relative importance of destabilizing factors can confidently be evaluated.




Unfavorable topographical and sedimentological predisposition put the rock glacier front in a state susceptible to debris flow initiation. Its steep slope angle supports shear stress and sensitivity to liquefaction while lowering shear strength (Iverson et al., 1997; Chowdhury et al., 2010; Reichenbach et al., 2018). Convex topography generally induces extensive deformation patterns

favoring the development of tensional stresses (Roer et al., 2008; Delaloye et al., 2013; Marcer et al., 2019). Since the rock glacier did not show discernible signs of destabilization prior to debris flow initiation, external drivers including the strong increase in air temperature and positive degree day sum, along with available snow and ice meltwater and their pronounced impact on permafrost during the months preceding the debris flow event are considered crucial preparatory factors (Table 3). Rock glacier creep balancing erosion rates in combination with a high energy environment favoring permafrost degradation at

the rock glacier front provide a large accumulation of loose, unfrozen sediment that is susceptible to mobilization down the adjacent steep slope (Fig. 5, Fig. S2).

Debris slides within saturated, loose sediment responding contractive to shear deformation are prone to mobilize into debris flows (Savage and Baum, 2005). Collapse of the soil structure and crushing of particles drives the development of transient excess pore-water pressures in parts of the deforming mass, which reacts by disintegration and acceleration (Hutchinson and

Bhandari, 1971; Iverson and Major, 1986; Anderson and Sitar, 1995; Iverson, 1997, 2005, 1997; Sassa and Wang, 2005). Within the displaced mass, the rates of excess pore-water pressure generation and dissipation critically depend on strain rate, bulk density, and grain size distribution (Iverson, 1997, 2005; Iverson et al., 1997). Poorly sorted sand and gravel, such as the Hüttekar Rock Glacier material (Fig. 6b), are susceptible to undrained deformation and attendant pore-water pressure coevolution, demonstrating the susceptibility of the rock glacier to debris flow mobilization (Iverson, 1997, 2005; Iverson

et al., 1997). The relative time scales for pore space contraction and pore-water pressure diffusion govern the persistence of high pore-water pressures (Iverson, 1997; Iverson and LaHusen, 1989; Iverson et al., 1997). The diffusion time scale of Hüttekar Rock Glacier debris is relatively short (on the order of ∼20 s, estimated from similar material investigated at the USGS debris flow flume, Fig. 6b; Iverson (1997)). Sustaining pore-water pressures high enough to keep debris flow surges in motion therefore requires rapid and persistent delivery of large amounts of water (on the order of a few m$^3$ s$^{-1}$).

Frequency analysis of rainfall events hitting Hüttekar-cirque indicates that the event immediately preceding the debris flow was not exceptionally severe, falling below the critical threshold for failure initiation (Fig. 8). Regarding the moderately dry hydrometeorological conditions during summer 2019 (Table 3), these observations suggest that the storm immediately preceding the event fails to provide the large amounts of water necessary to initiate and sustain the debris flow for several hours. Considering the coincident thermokarst lake outburst, following exceptionally warm weeks characterized by considerable energy

input promoting melting processes and permafrost degradation (Table 3, Table 4), a multiple trigger mechanism is regarded plausible instead.

The key issue in terms of debris flow mechanics is the rate of water transport to the initiation zone. Field evidence suggests that water flowed concentrated along a newly formed channel network from the thermokarst lake to the rock glacier front (Fig. 4e, Fig. 4f, Fig. 4h, Fig. S2). The rapid development of this efficient drainage network was facilitated by the short

timescale of advective heat transport along the meltwater channels. Driven by thermal convection, the establishment of such a channel system was possible within several weeks, despite the considerable distance of ∼350 m. The energy provided by the





thermokarst lake strikingly exceeded the latent heat of fusion necessary for channel evolution. Once established, this channel network provided the capability to transport large amounts of water from the thermokarst lake to the debris flow initiation zone within a short time. Water flow velocities along these highly permeable flow paths may reach up to several cm s[-1] (Tenthorey,

1992; Krainer and Mostler, 2002; Buchli et al., 2013; Wagner et al., 2021b), providing sufficient water to the debris flow to keep the deforming sediment saturated. Consequently, rapid drainage emptied the lake within hours, without apparent prior indications of a lake outburst.

With respect to GLOF hazard evaluation, the decisive factor is that thermokarst development and breakthrough happened within an extremely short time scale (on the order of weeks), driven by the high energy input during summer 2019. In terms of

mechanical properties, the ice-debris mixture composing the rock glacier represents a transitional form between the common glacier dammed and moraine dammed lakes (Clague and Evans, 1994; Huggel et al., 2004; Schaub, 2015). The Hüttekar Rock Glacier failure mechanism involved drainage channel enlargement as well as collapse of the rock glacier front, demonstrating that thermokarst development and slope failure operated synergistically. Since their contrasting time scales (~weeks for channel enlargement, ~seconds to hours for rock glacier front collapse) are commonly associated with different materials (ice and

debris, respectively), the integration of both mechanisms distinguishes rock glaciers from glaciers and moraines in terms of GLOF hazard assessment. The large amounts of debris provided by the rock glacier favored the evolution of the outburst flood into a debris flow.

Subsequently, the hazard cascade triggered in Hüttekar-cirque was expanded by an additional element when the displaced mass reached and blocked Radurschlbach, impounding a voluminous (~120,000 m$^3$) lake that threatened a large area down-

stream. Acknowledging that landslide dams frequently fail within a short period (hours to months) after their formation, and that common failure mechanisms including overtopping, piping and mechanical collapse are capable of triggering severe outburst floods, prediction and instant detection of these events in remote areas is crucial for risk assessment in mountainous areas (Clague and Evans, 1994; Ermini and Casagli, 2003; Schaub, 2015).

## 7 Conclusions

The 2019 Hüttekar debris flow impounded the main river of Radurschl Valley, threatening the downstream community and infrastructure by dam breakthrough. The rock glacier front failure was caused by an upstream thermokarst lake outburst and mobilized into a catastrophic debris flow displacing 40,000–50,000 m$^3$ within several hours. Analyzing a comprehensive set of potentially destabilizing factors reveals critical combinations of environmental influences that govern multi hazard characteristics in a complex periglacial setting. Evaluating the topographical and sedimentological context, quantifying rock glacier move-

ment rates preceding the failure, and analyzing climate and weather signals demonstrates the capability of rapid thermokarst evolution to induce highly hazardous situations at short time scales. In combination with challenges regarding hazard detection and prediction in remote areas, this complicates integrated multi hazard assessment (Kappes et al., 2012; Gallina et al., 2016). In this context, the combination of several raster data sets comprising terrain models, satellite imagery, and gridded climate and



weather variables proofed a valuable tool for assessing the individual impact of destabilizing factors in complex mountainous
terrain.

The observed failure differs from rock glacier front failures documented so far. Not only the amount of rainfall, but its combination with the thermokarst lake development, and the evolution of a drainage system within the rock glacier caused the debris flow. The fact that a comparable thermokarst lake had never been observed before on Hüttekar Rock Glacier, and that the outburst occurred only two months after its formation imply that comparable lake formation and outburst are hardly predictable.
As climate change progresses, this mechanism will likely gain importance due to accelerated permafrost degradation and increased energy available for thawing, altering ground properties and increasing the likelihood of thermokarst lake formation (Kääb and Haeberli, 2001; Patton et al., 2019). Since many rock glacier fronts exhibit grain size distributions similar to the Hüttekar Rock Glacier (Johnson, 1992; Haeberli and Vonder Mühll, 1996; Arenson et al., 2002; Krainer et al., 2010, 2012), they pose widespread multi hazard elements in mountainous regions, with significantly increased risk if rivers or infrastructure
pass potential runout zones of debris flows.

The observed debris flow mechanism is the most frequent one in colluvium, which covers most of the surface in mountainous terrain (Turner, 1996). Thus, we suggest that the results obtained in this study apply not only to rock glaciers, but to a wide range of colluvium in mountainous, permafrost-affected landscapes, with accordingly wide implications. Including rapid thermokarst evolution in landslide hazard assessment gains importance as climate change progresses, altering the boundary
conditions for periglacial landslides across the European Alps and mountain ranges around the world. The developed process-based understanding of the analyzed hazard cascade including thermokarst lake outburst, rock glacier front failure, debris flow evolution, and river blockage provides an expedient tool for multi hazard identification.

*Data availability.* Input data for this study are freely available. Meteorological data sets (SPARTACUS, INCA) are provided by the Central Institute for Meteorology and Geodynamics (https://data.hub.zamg.ac.at/, last access: 28 June 2022). The Government of the Province of
Tyrol provides the digital terrain model as well as current and historical ortho-images (https://www.data.gv.at/, last access: 28 June 2022). Sentinel-2 multi-spectral satellite data are provided by Copernicus and processed by Sentinel Hub (https://www.sentinel-hub.com/, last access: 28 June 2022).

**Appendix A**

Ice ablation rates $m$, given as ice water equivalent (m s$^{-1}$), are calculated using a surface energy balance approach (Cuffey and
Paterson, 2010):

$$m = \frac{Q_r + Q_h + Q_e + Q_p}{\lambda_f \rho_w} \tag{A1}$$

where $Q_r$ is net radiation (W m$^{-2}$), $Q_h$ is the sensible heat flux (W m$^{-2}$), $Q_e$ is the latent heat flux (W m$^{-2}$), $Q_p$ is the heat flux associated with precipitation (W m$^{-2}$), $\lambda_f$ is the latent heat of fusion ($3.34 \times 10^5$ (J kg$^{-1}$), and $\rho_w$ is the density of water (kg




m$^{-3}$) at 273.15 K. Equation (A1) assumes the glacier surface to stay at 273.15 K and neglects transfer of heat within the glacier,

both assumptions considered reasonable approximations for the ablation zones of temperate glaciers during the summer season (Hock, 2005; Cuffey and Paterson, 2010). For the operational purposes of this study, estimates of the individual energy flux contributions aim at evaluating their variation between different years rather than yielding precise absolute values. Net radiation is calculated according to (Cuffey and Paterson, 2010):

$$Q_r = Q_s(1 - \alpha) + \epsilon_a \sigma T_a^4 - \epsilon_s \sigma T_s^4 \tag{A2}$$

where $Q_s$ is incoming shortwave radiation (W m$^{-2}$), $T_a$ is air temperature (K), $T_s$ is surface temperature (K), $\alpha$ is broadband surface albedo (-), $\epsilon_a$ is atmospheric emissivity (-), $\epsilon_s$ is surface emissivity (-), and $\sigma$ is the Stefan-Boltzmann constant (5.67 × 10$^{-8}$ W m$^{-2}$ K$^{-4}$). As a first approximation, $\epsilon_s \approx 0.95$ for ice and water, while $\alpha \approx 0.4$ and $\alpha \approx 0.06$ for ice and water, respectively (Cuffey and Paterson, 2010). Incoming shortwave radiation is given by the superposition of direct and diffuse radiation, accounting for cloudiness (INCA) and the shadowing effect of surrounding terrain, as well as by shortwave radiation

reflected by the surrounding terrain ($\alpha \approx 0.2$, GRASS GIS 8.0, Hofierka et al. (2007)). Atmospheric emissivity is parameterized as outlined by Greuell et al. (1997) and Oerlemans (2000). Sensible and latent heat fluxes are calculated by parameterization of the respective transport processes based on turbulence similarity assuming a roughness length of 10$^{-2}$ m (Brutsaert, 2005; Brock et al., 2006; Cuffey and Paterson, 2010). The sensible heat flux is given by:

$$Q_h = c_a \rho_a C_h u (T_a - T_s) \tag{A3}$$

where $c_a$ is the specific heat capacity of air at constant pressure (1006 J kg$^{-1}$ K$^{-1}$), $C_h$ is the heat transfer coefficient (Stanton number) (-), $u$ is windspeed (m s$^{-1}$), and $\rho_a$ is air density (kg m$^{-3}$). The latent heat flux is obtained using:

$$Q_e = \lambda_v \rho_a C_e u (x_a - x_s) \tag{A4}$$

where $C_e$ is the water vapor transfer coefficient (Dalton number) (-), $x_a$ is the specific humidity of air (-), $x_s$ is the specific humidity at the surface (-), and $\lambda_v$ is the latent heat of vaporization (2.5 × 10$^6$ J kg$^{-1}$). The heatflux associated with precipitation

is estimated assuming that the rain droplet temperature $T_d$ approaches the near-surface air temperature:

$$Q_p = c_w \rho_w P (T_d - T_s) \tag{A5}$$

where $c_w$ is the specific heat capacity of water at constant pressure (4219 J kg$^{-1}$ K$^{-1}$), and $P$ is the precipitation intensity (m s$^{-1}$).

The plausibility of the caluclated ablation rates is evaluated by comparing them to a set of glacier ablation rates and associated

energy flux contributions across the European Alps (Table A1). The results obtained for Hüttekarferner and Glockturmferner



(Table A2) fall within the range given by this sample, reflecting the northward exposition and steep slope of both glaciers by comparatively smaller total ablation rates.

**Table A1.** Ablation rates and energy flux contributions measured and calculated at various glaciers across the European Alps (Escher-Vetter, 1985; Greuell and Oerlemans, 1989; van de Wal et al., 1992; Greuell et al., 1997; Arnold et al., 1996; Oerlemans, 2000; Brock et al., 2000; Klok and Oerlemans, 2002; Oerlemans and Klok, 2002; Willis et al., 2002; Brock et al., 2006; Oerlemans et al., 2009). Values represent averages taken over the outlined period (altitude Alt (m a.s.l.), slope Sl (°), aspect Asp, measured ice ablation rate $m$ (cm day$^{-1}$), given as water equivalent assuming an ice density of 900 kg m$^{-3}$, total energy flux density $Q$ (W m$^{-2}$), net shortwave radiation $Q_s$ (W m$^{-2}$), net longwave radiation $Q_l$ (W m$^{-2}$), sensible heat flux $Q_h$ (W m$^{-2}$), and latent heat flux $Q_e$ (W m$^{-2}$), Austria AT, Switzerland CH). The (small) precipitation heat flux is rarely reported, thus not included.

| Location | Alt | Sl | Asp | Period | $m$ | $Q$ | $Q_s$ | $Q_l$ | $Q_h$ | $Q_e$ |
|---|---|---|---|---|---|---|---|---|---|---|
| Haut Glacier d'Arolla (Pennine Alps, CH) | 2,964 | 16 | N | 30 May 1990–28 August 1990 | 3.8 | 101 | 130 | -52 | 26 | -3 |
| Haut Glacier d'Arolla (Pennine Alps, CH) | 2,964 | 16 | N | 10 August 1993–25 August 1993 | 6.9 | 167 | 165 | -20 | 10 | 12 |
| Hintereisferner (Ötztal Alps, AT) | 3,037 | 16 | NE | 12 July 1986–22 July 1986 | 5.1 | 210 | 236 | -45 | 22 | -3 |
| Hintereisferner (Ötztal Alps, AT) | 3,037 | 16 | NE | 13 July 1989–30 July 1989 | 7.9 | 158 | 155 | -8 | 32 | -21 |
| Pasterze[a] (Glockner Range, AT) | 2,914 | 16 | E | 18 June 1994–7 August 1994 | 5.9 | 232 | 194 | -16 | 45 | 9 |
| Pasterze[b] (Glockner Range, AT) | 2,914 | 16 | E | 21 June 1994–12 August 1994 | 6.3 | 236 | 185 | -17 | 56 | 12 |
| Pasterze[c] (Glockner Range, AT) | 2,914 | 16 | E | 19 June 1994–12 August 1994 | 6.5 | 242 | 199 | -19 | 53 | 9 |
| Pasterze[d] (Glockner Range, AT) | 2,914 | 16 | E | 15 June 1994–16 August 1994 | 3.0 | 109 | 119 | -33 | 20 | 3 |
| Pasterze[e] (Glockner Range, AT) | 2,914 | 16 | E | 15 June 1994–16 August 1994 | 2.0 | 87 | 109 | -39 | 19 | -2 |
| Vadret da Morteratsch (Bernina Range, CH) | 3,079 | 21 | N | 01 October 1995–30 September 1998 | 1.6 | 191 | 177 | -25 | 31 | 8 |
| Vadret da Morteratsch (Bernina Range, CH) | 3,079 | 21 | N | 01 June 1999–31 August 1999 | 4.0 | 171 | 139 | -17 | 37 | 12 |
| Vadret da Morteratsch (Bernina Range, CH) | 3,079 | 21 | N | 01 June 2000–31 August 2000 | 4.7 | 166 | 140 | -27 | 43 | 10 |
| Vadret da Morteratsch (Bernina Range, CH) | 3,079 | 21 | N | 01 June 2001–31 August 2001 | 4.8 | 179 | 147 | -29 | 49 | 12 |
| Vadret da Morteratsch (Bernina Range, CH) | 3,079 | 21 | N | 01 June 2002–31 August 2002 | 4.9 | 185 | 144 | -22 | 48 | 15 |
| Vadret da Morteratsch (Bernina Range, CH) | 3,079 | 21 | N | 01 June 2003–31 August 2003 | 5.7 | 240 | 173 | -24 | 70 | 21 |
| Vadret da Morteratsch (Bernina Range, CH) | 3,079 | 21 | N | 01 June 2004–31 August 2004 | 4.7 | 189 | 158 | -24 | 42 | 13 |
| Vadret da Morteratsch (Bernina Range, CH) | 3,079 | 21 | N | 01 June 2005–31 August 2005 | 4.9 | 203 | 173 | -23 | 42 | 11 |
| Vadret da Morteratsch (Bernina Range, CH) | 3,079 | 21 | N | 01 June 2006–31 August 2006 | 5.5 | 215 | 181 | -23 | 50 | 7 |
| Vernagtferner (Ötztal Alps, AT) | 3,166 | 14 | S | 01 June 1982–30 September 1982 | 1.4 | 57 | 82 | -24 | 38 | -39 |

[a] weather station at 2205 m a. s. l. [b] weather station at 2310 m a. s. l. [c] weather station at 2420 m a. s. l. [d] weather station at 2945 m a. s. l. [e] weather station at 3225 m a. s. l.

*Author contributions.* SK, TW, KK, and GW designed the study and prepared the initial draft. Formal analysis, methodology development, data collection, and visualization were performed by SK, TW, KK, MA and MO. MA, MO and KH edited and complemented the manuscript draft. GW managed and coordinated the project, including the funding acquisition.




**Table A2.** Ablation rates and energy flux contributions calculated in this study. Values represent averages taken over the outlined period (altitude Alt (m a.s.l.), slope Sl (°), aspect Asp, measured ice ablation rate $m$ (cm day$^{-1}$), given as water equivalent assuming an ice density of 900 kg m$^{-3}$, total energy flux density $Q$ (W m$^{-2}$), net shortwave radiation $Q_s$ (W m$^{-2}$), net longwave radiation $Q_l$ (W m$^{-2}$), sensible heat flux $Q_h$ (W m$^{-2}$), and latent heat flux $Q_e$ (W m$^{-2}$)).

| Location | Alt | Sl | Asp | Period | $m$ | $Q$ | $Q_s$ | $Q_l$ | $Q_h$ | $Q_e$ |
|---|---|---|---|---|---|---|---|---|---|---|
| Glockturmferner | 2,855 | 27 | N | 01 June 2015–31 August 2015 | 2.5 | 90 | 84 | -14 | 28 | -8 |
| Glockturmferner | 2,855 | 27 | N | 01 June 2016–31 August 2016 | 2.2 | 77 | 77 | -17 | 19 | -2 |
| Glockturmferner | 2,855 | 27 | N | 01 June 2017–31 August 2017 | 2.5 | 91 | 78 | -11 | 31 | -7 |
| Glockturmferner | 2,855 | 27 | N | 01 June 2018–31 August 2018 | 2.2 | 79 | 79 | -16 | 21 | -5 |
| Glockturmferner | 2,855 | 27 | N | 01 June 2019–31 August 2019 | 2.6 | 97 | 82 | -12 | 30 | -3 |
| Hüttekarferner | 3,043 | 21 | N | 01 June 2015–31 August 2015 | 3.9 | 145 | 135 | -9 | 32 | -13 |
| Hüttekarferner | 3,043 | 21 | N | 01 June 2016–31 August 2016 | 3.5 | 127 | 123 | -12 | 24 | -8 |
| Hüttekarferner | 3,043 | 21 | N | 01 June 2017–31 August 2017 | 3.7 | 139 | 125 | -6 | 32 | -12 |
| Hüttekarferner | 3,043 | 21 | N | 01 June 2018–31 August 2018 | 3.6 | 133 | 127 | -10 | 26 | -10 |
| Hüttekarferner | 3,043 | 21 | N | 01 June 2019–31 August 2019 | 3.9 | 146 | 134 | -8 | 27 | -7 |

*Competing interests.* The authors declare that they have no conflict of interest.

*Acknowledgements.* This research was funded by the Austrian Federal Ministry of Agriculture, Regions and Tourism and the Federal Provinces of Vorarlberg, Tyrol, Salzburg, Styria and Carinthia within the DaFNE project RG-AlpCatch. We thank Josef Waldner and Gerhard Schaffenrath for witness reports, local information, discussion, and photographs. Additional photographs and information were kindly
provided by Roman Außerlechner, Thomas Figl, Werner Thöny (Geological Survey of Tyrol), Rudolf Philippitsch, Danny F. Minahan, Ainur Kokimova, and Irène Apra. The authors acknowledge the financial support by the University of Graz.



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
