# Peer review of "The role of thermokarst evolution in debris flow initiation (Hüttekar Rock Glacier, Austrian Alps)"

_EGUsphere, 2022_

## Referee Comment (RC1)

**By anonymous referee**

**General comments:**

The manuscript investigates a cascade of processes in the Radurschl Valley in the Austrian Alps. The outburst of a (supra-) glacial lake initiated the event, which continued with the failure of a rock glacier front with subsequent debris flow initiation. The deposit of the latter caused the blocking of the valley and the formation of a lake, which was drained to manage the risk related to its possible outburst. The series of events and their description are very interesting and, in my opinion, would enrich the scientific literature. However, I have some concerns regarding the manuscript, which in my opinion could profit from some revisions.

The methods are described at a level of detail that is not sufficient for understanding the analysis and reproduce the study. In this sense, they need to be revised and improved. The results and the discussion are mixed at several points in the text, with several repetitions and omissions. The logic structure is at times difficult to follow.

As a general suggestion, I would welcome a major revision of the text. Firstly, the authors could more effectively highlight the original analysis and results of the study, avoid substantial repetitions, and increase the logic structure of the manuscript. Secondly, the use of technical language should be always preferred to common-language expressions (e.g. "temporal changes" vs "dynamics", see detailed comments later) in all cases where this is possible and the methods should be described in more detail. Some minor analysis and addition to the data presented might improve the manuscript and its relevance in the community.

In the following, I provide some more specific comments, that I believe could improve the manuscript. I hope that the authors can receive the feedback in the in the same spirit in which it was written and for what it is: constructive suggestion to improve the paper.

**Some other comments (non-comprehensive):**

Page 1, Line 1: Is the front of the rock glacier failing or are the debris accumulated in the gully (below the rock glacier) that are triggered and flow downhill? I write the comment here, but it applies to several other sections in the manuscript. I think that there is a distinction between rock glacier failure, active layer detachment, and debris flow initiation at a rock glacier front. Neglecting the differences between all these processes is in my opinion a first-order approach, which might well be justified in certain contexts, but should not be used in this manuscript, where the focus is on this specific topic.

Page 1, Line 3: What does "multivariate permafrost degradation" mean? What is "retrogressive debris flow"?

Page 1, Line 5: What do you mean by "environmental forces" and what are the "ambiguous conditions"?

Page 1, Line 6: The paper is an application of the methodology presented in Glade and Crozier, 2005. If I understood this correctly, the authors do not establish a basis for multi-hazard assessment but apply an existing method. If contrarily the authors argue that they implemented the methods for hazard assessment, I would welcome a clearer explanation of the original methodology, particularly with regards to the concept of hazard.

Page 2, Line 55: I would argue that destabilizing rock glaciers are defined by the combination of abnormal velocities and the presence of surface features such as cracks and scarps. Several recent papers in the literature provide a more accurate description of rock glacier destabilization.

Page 3, Line 63: reading this line, the expectations to this paper sky-rocketed. Unfortunately, I must say that they were not satisfied by the manuscript, especially with regards to hydrogeological and mechanical aspects (some comments later). With this I don't want to suggest major revisions, but simply to be more precise in the description of the goals of the paper.

Page 4, Line 96: This sentence is in my opinion very controversial. Are the authors stating that the rock glacier originated 170 years ago as a result of the LIA glacier advance? The next sentence quickly describes the surface morphology of the terrain influenced by recent glacier advances. Why not considering the simplest and most-supported hypothesis that the rock glacier pre-existed and was simply overridden by a glacier during the LIA? Anyway, I would simply avoid this argument, which is irrelevant for the discussion of the results and the topic presented in the paper. If the authors want to keep the statement, I would welcome more extensive explanations of their theory.

Page 5, Line 105: how do the authors know that the channels are eroded into the ice core? Are the authors referring to water flowing on top of the permafrost table or to water infiltrating into the ice-core? This is a very interesting topic and if observations are available, they could be presented and discussed in more detail. If no more details are available, I would still welcome a clearer description of the observations (if considered relevant).

Page 6, Line 106: what is a "retrogressive debris flow"? I see here some confusion between "retrogressive erosion" and "debris flow".

Page 6, Line 111 and following: This paragraph is extremely interesting, and I suggest to extend the description (here) and discussion (later) of the observations, the numbers presented, the methodology used to derive them, and their uncertainties. These points might be extended in several locations in the manuscript, maybe replacing some of the repetitions and digressions that are currently present.

Page 7, Line 123: Concurrently means at the same time. The question of the timing is essential for the process described. Please explain in more detail what is known (and how, see previous comment) and what are your hypothesis. Later in the manuscript, they should be tested and discussed.

Page 7, Line 126: it is interesting that local knowledge is taken into account. Still, I suggest to perform a systematic analysis with modern remote sensing data (Sentinel 2 for example) to quantify the evolution of the lake in recent times.

Page 7, Line 139: Are the authors talking about dynamic forces here? Impact from external mass movements? Earthquakes?

Page 8, from Figure 4:

Very interesting observations of the rock glacier front. What's the height of the shear horizon depth here? I would be very happy to see in the published version of the manuscript a detailed description of this area (possibly before and after the event?), which is highly relevant to the debris flow initiation.

A figure showing a simple comparison before-after the event (maybe in appendix?) would be very useful.

What is the thickness of the debris around the lake? And around the front? Panel g and h seem to indicate a debris covered glacier. Would It be possible to provide a more detailed geomorphological assessment of the landform and a map of it?

Page 9, Section 4.1: some of the text presented here might be more suited for the introduction. Also, the analysis of the presented factors on slope stability is not clear to me. I don't see any stability assessment, or somehow I missed it. Maybe a clearer description of the methods might help.

Page 9, Line 165 and following: This is in my opinion an interesting approach, but cannot be introduced so lightly. Please provide more details, assumptions, rationale, and potential limitations. As known, the internal structure of rock glacier is very different from the active layer (also not homogeneous in the active layer!). Assuming the the latter is representative of the first is not valid in my opinion, especially given the photographs presented in figure 4. Were the boulders only collected at the surface? Only evaluating the boulders at the surface creates a bias towards boulders. How were the samples collected at the front? Who went there? Where exactly were they collected? How representative should this be of the internal part of the rock glacier, especially the shear horizon if you suggest that water was flowing there (erosion/deposition at the surface)?

Page 9, line 177: this is not a sufficient description for the data used.

Page 10, line 183 and following: is the use of climate relevant here? Are the authors not limiting the analysis to meteorological forcing? Also, what is meant by "progressive change of climatic factors"? In general, the description of the method is not sufficient for reproduction, and in this sense, it should be improved and extended.

Page 10, Line 210: are the calculations about snow cover validated in any way?

Page 11, Line 218: The statements about the hydrology seems to be results. What observations are available? Please expand the description of the data and methods here. E.g. why until the 13th of August?

Page 14, line 308: Data of surface velocities would be very interesting. Is it possible to present them? Especially relevant if the authors speak about destabilization. Possibly you could also see some signal in ice melt (vertical component – once subtracted subsidence due to dynamics).

Page 17, Figure 8: great figure. I suggest to add the equations to the lines and expand the discussion with regards to failure processes (Lou and Similaun different from this case, no?)

Discussion and Conclusions: I would welcome a more detailed discussion of the original results of the manuscript, limiting long review of previous literature. Same for the conclusions. Maybe some of the comments above can help in the process.

---

## Referee Comment (RC2)

Review of the article

"The role of thermokarst evolution in debris flow initiation (Huttekar Rock Glacier, Austrian Alps)"

by Simon Kainz et al.

This study analyses the possibility of a gravitational event occurring as a result of a cascade of rapid events, namely the rupture of a thermokarst lake, the rupture of a rock glacier front, the development of a debris flow and the blockage of a river impacting the Radurschl valley (Ötztal Alps, Tyrol) on 13 August 2019. The analysis of the overflow risk of an artificial lake created during such events is a topical issue with important societal challenges.

This article proposes an interesting methodology based on multi-criteria analysis that is of interest to the scientific community. It is an interesting article well-argued and can be published in your journal

From a formal point of view

The bibliographical references are numerous. These citations support the scientific demonstration but sometimes make the reading more difficult. For example, is it necessary to cite references from 1984 to 2010 (see line 231) or between 1990 and 2010 in line 235. It is not a question of ignoring previous work but of finding the right balance between citations.

Also, there are many examples where authors are quoted twice in two consecutive sentences (line 137 and 138 for Glade and Crozier et al., lines 183 and 185 for Patton et al. to name just two examples). Is this necessary? Why not choose the most striking points of the work without wanting to refer to everything that is said in the article. I think the readability would be improved by reworking the citations. But this is a side remark that does not detract from the quality of the work.

On the content

The article is well constructed, and the argument is well presented.

In the introduction why not mention the study site in the last paragraph?

Line 63 the authors underline the fact that the assessment of the hazard potential of rock glaciers requires an integrated approach combining hydrogeological, meteorological, thermal, geomorphological, and mechanical aspects in a coherent framework. As these aspects are well identified in the introduction, why not go into more detail later? The hydrogeological descriptions are described succinctly from line 104 onwards. What arguments do you have to say that the cirque is drained only by subsurface water? Do you have any idea of the mineralization of the water from your various sources? Is it possible to indicate them on your figure 2 or 3. What arguments do you have for saying that the water flows along distinct channels eroded into the ice core?

Line 105 you refer to a considerable catchment area when the only indication given of the size of the catchment area is 2.8km$^2$. Can you provide additional information to clarify this point?

Meteorology part: Line 88 can you clarify what you mean by "moderate annual precipitation"?

Table 1 the legend of your table shows a chronicle of data from 1976 to 2019 which is based on references from 2016 and 2018, there is a problem in your dates

2019 does seem to be a special year from a meteorological point of view, with more snow and precipitation than the previous 4 years. Since 1976, have there been similar rain and snow conditions as those that caused the disaster, i.e. have you analysed all your weather data by year with the same finesse to see if there were similar weather conditions without necessarily reaching the 2019 disaster?

The particle size analysis was done on 8 samples. Is this enough to cover the whole surface?

Discussion part

The discussion is really interesting and poses the problem in the long term, especially the effect of climate change which accelerates the degradation of permafrost and favours the creation of a thermokarst and the possible consequences.

Finally, the authors point out that debris flows are initiated by the destabilisation of rocky glacier fronts and most often occur in response to heavy rainfall. They also state that intense snowmelt or rain-on-snow events and exceptionally warm periods have also been identified as triggering factors. My question is, does climate change mean less snow and less rain? If so, what is the real impact of climate change on the risk? Can the authors elaborate on how they pro climate change: more snowmelt because more heat with less rain? less snow so a lower snowline and fewer glaciers? What is their scenario?

In conclusion, this article sheds important light on the risks involved weather conditions. The proposed retrospective analysis is interesting and deserves to be published in your journal

---

## Author Comment (AC1)

**Response to Anonymous Referee #1**

We thank Anonymous Referee #1 for his or her time and effort in reviewing our manuscript. We sincerely appreciate the constructive feedback and helpful comments that will improve the quality of the manuscript. Below we have addressed all the points by stating the referee comments (RC) and outlining the corresponding author's response (AR) in *italics*.

**RC:** The manuscript investigates a cascade of processes in the Radurschl Valley in the Austrian Alps. The outburst of a (supra-) glacial lake initiated the event, which continued with the failure of a rock glacier front with subsequent debris flow initiation. The deposit of the latter caused the blocking of the valley and the formation of a lake, which was drained to manage the risk related to its possible outburst. The series of events and their description are very interesting and, in my opinion, would enrich the scientific literature. However, I have some concerns regarding the manuscript, which in my opinion could profit from some revisions.

The methods are described at a level of detail that is not sufficient for understanding the analysis and reproduce the study. In this sense, they need to be revised and improved. The results and the discussion are mixed at several points in the text, with several repetitions and omissions. The logic structure is at times difficult to follow.

As a general suggestion, I would welcome a major revision of the text. Firstly, the authors could more effectively highlight the original analysis and results of the study, avoid substantial repetitions, and increase the logic structure of the manuscript. Secondly, the use of technical language should be always preferred to common-language expressions (e.g. "temporal changes" vs "dynamics", see detailed comments later) in all cases where this is possible and the methods should be described in more detail. Some minor analysis and addition to the data presented might improve the manuscript and its relevance in the community.

In the following, I provide some more specific comments, that I believe could improve the manuscript. I hope that the authors can receive the feedback in the in the same spirit in which it was written and for what it is: constructive suggestion to improve the paper.

*AR: We welcome the constructive feedback provided by Anonymous Referee #1 and thank him or her for regarding our contribution as relevant to the scientific literature. We agree that the manuscript would benefit from revisions based on the comments provided by Anonymous Referee #1. Below we address the outlined issues point by point, including major revisions of the text, the figures and the provided Supplementary Material.*

*Specifically, we will streamline the description of methods, eliminate repetitions and omissions, and highlight the largely open-source based nature of our study. We will provide the corresponding public access options directly in the text, extend the 'Data availability' section, and include data collected by the authors in the Supplementary Material, ensuring full reproducibility of our analysis.*

*In line with the referee's comments, we suggest revising the manuscript in order to highlight the original analysis and results of the study. In this sense, recapitulation of previous literature will be reduced and generic statements underpinning the chosen methods will be replaced by more specific descriptions. Regarding the use of technical language, we will revise the entire manuscript accordingly, aiming at more precise descriptions and avoidance of redundant adjectives. We added additional data and analysis based on the valuable suggestions of Anonymous Referee #1 and Catherine Bertrand (Referee #2), improving the lines of evidence according to the aims of our study.*

**RC: Page 1, Line 1:** Is the front of the rock glacier failing or are the debris accumulated in the gully (below the rock glacier) that are triggered and flow downhill? I write the comment here, but it applies to several other sections in the manuscript. I think that there is a distinction between rock glacier failure, active layer detachment, and debris flow initiation at a rock glacier front. Neglecting the differences between all these processes is in my opinion a first-order approach, which might well be justified in certain contexts, but should not be used in this manuscript, where the focus is on this specific topic."

*AR: We recognize that precise terminology is essential. The paper addresses debris flow initiation at a rock glacier front, as indicated by distinct features of the debris flow initiation zone (figure 4a, 4b). The appearance of cut-off ice lenses (figure 4b) as well as the depth/width ratio (figure 4a) preclude an active layer detachment. The rock glacier is not undergoing a phase of destabilization as entire landform, but local failure in a specific part of the*

*rock glacier front, where a newly formed meltwater channel connects it to a temporary thermokarst lake (please see also RC and AR addressing page 2, line 55; RC and AR addressing page 8, figure 4; RC and AR regarding page 6, line 111 and following; as well as RC and AC addressing page 14, line 308). Observations supporting this specification are presented below (revised text and Supplementary Figure 2; new Supplementary Figures 3, 4, and 5). We will update the entire manuscript accordingly, ensuring consistent and precise language.*

**RC: Page 1, Line 3**: What does "multivariate permafrost degradation" mean? What is "retrogressive debris flow"?

*AR: We suggest replacing "multivariate permafrost degradation" by "permafrost degradation" as well as the term "retrogressive debris flow" by "debris flow", as these are the essential features we would like to highlight in the abstract.*

**RC: Page 1, Line 5:** What do you mean by "environmental forces" and what are the "ambiguous conditions"?

*AR: We suggest replacing the sentence in line 5–7 by "Potentially destabilizing factors were analyzed systematically to deduce the failure mechanism." We acknowledge that the terms "environmental forces" and "ambiguous conditions" lack the desired precision and are not strictly necessary.*

**RC: Page 1, Line 6:** The paper is an application of the methodology presented in Glade and Crozier, 2005. If I understood this correctly, the authors do not establish a basis for multi-hazard assessment but apply an existing method. If contrarily the authors argue that they implemented the methods for hazard assessment, I would welcome a clearer explanation of the original methodology, particularly with regards to the concept of hazard.

*AR: As correctly indicated by Anonymous Referee #1, the paper applies the methodology presented by Glade and Crozier (2005). We agree that our original description was misleading and suggest deleting the phrase "and establish a basis for multi hazard assessment in similar settings".*

**RC: Page 2, Line 55:** I would argue that destabilizing rock glaciers are defined by the combination of abnormal velocities and the presence of surface features such as cracks and scarps. Several recent papers in the literature provide a more accurate description of rock glacier destabilization.

*AR: We follow the referee's argument and suggest replacing the sentence in line 55–57 by "Destabilizing rock glaciers are defined by a sudden acceleration of the landform, exhibiting displacement rates up to several meters per year, and the presence of surface features such as cracks and scarps (e. g. Marcer et al., 2021)."*

*Lit.: Marcer, M.; Cicoira, A.; Cusicanqui, D.; Bodin, X.; Echelard, T.; Obregon, R.; Schoeneich, P. (2021): Rock glaciers throughout the French Alps accelerated and destabilised since 1990 as air temperatures increased. Commun Earth Environ 2 (1), 383. DOI: 10.1038/s43247-021-00150-6.*

**RC: Page 3, Line 63:** reading this line, the expectations to this paper sky-rocketed. Unfortunately, I must say that they were not satisfied by the manuscript, especially with regards to hydrogeological and mechanical aspects (some comments later). With this I don't want to suggest major revisions, but simply to be more precise in the description of the goals of the paper.

*AR: We thank Anonymous Referee #1 for communicating concerns about sky-rocketed expectations. We suggest deleting the sentence in line 62–64.*

*To be more precise in the description of the goals of the paper, we suggest revising the last paragraph of the introduction (line 65–69) as follows: "The aim of this paper is to explore the destabilizing factors leading to local failure of an active rock glacier front in the high-mountain cirque Hüttekar in the Austrian Alps. We analyze the cascading processes involving thermokarst lake outburst, debris flow, and river blockage. We evaluate a set of potentially contributing factors, assess critical combinations, and develop a consistent conception explaining debris flow initiation at the rock glacier front. Similarities and differences with respect to documented debris flows at other rock glacier fronts are analyzed, and conclusions drawn regarding hazard potential."*

**RC: Page 4, Line 96:** This sentence is in my opinion very controversial. Are the authors stating that the rock glacier originated 170 years ago as a result of the LIA glacier advance? The next sentence quickly describes the surface morphology of the terrain influenced by recent glacier advances. Why not considering the simplest and mostsupported hypothesis that the rock glacier pre-existed and was simply overridden by a glacier during the LIA? Anyway, I would simply avoid this argument, which is irrelevant for the discussion of the results and the topic presented in the paper. If the authors want to keep the statement, I would welcome more extensive explanations of their theory.

*AR: We agree and follow the suggestion of Anonymous Referee #1 to avoid this argument. Thus, we suggest deleting the two sentences in line 96–99 and revising the associated paragraph as outlined in our response to the referee comment addressing page 5, line 105 (please see below). Accordingly, we suggest deleting Supplementary Figure 1.*

**RC: Page 5, Line 105:** how do the authors know that the channels are eroded into the ice core? Are the authors referring to water flowing on top of the permafrost table or to water infiltrating into the ice-core? This is a very interesting topic and if observations are available, they could be presented and discussed in more detail. If no more details are available, I would still welcome a clearer description of the observations (if considered relevant).

*AR: We follow the suggestion of extending the description of our observations, including more relevant details and presentation of the available material. Regarding the observed channels along the rock glacier permafrost table, we suggest deleting the term 'eroded' to avoid hypotheses about the generating process at this point. We revised our description to include all available details, given the difficulties in precisely identifying flow paths in active rock glaciers. The updated description is presented below, along with a new figure including a photograph of an observed water current along the permafrost table (please see below). As a preliminary suggestion, the new figure could be included in the Supplementary Material and is provisionally termed „Supplementary Figure 3". While it is possible that some water is infiltrating into the rock glacier ice-core as well, we cannot prove this by direct observations and thus avoid speculations regarding this issue. In the context of the present study, we consider it sufficient to demonstrate the establishment of a thermokarst channel connecting the thermokarst lake to the debris flow initiation zone, irrespective of its exact vertical position. Evidence for this connection is described in our response to the referee comment addressing page 7, line 123 (please see below) and documented through photographs in Fig. 4 of the original manuscript, the revised Supplementary Figure 2 presented below, as well as the new Supplementary Figure 3 presented below. Accordingly, we suggest revising the last paragraph of section 2 'Study site' by replacing line 96–107 by:*

*"The largest rock glacier in Hüttekar-cirque is 1408 m long and up to 493 m wide, covering an area of 0.5 km² in the lowermost parts of the cirque. Its gently sloping surface is characterized by distinct furrows and ridges, while the steep front rests on top of a slope above Radurschl Valley. Rock glacier debris is composed of orthogneiss derived from the Glockturm massif. In the southeast, the lower, debris covered parts of Glockturmferner transition into the rock glacier rooting zone. The exact boundary between debris covered glacier and rock glacier is not known. Massive ice is frequently visible beneath a ~1–2 m thick debris layer in a shallow depression at the southeastern edge of the rock glacier, close to the suspected transition zone. The debris layer consists of poorly sorted boulders with individual blocks measuring up to 4 m, arranged in a loose, clast-supported structure. The blocky surface layer covers a finer-grained layer dominated by well-graded gravel and sand that is exposed along the rock glacier front. The unfrozen domain of this heterogeneous debris layer increases irregularly in thickness towards the rock glacier front, reaching a maximal thickness of ~5–10 m.*

*A small meltwater current from Glockturmferner infiltrates into the rock glacier rooting zone (Fig. 3). Water flowing along the permafrost table is visible and audible between boulders at several places in the southern part of the rock glacier. These water currents follow distinct channels that can be traced below the boulders along the rock glacier surface. Where visible between the boulders, the channels are up to 1 m wide and 20 cm deep (Supplementary Figure 3). Surface water bodies in Hüttekar-cirque are restricted to small meltwater lakes and currents in immediate vicinity of the two glaciers. The absence of surface creeks in the lower parts of Hüttekar-cirque indicates that it is drained exclusively by subsurface flow, emerging as a group of small springs at the toe of the slope descending to Radurschl Valley (Fig. 3, Supplementary Figure 3). The exact number and position of these springs varies during the year but is always constrained to the sedimentary cone covering the toe of the slope, as evident from 23 field surveys between 2019 and 2022 covering all months. Spring discharge of individual outlets is < 1 l/s and electrical conductivity ranges from 60 to 75 µS/cm, as indicated by repeated measurements during these surveys."*

*To visualize these observations and allow for direct assessment of these features by the interested reader, we suggest including a new figure in the Supplementary Material ("Supplementary Figure 3") showing photographs of key observations:*

[Figure]

**Supplementary Figure 3.**

*In addition, we updated Fig. 3 to include the position of the springs and highlighted the observed meltwater channel:*

[Figure]

**Figure 3.** *Geological-geomorphological map, compiled using the most recent geological map provided by the Geological Survey of Austria (Moser, 2012). It is complemented by ortho-images and a high-resolution digital terrain model (DTM) derived from airborne laser scanning data (Government of the Province of Tyrol, 2021a). The map is based on comprehensive field mapping (2019–2021).*

**RC: Page 6, Line 106:** what is a "retrogressive debris flow"? I see here some confusion between "retrogressive erosion" and "debris flow".

*AR: We suggest replacing the term by "debris flow".*

**RC: Page 6, Line 111 and following:** This paragraph is extremely interesting, and I suggest to extend the description (here) and discussion (later) of the observations, the numbers presented, the methodology used to derive them, and their uncertainties. These points might be extended in several locations in the manuscript, maybe replacing some of the repetitions and digressions that are currently present.

*AR: We thank Anonymous Referee #1 for expressing his or her interest and suggestions to extend the descriptions. Please note that the presented numbers merely serve as an indication of the scales involved and are not used for any calculations. We revised the paragraph (line 109–122) accordingly, focusing on the issues addressed by Anonymous Referee #1:*

*„A debris flow erodes the steep slope bordering Hüttekar-cirque to the west (Fig. 2; classification according to Hungr et al., 2014). Following a moderate precipitation event, destabilization initiated on 13 August 2019, mobilizing a volume of several thousand m³ from the steep rock glacier front (Fig. 4a). The debris flow started at 03:00 AM (Central European Summer Time, UTC+2), the main event lasted until about 12:00 PM, followed by reduced debris flow activity that persisted until the next day. The debris flow was observed by Josef Waldner (staff of the nearby hut Hohenzollernhaus) and Gerhard Schaffenrath (local shepherd). On 14 and 26 August 2019, respectively, the debris flow was documented during helicopter flights by Roman Außerlechner, Thomas Figl and Werner Thöny (Geological Survey of Tyrol). Slope failure initiated along an irregularly shaped rupture in ice-*

*cemented debris, exposed at the main scarp (Fig. 4b). Accelerating and disintegrating, the transported mass evolved into a debris flow following a narrow channel down the steep slope below the rock glacier front (Fig. 2, Fig. 3; Supplementary Figures 3 and 4, respectively). About 200 m below the initiation zone, the debris flow spread out and formed a fan of 33,000 $m^2$, thereby damming the river Radurschlbach at an elevation of 2,200 m a. s. l. (Fig. 4c, Fig. 4d; Supplementary Figures 3 and 4, respectively). Based on in situ observations, aerial photographs, and several surveys conducted before as well as after the debris flow event, a total volume of 40,000 – 50,000 $m^3$ of mobilized sediment is estimated, providing a rough indication of the event magnitude. Consequently, a lake covering an area of ~60,000 $m^2$ developed in Radurschl Valley, causing the downstream riverbed to fall dry temporarily (Fig. 4c; Supplementary Figures 3 and 4, respectively). Excavation of a drainage channel lowered the mean water depth from 2 m to 1 m during the following days to prevent a potentially catastrophic outburst. The corresponding water depths are reconstructed from in situ observations and reviewed by mapping the maximum lake extent as well as the lake extent following excavation, and corresponding volume calculations employing a high-resolution digital terrain model described in section 4.1. Subsequently, a dam was constructed on the debris fan to restrain future debris flows from damming Radurschlbach again."*

*We suggest supporting this description with a new figure, showing the entire debris flow including initiation zone, transportation zone, and deposition zone, as well as details including frozen material, levees, and river blockage. As a preliminary suggestion, the new figure is provisionally termed „Supplementary Figure 4" and presented below:*

[Figure]

***Supplementary Figure 4***

**RC: Page 7, Line 123:** Concurrently means at the same time. The question of the timing is essential for the process described. Please explain in more detail what is known (and how, see previous comment) and what are your hypothesis. Later in the manuscript, they should be tested and discussed.

*AR: To clearly distinguish observations from hypotheses, we suggest revising the entire paragraph (line 123–134), addressing the issues outlined by Anonymous Referee #1 as precisely as possible. Please note that due to the remoteness of the study site and the absence of instrumentation at the time of the event, the timing of individual processes can only be roughly reconstructed (temporal resolution on the order of several hours). In accordance*

*with these constraints, our study presents rough estimates of the involved time scales (thermokarst lake development ~2 months, thermokarst lake drainage ~1 day, debris flow event ~1.5 days) that can reliably be deduced. Throughout the manuscript, we will highlight the fact that the discussed implications and drawn conclusions are consistently reconciled with the accuracy of these estimates. The revised event description (line 123–134) addresses the issues outlined by Anonymous Referee #1:*

*"During the same night as the debris flow initiation, a thermokarst lake on top of Hüttekar Rock Glacier, ~350 m behind the debris flow initiation zone, started draining and emptied almost completely during the following day, as indicated by local observations (Fig. 3, Fig. 4e, Fig. 4f). The thermokarst lake had started to develop coincidently with the onset of snowmelt in early June 2019 within a shallow depression where massive ice within the rock glacier prevented drainage (Fig. 4g). During the last decades, a comparable feature had never been observed before in Hüttekar-cirque, despite frequent visits by hikers, hunters, shepherds, and staff of Hohenzollernhaus (Josef Waldner, pers. comm.). In line with these observations, publicly available remote sensing data and historical maps of Hüttekar do not exhibit any indications of thermokarst lake development on Hüttekar Rock Glacier before June 2019 (evident from Sentinel-2 satellite imagery provided by Copernicus and processed by Sentinel Hub (https://www.sentinel-hub.com/), historical ortho-images and laserscans provided by the Government of the Province of Tyrol, (https://www.data.gv.at/; https://lba.tirol.gv.at/), historical maps provided by the Government of the Province of Tyrol (https://hik.tirol.gv.at)). In the stage of its largest extent, the thermokarst lake was approximately 300 m long, up to about 150 m wide and 4–5 m deep, comprising an estimated water volume of ~150,000 m$^3$.*

*Effective drainage occurred through a large crevasse (width ~1.5 m, height ~2 m) that formed in association with the debris flow initiation (Fig. 4h). The crevasse is part of a newly formed channel system connecting the thermokarst lake to that part of the rock glacier front that failed, constituting the debris flow initiation zone. While the exact vertical position of this channel system is not known, clearly discernible and precisely confined collapse structures within the debris layer indicate the trace of this channel network along the rock glacier surface (Fig. 4e, Fig. 4f; Supplementary Figures 2 and 3, respectively). Large amounts of water were rapidly transferred to the debris flow initiation zone and torrent beneath. Retrogressive linear erosion visible in the initiation zone indicates that concentrated water flow emerged at the main scarp (Fig. 4b), in good agreement with earlier observations of rock glacier front failures (Kummert et al., 2018; Marcer et al., 2020; Kofler et al., 2021). Erosion of unfrozen sediment in the vicinity of the debris flow initiation zone persistently modifies its shape and continues to widen it until today. Since then, the depression storing the former thermokarst lake never filled again but still shows distinct morphology (Supplementary Figures 2 and 3)."*

**RC: Page 7, Line 126:** it is interesting that local knowledge is taken into account. Still, I suggest to perform a systematic analysis with modern remote sensing data (Sentinel 2 for example) to quantify the evolution of the lake in recent times.

*AR: We agree with this valuable suggestion and performed a systematic analysis with modern remote sensing data, carefully scanning available archives for indications of thermokarst lakes appearing on Hüttekar Rock Glacier before June 2019. In addition, we included historical and modern maps in our analysis. We suggest communicating our results as outlined in the completely revised section 3 (please see our response to the referee comment addressing page 7, line 123).*

**RC: Page 7, Line 139:** Are the authors talking about dynamic forces here? Impact from external mass movements? Earthquakes?

*AR: We acknowledge that the term "dynamic forces" as used in our original manuscript is misleading. Throughout the manuscript, we suggest replacing the term "dynamic forces" by "dynamic factors", strictly following the nomenclature suggested by Glade and Crozier (2005). In this way, we highlight the consistent application of the factor mapping methodology for landslide susceptibility assessment provided by these authors, in accordance with our responses to the referee comments addressing page 1, line 6 (please see above) and page 9, section 4.1 (please see below), respectively.*

**RC: Page 8, from Figure 4:** Very interesting observations of the rock glacier front. What's the height of the shear horizon depth here? I would be very happy to see in the published version of the manuscript a detailed

description of this area (possibly before and after the event?), which is highly relevant to the debris flow initiation. A figure showing a simple comparison before-after the event (maybe in appendix?) would be very useful. What is the thickness of the debris around the lake? And around the front? Panel g and h seem to indicate a debris covered glacier. Would It be possible to provide a more detailed geomorphological assessment of the landform and a map of it?

*AR: We thank Anonymous Referee #1 for appreciating the observations of the rock glacier front and his or her suggestions to present them in more detail. To communicate these observations as objectively as possible, we followed these suggestions by preparing a new figure that shows a comparison before-after the event. The new figure could be included in the appendix or in the Supplementary Material, where further photographs are already available. As a preliminary suggestion, the new figure is provisionally termed „Supplementary Figure 5" and presented below:*

[Figure]

***Supplementary Figure 5.***

*In addition, we extended Supplementary Figure 2 showing ortho-images of Hüttekar Rock Glacier before the 2019 debris flow event (1970, 2003, 2007, 2010, and 2015) as well as after the event (2020):*

[Figure]

Government of the Province of Tyrol: Current and historical orthoimagery of Tyrol, Land Tirol - data.tirol.gv.at, CC BY 4.0, 2021.

*The depth of the shear horizon is highly variable, as the debris flow initiation zone is characterized by an irregularly shaped rupture in ice-cemented debris, adjusting its shape and widening by erosion of unfrozen debris in its vicinity. These issues are now highlighted in the revised description of the debris flow event in section 3 (please see above). The unfrozen debris around the front is up to 5-10 m thick. The thickness of the debris around the lake is ~1-2 m (now included in the revised description of the study site in section 2, please see above). As the glacier (Glockturmferner) transitions into the rock glacier (Hüttekar Rock Glacier), a clear boundary between the two landforms cannot be drawn. We summarized what is known as precisely and objectively as possible in the revised description of the study site (please see above).*

*Accordingly, we suggest complementing line 271 and the following with a detailed description of the debris flow initiation zone before and after the debris flow event:*

*"Prior to landslide initiation, the prospective initiation zone was characterized by steep slope angle and convex downslope curvature (Table 2). It formed a part of the 400 m wide and 100 m high rock glacier front, bounded to the top by a distinct erosional edge separating it from the flat rock glacier surface. The steep rock glacier front (35° on average) exhibited roughly homogeneous appearance, except for two bedrock outcrops in its southern part (Supplementary Figures 3 and 5, respectively). Since the debris flow initiated on 13 August 2019, the 80 m*

*wide and 200 m long debris flow initiation zone eroded into the rock glacier front exposes its internal structure. It forms an irregular, concave niche characterized by steep flanks that are up to 30 m high and composed of loose, poorly-sorted sediment (Fig. 4a, b; Supplementary Figures 2–5). Its top is connected to collapse structures connecting it to the former position of the thermokarst lake as described in Section 3, with linear erosion features indicating emergence of concentrated water. After the debris flow initiation, frozen material was exposed immediately below the top (Fig. 4a,b; Supplementary Figures 3 and 4, respectively)."*

*We hope that the revised geomorphological description given above, the revised Fig. 3 shown above, the revised description of the area before and after the event, the revised Supplementary Figure 2 shown above, and the new Supplementary Figures 3, 4, and 5 presented above clarify the issues outlined by Anonymous Referee #1.*

**RC: Page 9, Section 4.1:** some of the text presented here might be more suited for the introduction. Also, the analysis of the presented factors on slope stability is not clear to me. I don't see any stability assessment, or somehow I missed it. Maybe a clearer description of the methods might help.

*AR: As correctly stated by Anonymous Referee #1, our study does not provide a full stability assessment of the slope, but rather an analysis of landslide susceptibility employing factor mapping as outlined by Crozier and Glade (2005). We suggest revising Section 4.1 (line 145–172), removing general statements that are misplaced in this section and providing a more precise method description. The revised text will also highlight the public availability of the analyzed data sets by specifying corresponding weblinks and providing the grain size analysis data in the Supplementary Material.*

**RC: Page 9, Line 165 and following:** This is in my opinion an interesting approach, but cannot be introduced so lightly. Please provide more details, assumptions, rationale, and potential limitations. As known, the internal structure of rock glacier is very different from the active layer (also not homogeneous in the active layer!). Assuming the the latter is representative of the first is not valid in my opinion, especially given the photographs presented in figure 4. Were the boulders only collected at the surface? Only evaluating the boulders at the surface creates a bias towards boulders. How were the samples collected at the front? Who went there? Where exactly were they collected? How representative should this be of the internal part of the rock glacier, especially the shear horizon if you suggest that water was flowing there (erosion/deposition at the surface)?

*AR: We agree with Anonymous Referee #1 that the internal structure of rock glaciers is highly heterogeneous and that the differentiation between active layer, rock glacier core, and rock glacier front is essential. To clarify this point and to restrict the analysis to the most essential features of the studied rock glacier, we suggest cancelling the discussion of the coarse-grained surface layer and focus on the composition of the rock glacier front, which is where the debris flow initiated. We revised Fig. 3 (please see above) accordingly, and restricted Fig. 6 to the grain size distribution of the rock glacier front (please see below). We suggest deleting line 284–290 and revising 165–172, addressing the issues outlined by Anonymous Referee #1:*

*"The structural characteristics of the material involved in the debris flow are inspected by analyzing the grain size distribution of the rock glacier front. In August 2009, four samples were taken 50 m southwest of the later debris flow initiation zone and analyzed by wet sieving (position indicated in Fig. 3). Due to the heterogeneous structure of the rock glacier, these samples are not representative for the grain size composition of the entire rock glacier. In contrast, the material at the rock glacier front appears comparatively homogenous. Considering the proximity of the sampling locations to the subsequent debris flow initiation zone, and the dangerous sampling conditions at active rock glacier fronts in general, we consider the samples reasonable approximations to the grain size composition of the material that was mobilized during the debris flow event on 13 August 2019."*

*In addition, we suggest including the grain size analysis data in the 'Supplementary Material'. The revised Fig. 6 is presented below:*

[Figure]

***Figure 6.*** *Grain-size distributions of material composing the rock glacier front. The corresponding sampling locations are 50 m southwest of the debris flow initiation zone, as indicated in Fig. 3. Dashed grey lines represent source compositions of a set of experimentally investigated debris flows showing contractive shear response and undrained failure (USGS debris flow flume; Major (1996); Iverson (1997); Iverson et al. (1997)). Classification according to ISO 14688-1 (International Organization for Standardization, 2017).*

**RC: Page 9, line 177:** this is not a sufficient description for the data used.

*AR: We suggest complementing the description by expanding line 178–182 as follows: "In the observation period 1970–2020, a simple but effective approach was used to assess the surface displacement rates within single periods (1970-2003-2007-2010-2015-2020). In doing so, 200 prominent blocks, geometrically well distributed on the rock glacier surface, were visually identified at each ortho-image epoch to approximate the horizontal surface displacement rates at the respective location. Visual detectability within optical imagery is approximately half of a pixel size (in this case 10 cm). In this study assessed surface displacement rates cover at least 3 years, so with a given spatial resolution of 20 cm of the ortho-images, minimum rates of 3–4 cm are theoretically detectable on Hüttekar Rock Glacier. Additionally, the position of the rock glacier front line (top of the erosional slope) is mapped at every epoch. Both analyses provide the basis for assessing the kinetic patterns on the rock glacier surface (Avian et al., 2009; Kummert and Delaloye, 2018)."*

**RC: Page 10, line 183 and following:** is the use of climate relevant here? Are the authors not limiting the analysis to meteorological forcing? Also, what is meant by "progressive change of climatic factors"? In general, the description of the method is not sufficient for reproduction, and in this sense, it should be improved and extended.

*AR: We will address these issues carefully by revising and extending the method description to ensure reproducibility and demonstrate the relevance of climate in the context of the study. We consider the long-term influence of climate relevant for two reasons: (1) The limiting factor governing the rate and extent of thermokarst evolution at Hüttekar Rock Glacier is the available energy input, which is outstandingly high in summer 2019 compared to earlier years (as indicated in Table 3; please see also line 188–190 on page 10 and line 315–316 on page 16). In this sense, the analysis of climate allows putting the conditions during summer 2019 in a long-term context. (2) The strong increase in air temperature and positive degree day sum, outpacing corresponding trends at the global and European Alps scale (please see line 317–320 on page 16), suggests that permafrost degradation and upward shifting of the lower permafrost boundary affected Hüttekar-cirque strongly during the years preceding 2019. We agree with Catherine Bertrand (Referee #2), who outlined that the discussion of our study results "poses the problem in the long term, especially the effect of climate change which accelerates the degradation of permafrost and favors the creation of a thermokarst and the possible consequences." (please see comment on egusphere 2022-567 by Catherine Bertrand, https://doi.org/10.5194/egusphere-2022-567-RC2).*

*Accordingly, we believe that the results are relevant in the context of our study and suggest revising the manuscript to explain and highlight this relevance.*

*We recognize that the open-source based nature of the entire study was not adequately highlighted in the original manuscript. The dataset analyzed here is almost completely publicly available, implying that the analysis is reproducible. Currently, the only exception is the employed snow model (SNOWGRID). However, full access to this dataset will be provided in the future via the Central Institute for Meteorology and Geodynamics (Austria). Similarly, the methods of analysis rely on open-source software that is publicly available. To address the issues outlined by Anonymous Referee #1, we suggest revising the main text of the manuscript accordingly, highlighting public access options to the analyzed data, extending the 'Data availability' section, and including processed data in the Supplementary Material.*

**RC: Page 10, Line 210:** are the calculations about snow cover validated in any way?

*AR: We recognize that the original manuscript is lacking information regarding the validation of the snow model. The analyzed dataset is a subset of a nationwide dataset that is continuously updated through the distributed SNOWGRID model (Olefs et al., 2013). The resulting snow cover data are routinely evaluated and validated by the Central Institute for Meteorology and Geodynamics using more than 50 station measurements of snow depth, additional measurements regarding snow depth and snow water equivalent (5 stations along with more than 200 individual measurements), snow depth measurements employing laser sensors, winter mass balance measurements of glaciers within the model domain, spatial validation of snow cover extent using satellite-based fractional snow cover area provided by MODIS, as well as cumulative runoff data (Olefs et al., 2020). Additional verification stems from feedback provided by a large range of SNOWGRID applications including research projects as well as commercial applications (for an overview, please see https://www.zamg.ac.at/cms/de/forschung/ klima/klimatografien/snowgrid). We suggest revising and extending the snow model description in line 207–214 of the manuscript, including an explanation of the model validation procedure.*

*Lit.: Olefs M., Koch R., Schöner W. and Marke T. (2020): Changes in Snow Depth, Snow Cover Duration, and Potential Snowmaking Conditions in Austria, 1961–2020—A Model Based Approach. Atmosphere 11(12):1330. DOI: 10.3390/atmos11121330.*

**RC: Page 11, Line 218:** The statements about the hydrology seems to be results. What observations are available? Please expand the description of the data and methods here. E.g. why until the 13th of August?

*AR: We agree with Anonymous Referee #1 that these observations are misplaced here. As outlined in our response to the referee comment addressing page 5, line 105, we suggest reporting the role of glacial meltwater in the last paragraph of section 2 (Study site), summarizing the hydrology and hydrogeology of Hüttekar (please see above). The employed methods aim at maximizing comparability with a similar study performed by Kofler et al. (2021), who analyzed the initiation of debris flows from two rock glacier fronts in nearby mountain ranges. To estimate the total water input received by Hüttekar-cirque in 2019 prior to the debris flow initiation, we calculated the cumulative rainfall, snow melt and ice melt from 1 January 2019 to 13 August 2019 (failure date) and compared it to the respective water volumes in previous years (consistently totalized from 1 January to 13 August). We will revise the description of data and methods accordingly, addressing the issues outlined by Anonymous Referee #1.*

**RC: Page 14, line 308:** Data of surface velocities would be very interesting. Is it possible to present them? Especially relevant if the authors speak about destabilization. Possibly you could also see some signal in ice melt (vertical component – once subtracted subsidence due to dynamics).

*AR: We thank Anonymous Referee #1 for this valuable suggestion and will prepare a new figure illustrating our results. In addition, we suggest providing the feature tracking data showing the rock glacier surface kinematics as Supplementary Material. Please note, however, that the study addresses local failure in a specific part of the rock glacier front, not destabilization of the entire landform. We believe that several of the comments answered above will help clarifying this important distinction in a revised version of the manuscript. Therefore, in our opinion calculating the subsidence due to rock glacier dynamics and extracting some signal in ice melt is beyond the scope of the current study.*

**RC: Page 17, Figure 8:** great figure. I suggest to add the equations to the lines and expand the discussion with regards to failure processes (Lou and Similaun different from this case, no?)

*AR: We thank Anonymous Referee #1 for appreciating the figure and for providing valuable suggestions that will improve the figure as well as the main text of the manuscript. We agree with these suggestions and prepared an updated figure including the equations (please see below). As correctly noted by Anonymous Referee #1, the documented debris flows at Murfreit Rock Glacier, Lou Rock Glacier, Similaungrube Rock Glacier, Hintergrat Rock Glacier, and Ritigraben Rock Glacier occurred in response to heavy rainfall, in contrast to the debris flow at Hüttekar Rock Glacier. To highlight this key difference, we suggest modifying line 336–340 of the manuscript as follows:*

*"These characteristics distinctly contrast with the rainfall events preceding well documented debris flows from rock glacier fronts in the European Alps. Those occurred most often in response to heavy rainfall (Fig. 8): the Ritigraben event on 24 September 1993 (Lugon and Stoffel, 2010), the Murfreit event on 02 July 2003 (Krainer et al., 2012), the Hintergrat and Similaungrube events, both on 13 August 2014 (Kofler et al., 2021), as well as the Lou event on 14 August 2015 (Marcer et al., 2020). In contrast, the rainfall event preceding the debris flow at Hüttekar Rock Glacier was neither especially intense, nor especially persistent (Fig. 8)."*

[Figure]

*Figure 8. Rainfall event frequency analysis with respect to (a) rainfall event intensity I vs. rainfall event duration D and (b) rainfall event volume E vs. rainfall event duration D based on INCA (note logarithmic scale). Regional thresholds describing minimum conditions for rainfall-induced debris flows are based on Nikolopoulos et al. (2015b) and Marra et al. (2016), with equations given below the corresponding lines (thresholds are representative for the summer season). Characteristics of triggering rainfall events are based on Lugon and Stoffel (2010), Krainer et al. (2012), Kofler et al. (2021), and Marcer et al. (2020).*

**RC: Discussion and Conclusions:** I would welcome a more detailed discussion of the original results of the manuscript, limiting long review of previous literature. Same for the conclusions. Maybe some of the comments above can help in the process.

*AR: We thank Anonymous Referee #1 for his or her valuable comments and suggestions and agree that the comments above will help in the process of revising the 'Discussion' and 'Conclusion' section, respectively. Specifically, we will restrict the review of previous literature and instead focus on the original results of the manuscript, including (1) the unfavorable predisposition of the front of Hüttekar Rock Glacier, (2) the rapid development and sudden drainage of the thermokarst lake on Hüttekar Rock Glacier, driven by extraordinarily high atmospheric energy input during summer 2019 following a long-lasting general warming trend promoting permafrost degradation, (3) the quantification of the energy available for thermokarst evolution at Hüttekar Rock Glacier, critically linked to the development of the thermokarst lake and channel considering the considerable*

*distance (about 350 m) between thermokarst lake and rock glacier front, and (4) implications for future considerations regarding debris flow initiation in permafrost-affected terrain.*

---

## Author Comment (AC2)

**Response to Catherine Bertrand**

We thank Catherine Bertrand for her time and effort in reviewing our manuscript. We sincerely appreciate the constructive feedback and helpful comments that will improve the quality of the manuscript. Below we have addressed all the points by stating the referee comments (RC) and outlining the corresponding author's response (AR) in *italics*.

**RC:** This study analyses the possibility of a gravitational event occurring as a result of a cascade of rapid events, namely the rupture of a thermokarst lake, the rupture of a rock glacier front, the development of a debris flow and the blockage of a river impacting the Radurschl valley (Ötztal Alps, Tyrol) on 13 August 2019. The analysis of the overflow risk of an artificial lake created during such events is a topical issue with important societal challenges.

This article proposes an interesting methodology based on multi-criteria analysis that is of interest to the scientific community. It is an interesting article well-argued and can be published in your journal.

The bibliographical references are numerous. These citations support the scientific demonstration but sometimes make the reading more difficult. For example, is it necessary to cite references from 1984 to 2010 (see line 231) or between 1990 and 2010 in line 235. It is not a question of ignoring previous work but of finding the right balance between citations.

Also, there are many examples where authors are quoted twice in two consecutive sentences (line 137 and 138 for Glade and Crozier et al., lines 183 and 185 for Patton et al. to name just two examples). Is this necessary? Why not choose the most striking points of the work without wanting to refer to everything that is said in the article. I think the readability would be improved by reworking the citations. But this is a side remark that does not detract from the quality of the work.

The article is well constructed, and the argument is well presented.

*AR: We welcome the constructive feedback provided by Catherine Bertrand and thank her for regarding our contribution as relevant to the scientific community, for highlighting the associated societal challenges, and for appreciating the article construction and our presented line of argument. We recognize that the numerous bibliographical references make our submitted manuscript difficult to read and will follow her recommendations regarding the right balance between citations. Accordingly, we suggest revising the entire manuscript, reworking the citations focusing on the most striking points, avoiding quotes in consecutive sentences as well as excessive quotations, and restricting recapitulation of previous literature to highlight the original analysis and results of our study.*

**RC:** In the introduction why not mention the study site in the last paragraph?

*AR: We agree with Catherine Bertrand and suggest revising the last paragraph of the introduction (line 65–69) to describe the goals of the paper more precisely as follows:*

*"The aim of this paper is to explore the destabilizing factors leading to local failure of an active rock glacier front in the high-mountain cirque Hüttekar in the Austrian Alps. We analyze the cascading processes involving thermokarst lake outburst, debris flow, and river blockage. We evaluate a set of potentially contributing factors, assess critical combinations, and develop a consistent conception explaining debris flow initiation at the rock glacier front. Similarities and differences with respect to documented debris flows at other rock glacier fronts are analyzed, and conclusions drawn regarding hazard potential."*

**RC:** Line 63 the authors underline the fact that the assessment of the hazard potential of rock glaciers requires an integrated approach combining hydrogeological, meteorological, thermal, geomorphological, and mechanical aspects in a coherent framework. As these aspects are well identified in the introduction, why not go into more detail later?

*AR: We agree with Catherine Bertrand and suggest deleting the sentence in line 62–64. The applied methodology will be described in more detail in Section 4 (Methods) that will be revised accordingly.*

**RC:** The hydrogeological descriptions are described succinctly from line 104 onwards. What arguments do you have to say that the cirque is drained only by subsurface water? Do you have any idea of the mineralization of the water from your various sources? Is it possible to indicate them on your figure 2 or 3. What arguments do you have for saying that the water flows along distinct channels eroded into the ice core?

*AR: We recognize that the site description provided in the original manuscript should be extended, including more relevant details and presentation of the available material. The slope downstream from Hüttekar-cirque, representing its orographic catchment outlet and connecting it to Radurschl Valley, does not show any indications of surface water. A group of springs emerging at the toe of the slope is the only observable outflow from Hüttekar-cirque. We mapped these springs and measured the electrical conductivity of spring water during several field surveys between 2019 and 2022. We suggest revising the original description as outlined below, including the results of these measurements. We followed the suggestion of Catherine Bertrand and included the position of these springs in the geological-geomorphological map (revised Fig. 3 presented below). In addition, we present a new figure below that could be included in the Supplementary Material and is provisionally termed „Supplementary Figure 3". The photographs document the absence of surface water bodies in Hüttekar-cirque as well as along the downstream slope connecting it to Radurschl Valley. Some of the springs at the toe of the slope are visible in two of the photographs and are outlined accordingly.*

*Regarding the observed channels along the rock glacier permafrost table, we suggest deleting the term 'eroded' to avoid hypotheses about the generating process at this point. We revised our description to include all available details, given the difficulties in precisely identifying flow paths in active rock glaciers. The updated description is presented below, along with a photograph of an observed water current following one of these channels along the permafrost table that is included in Supplementary Figure 3.*

*Accordingly, we suggest revising the last paragraph of section 2 'Study site' by replacing line 96–107 as follows:*

*"The largest rock glacier in Hüttekar-cirque is 1408 m long and up to 493 m wide, covering an area of 0.5 km² in the lowermost parts of the cirque. Its gently sloping surface is characterized by distinct furrows and ridges, while the steep front rests on top of a slope above Radurschl Valley. Rock glacier debris is composed of orthogneiss derived from the Glockturm massif. In the southeast, the lower, debris covered parts of Glockturmferner transition into the rock glacier rooting zone. The exact boundary between debris covered glacier and rock glacier is not known. Massive ice is frequently visible beneath a ~1–2 m thick debris layer in a shallow depression at the southeastern edge of the rock glacier, close to the suspected transition zone. The debris layer consists of poorly sorted boulders with individual blocks measuring up to 4 m, arranged in a loose, clast-supported structure. The blocky surface layer covers a finer-grained layer dominated by well-graded gravel and sand that is exposed along the rock glacier front. The unfrozen domain of this heterogeneous debris layer increases irregularly in thickness towards the rock glacier front, reaching a maximal thickness of ~5–10 m.*

*A small meltwater current from Glockturmferner infiltrates into the rock glacier rooting zone (Fig. 3). Water flowing along the permafrost table is visible and audible between boulders at several places in the southern part of the rock glacier. These water currents follow distinct channels that can be traced below the boulders along the rock glacier surface. Where visible between the boulders, the channels are up to 1 m wide and 20 cm deep (Supplementary Figure 3). Surface water bodies in Hüttekar-cirque are restricted to small meltwater lakes and currents in immediate vicinity of the two glaciers. The absence of surface creeks in the lower parts of Hüttekar-cirque indicates that it is drained exclusively by subsurface flow, emerging as a group of small springs at the toe of the slope descending to Radurschl Valley (Fig. 3, Supplementary Figure 3). The exact number and position of these springs varies during the year but is always constrained to the sedimentary cone covering the toe of the slope, as evident from 23 field surveys between 2019 and 2022 covering all months. Spring discharge of individual outlets is < 1 l/s and electrical conductivity ranges from 60 to 75 µS/cm, as indicated by repeated measurements during these surveys."*

*To visualize these observations and allow for direct assessment of these features by the interested reader, we suggest including a new figure in the Supplementary Material ("Supplementary Figure 3") showing photographs of key observations:*

[Figure]

**Supplementary Figure 3.**

*In addition, we updated Fig. 3 to include the position of the springs and highlighted the observed meltwater channel:*

[Figure]

***Figure 3.*** *Geological-geomorphological map, compiled using the most recent geological map provided by the Geological Survey of Austria (Moser, 2012). It is complemented by ortho-images and a high-resolution digital terrain model (DTM) derived from airborne laser scanning data (Government of the Province of Tyrol, 2021a). The map is based on comprehensive field mapping (2019–2021).*

**RC:** Line 105 you refer to a considerable catchment area when the only indication given of the size of the catchment area is 2.8km$^2$. Can you provide additional information to clarify this point?

**AR:** *We suggest revising the entire paragraph (line 96–107) as outlined above (please see response to the referee comment addressing line 104 of the manuscript). With the revised paragraph, we now suggest avoiding the term "considerable" as it could be misleading here. While Hüttekar-cirque is the most prominent cirque in Upper Radurschl Valley, we acknowledge that it cannot be regarded "considerable" in a general sense.*

**RC:** Line 88 can you clarify what you mean by "moderate annual precipitation"?

**AR:** *We suggest clarifying this issue by replacing the sentence in line 88–89 as follows:*

*"With respect to long-term averages across Austria, annual precipitation is moderate (1042 mm compared to 1077 mm areal average across Austria) and mean annual air temperature is low (-2.5°C compared to 6.6°C areal average across Austria), reflecting the high altitude of Hüttekar-cirque and its central position close to the main chain of the Alps."*

**RC:** Table 1 the legend of your table shows a chronicle of data from 1976 to 2019 which is based on references from 2016 and 2018, there is a problem in your dates.

**AR:** *We acknowledge that our original table caption was misleading and suggest revising it as follows:*

*"Long-term (1976–2019) mean monthly air temperature and precipitation in Hüttekar-cirque. Numbers are calculated based on the continuously updated SPARTACUS dataset (Spatiotemporal Reanalysis Dataset for Climate in Austria), as described in Hiebl and Frei (2016, 2018)".*

**RC:** 2019 does seem to be a special year from a meteorological point of view, with more snow and precipitation than the previous 4 years. Since 1976, have there been similar rain and snow conditions as those that caused the disaster, i.e. have you analysed all your weather data by year with the same finesse to see if there were similar weather conditions without necessarily reaching the 2019 disaster?

*AR: We agree with Catherine Bertrand that 2019 was a special year from a meteorological point of view. Specifically, air temperature, available melting energy as well as the timing and rate of snowmelt distinguished it from the previous years, suggesting that these are the most promising weather factors for further analysis. In contrast, rainfall and ice melt volume were relatively low in 2019 compared to the previous years.*

*The winter season preceding the debris flow was characterized by a prominent snow cover, almost monotonically gaining volume until 29 May 2019. Accordingly, snowmelt was restricted to June, starting at a high SWE stored in the snowpack (Fig. 7). The late onset of snowmelt is responsible for the unusually high peak on 29 May 2019 (while during the preceding years, phases of ablation and accumulation throughout spring precluded a comparable peak). Due to the exceptionally high temperatures in June, the snowmelt process proceeded under a high-energy environment, leading to very high rates of meltwater production that caused widespread flooding in Tyrol (despite the fact that June 2019 was a particularly dry month).*

*In contrast to the high intensity of the snowmelt process, comparing total volumes of snowmelt, ice melt and rainfall between 1 January 2019 and 13 August 2019 (failure date) to earlier years (consistently totalized from 1 January to 13 August) indicates inconspicuous overall conditions in 2019 (Fig. 7c), acknowledging that the relatively short record precludes statistically substantiated conclusions. Compared to the four years before 2019 (for which continuous SNOWGRID data are available), the total snow melt volume exceeded the 5-year-average by 19 %, while the total ice melt volume and rainfall volume were slightly below this average (-3 % and -2 %, respectively). The rapid snowmelt efficiently eliminated the snow cover in Hüttekar, so that snowmelt ceased on 22 July 2019, i.e. 22 days before the debris flow initiated. Accordingly, we infer that there is no indication for particularly high antecedent moisture conditions at the rock glacier front at the timing of the debris flow, except for the water stored in the thermokarst lake. We suggest including the results of this analysis in a new table that could be included in the Supplementary Material.*

*The critical issue here is the combination of (1) large amounts of snow available for melting at the beginning of June 2019 and (2) abnormally warm and dry weather conditions during June 2019, implying an extraordinary high-energy environment. The rapid meltdown of large amounts of water under high-energy conditions favored the development of a small meltwater lake on the surface of Hüttekar rock glacier. The high atmospheric energy input during June 2019 likely speeded up the development of thermal convection in this lake, bridging the thermal insulation provided by a 'dry' debris layer (air filling the pore space). Accordingly, we suppose that the convective transport of thermal energy within the lake started melting the massive ice constituting the bed of the thermokarst lake, incising the depression and promoting further accumulation of water. Once initiated, this processes allowed for the development of the large thermokarst lake that developed on Hüttekar Rock Glacier from 3 June 2019 – 13 August 2019. Thus, we propose that the crucial factors distinguishing 2019 from earlier years are the rapid and late snowmelt, the high energy environment during the summer months, and the storage of water in the lake.*

*We thank Catherine Bertrand for her valuable feedback and suggest addressing these issues by revising the method description (line 183–228) as well as the presentation of results (line 313–330). In addition, we suggest extending the discussion of implications in the revised 'Discussion' section, following the lines of evidence outlined above.*

**RC:** The particle size analysis was done on 8 samples. Is this enough to cover the whole surface?

*AR: We agree with Catherine Bertrand that this is a critical issue. Since the internal structure of rock glaciers is highly heterogeneous, the 8 samples cannot be regarded representative for the entire rock glacier. To clarify this point and to restrict the analysis to the most essential features of the studied rock glacier with respect to aims of our study, we suggest cancelling the discussion of the coarse-grained surface layer and focusing on the composition of the rock glacier front, which is where the debris flow initiated. This is in line with comments provided by Anonymous Referee #1. We revised Fig. 3 (please see above) accordingly, and restricted Fig. 6 to the*

grain size distribution of the rock glacier front (please see below). We suggest deleting line 284–290 and replacing line 165–172 as follows:

*"The structural characteristics of the material involved in the debris flow are inspected by analyzing the grain size distribution of the rock glacier front. In August 2009 four samples were taken 50 m southwest of the later debris flow initiation zone and analyzed by wet sieving (position indicated in Fig. 3). Due to the heterogeneous structure of the rock glacier, these samples are not representative for the grain size composition of the entire rock glacier. In contrast, the material at the rock glacier front appears comparatively homogenous. Considering the proximity of the sampling locations to the subsequent debris flow initiation zone, and the dangerous sampling conditions at active rock glacier fronts in general, we consider the samples reasonable approximations to the grain size composition of the material that was mobilized during the debris flow event on 13 August 2019."*

In addition, we suggest including the grain size analysis data in the 'Supplementary Material'. The revised Fig. 6 is presented below:

[Figure]

***Figure 6.*** *Grain-size distributions of material composing the rock glacier front. The corresponding sampling locations are 50 m southwest of the debris flow initiation zone, as indicated in Fig. 3. Dashed grey lines represent source compositions of a set of experimentally investigated debris flows showing contractive shear response and undrained failure (USGS debris flow flume; Major (1996); Iverson (1997); Iverson et al. (1997)). Classification according to ISO 14688-1 (International Organization for Standardization, 2017).*

**RC:** The discussion is really interesting and poses the problem in the long term, especially the effect of climate change which accelerates the degradation of permafrost and favours the creation of a thermokarst and the possible consequences.

***AR:*** *We thank Catherine Bertrand for her appreciation and her precise summary of the implications of permafrost degradation and thermokarst development in a changing mountain environment. To highlight this key result of our study more clearly, we suggest putting more emphasis on this topic in the 'Discussion' section, in turn shortening the literature review provided in the original manuscript.*

**RC:** Finally, the authors point out that debris flows are initiated by the destabilisation of rocky glacier fronts and most often occur in response to heavy rainfall. They also state that intense snowmelt or rain-on-snow events and exceptionally warm periods have also been identified as triggering factors. My question is, does climate change mean less snow and less rain? If so, what is the real impact of climate change on the risk? Can the authors elaborate on how they pro climate change: more snowmelt because more heat with less rain? less snow so a lower snowline and fewer glaciers? What is their scenario?

***AR:*** *We thank Catherine Bertrand for this valuable suggestion and will include an additional paragraph in a revised version of the 'Discussion' section addressing these issues. With respect to thermokarst evolution, the key*

*issue is the expected increase in available melting energy that is closely linked to rising air temperatures. Since the development of thermokarst features, such as channel networks, depends critically on the energy input, we expect the establishment of these features in areas currently characterized by continuous permafrost that generally acts as an aquitard. Our study has shown that such channel systems might develop within weeks, and subsequently are able to rapidly transfer large amounts of water. It is this rapid and hardly predictable development of thermokarst features that challenges the evaluation of slope stability under a warming mountain climate. Once established, the channel system might concentrate water at points of converging channels regardless of the details of the triggering mechanism.*

**RC:** In conclusion, this article sheds important light on the risks involved weather conditions. The proposed retrospective analysis is interesting and deserves to be published in your journal.

*AR: We thank Catherine Bertrand for her valuable and constructive feedback and would welcome the opportunity to address the issues outlined above in a revised version of the manuscript.*

---

## Author Response (AR1)

**Author's response to Anonymous Referee #1**

We thank Anonymous Referee #1 for his or her time and effort in reviewing our manuscript. We sincerely appreciate the constructive feedback and helpful comments that will improve the quality of the manuscript. Below we have addressed all the points by stating the referee comments (RC) and outlining the corresponding author's response (AR) in *italics*.

**RC:** The manuscript investigates a cascade of processes in the Radurschl Valley in the Austrian Alps. The outburst of a (supra-) glacial lake initiated the event, which continued with the failure of a rock glacier front with subsequent debris flow initiation. The deposit of the latter caused the blocking of the valley and the formation of a lake, which was drained to manage the risk related to its possible outburst. The series of events and their description are very interesting and, in my opinion, would enrich the scientific literature. However, I have some concerns regarding the manuscript, which in my opinion could profit from some revisions.

The methods are described at a level of detail that is not sufficient for understanding the analysis and reproduce the study. In this sense, they need to be revised and improved. The results and the discussion are mixed at several points in the text, with several repetitions and omissions. The logic structure is at times difficult to follow.

As a general suggestion, I would welcome a major revision of the text. Firstly, the authors could more effectively highlight the original analysis and results of the study, avoid substantial repetitions, and increase the logic structure of the manuscript. Secondly, the use of technical language should be always preferred to common-language expressions (e.g. "temporal changes" vs "dynamics", see detailed comments later) in all cases where this is possible and the methods should be described in more detail. Some minor analysis and addition to the data presented might improve the manuscript and its relevance in the community.

In the following, I provide some more specific comments, that I believe could improve the manuscript. I hope that the authors can receive the feedback in the in the same spirit in which it was written and for what it is: constructive suggestion to improve the paper.

*AR: We welcome the constructive feedback provided by Anonymous Referee #1 and thank him or her for regarding our contribution as relevant to the scientific literature. We believe that the manuscript benefited from the suggested revisions based on the referee comments. Below we address the outlined issues point by point, including major revisions of the text, the figures and the provided Supplementary Material.*

*Specifically, we streamlined the description of methods, eliminated repetitions and omissions, and highlighted the open-source based nature of our study in the revised manuscript. We provided the corresponding public access options and included the data collected by the authors in the Supplementary Material, ensuring full reproducibility of our analysis. We reduced the recapitulation of previous literature substantially and replaced generic statements by specific descriptions. Regarding the use of technical language, we revised the entire manuscript accordingly, aiming at more precise descriptions and avoidance of redundant adjectives. We added additional data and analysis based on the valuable suggestions of Anonymous Referee #1 and Catherine Bertrand (Referee #2), improving the lines of evidence according to the aims of our study.*

**RC: Page 1, Line 1:** Is the front of the rock glacier failing or are the debris accumulated in the gully (below the rock glacier) that are triggered and flow downhill? I write the comment here, but it applies to several other sections in the manuscript. I think that there is a distinction between rock glacier failure, active layer detachment, and debris flow initiation at a rock glacier front. Neglecting the differences between all these processes is in my opinion a first-order approach, which might well be justified in certain contexts, but should not be used in this manuscript, where the focus is on this specific topic."

*AR: We recognize that precise terminology is essential. The paper addresses debris flow initiation at a rock glacier front, as indicated by distinct features of the debris flow initiation zone (Figures 4a and 4b). The appearance of cut-off ice lenses (Figure 4b) as well as the depth/width ratio (Figure 4a) preclude an active layer detachment. The rock glacier is not undergoing a phase of destabilization as entire landform, but local failure in a specific part of the rock glacier front, where a newly formed meltwater channel connects it to a temporary thermokarst lake (please see also RC and AR addressing page 2, line 55; RC and AR addressing page 8, figure 4; RC and AR regarding page 6, line 111 and following; as well as RC and AC addressing page 14, line 308). We revised the entire manuscript accordingly, ensuring consistent and precise language.*

**RC: Page 1, Line 3**: What does "multivariate permafrost degradation" mean? What is "retrogressive debris flow"?

*AR: We replaced "multivariate permafrost degradation" by "permafrost degradation" as well as the term "retrogressive debris flow" by "debris flow", as these are the essential features we would like to highlight in the abstract.*

**RC: Page 1, Line 5:** What do you mean by "environmental forces" and what are the "ambiguous conditions"?

*AR: We replaced the sentence in line 5–7 by "We present a systematic analysis of destabilizing factors to deduce the failure mechanism." We acknowledge that the terms "environmental forces" and "ambiguous conditions" lack the desired precision and are not strictly necessary.*

**RC: Page 1, Line 6:** The paper is an application of the methodology presented in Glade and Crozier, 2005. If I understood this correctly, the authors do not establish a basis for multi-hazard assessment but apply an existing method. If contrarily the authors argue that they implemented the methods for hazard assessment, I would welcome a clearer explanation of the original methodology, particularly with regards to the concept of hazard.

*AR: As correctly indicated by Anonymous Referee #1, the paper applies the methodology presented by Glade and Crozier (2005). We agree that our original description was misleading and deleted the phrase "and establish a basis for multi hazard assessment in similar settings".*

**RC: Page 2, Line 55:** I would argue that destabilizing rock glaciers are defined by the combination of abnormal velocities and the presence of surface features such as cracks and scarps. Several recent papers in the literature provide a more accurate description of rock glacier destabilization.

*AR: We agree with the referee's argument. Following the general recommendations of Anonymous Referee #1 to highlight the original analysis and results of the study, we revised the entire paragraph, focusing on key issues regarding the studied process. Accordingly, we deleted the description of destabilizing rock glaciers, since the studied rock glacier is not undergoing a phase of destabilization as entire landform, but local failure in a specific part of the rock glacier front.*

**RC: Page 3, Line 63:** reading this line, the expectations to this paper sky-rocketed. Unfortunately, I must say that they were not satisfied by the manuscript, especially with regards to hydrogeological and mechanical aspects (some comments later). With this I don't want to suggest major revisions, but simply to be more precise in the description of the goals of the paper.

*AR: We thank Anonymous Referee #1 for communicating concerns about sky-rocketed expectations and deleted the sentence in line 62–64. In addition, we revised the last paragraph of the introduction, thereby clarifying the goals of the paper.*

**RC: Page 4, Line 96:** This sentence is in my opinion very controversial. Are the authors stating that the rock glacier originated 170 years ago as a result of the LIA glacier advance? The next sentence quickly describes the surface morphology of the terrain influenced by recent glacier advances. Why not considering the simplest and most-supported hypothesis that the rock glacier pre-existed and was simply overridden by a glacier during the LIA? Anyway, I would simply avoid this argument, which is irrelevant for the discussion of the results and the topic presented in the paper. If the authors want to keep the statement, I would welcome more extensive explanations of their theory.

*AR: We agree and follow the suggestion of Anonymous Referee #1 to avoid this argument. Thus, we deleted the two sentences in line 96–99 and revised the associated paragraph (please see also RC and AR addressing page 5, line 105). Accordingly, we deleted the former Supplementary Figure S1.*

**RC: Page 5, Line 105:** how do the authors know that the channels are eroded into the ice core? Are the authors referring to water flowing on top of the permafrost table or to water infiltrating into the ice-core? This is a very interesting topic and if observations are available, they could be presented and discussed in more detail. If no more details are available, I would still welcome a clearer description of the observations (if considered relevant).

*AR: We follow the suggestion of extending the description of our observations, including more relevant details and presentation of the available material. Regarding the observed channels along the rock glacier permafrost table, we suggest deleting the term 'eroded' to avoid hypotheses about the generating process at this point. We revised our description to include all available details, given the difficulties in precisely identifying flow paths in active rock glaciers. The updated description is complemented by a new figure including a photograph of an observed water current along the permafrost table (Supplementary Figure S1).*

*While it is possible that some water is infiltrating into the rock glacier ice-core as well, we cannot prove this by direct observations and thus avoid speculations regarding this issue. In the context of the present study, we consider it sufficient to demonstrate the establishment of a thermokarst channel connecting the thermokarst lake to the debris flow initiation zone, irrespective of its exact vertical position. Evidence for this connection is explicitly highlighted in the revised manuscript and described in detail (please see also RC and AR addressing page 7, line 123). Additional evidence is provided by the new Supplementary Figures S1, S3, and S4. The trace of this thermokarst channel along the rock glacier surface is highlighted in the revised Figure 3.*

**RC: Page 6, Line 106:** what is a "retrogressive debris flow"? I see here some confusion between "retrogressive erosion" and "debris flow".

*AR: We suggest replacing the term by "debris flow".*

**RC: Page 6, Line 111 and following:** This paragraph is extremely interesting, and I suggest to extend the description (here) and discussion (later) of the observations, the numbers presented, the methodology used to derive them, and their uncertainties. These points might be extended in several locations in the manuscript, maybe replacing some of the repetitions and digressions that are currently present.

*AR: We thank Anonymous Referee #1 for expressing his or her interest and suggestions to extend the descriptions. Please note that the presented numbers merely serve as an indication of the scales involved and are not used for any calculations (now explicitly highlighted in the text). We revised the paragraph (line 109–122) accordingly, included new data, explained the underlying sources of information, and prepared a new figure showing the entire debris flow as well as details including frozen material, levees, and river blockage (Supplementary Figure S2).*

**RC: Page 7, Line 123:** Concurrently means at the same time. The question of the timing is essential for the process described. Please explain in more detail what is known (and how, see previous comment) and what are your hypothesis. Later in the manuscript, they should be tested and discussed.

*AR: To clearly distinguish observations from hypotheses, we suggest revising the entire paragraph (line 123–134), addressing the issues outlined by Anonymous Referee #1 as precisely as possible. Please note that due to the remoteness of the study site and the absence of instrumentation at the time of the event, the timing of individual processes can only be roughly reconstructed (temporal resolution on the order of several hours). In accordance with these constraints, our study presents rough estimates of the involved time scales (thermokarst lake development ~10 weeks, thermokarst lake drainage ~1 day, debris flow event ~1.5 days) that can reliably be deduced. Throughout the revised manuscript, we highlighted the fact that the discussed implications and drawn conclusions are consistently reconciled with the limited accuracy of these estimates.*

*The revised event description (line 123–134) clearly specifies what is known, includes new data, and is supplemented by a video documentation showing the rapidly draining thermokarst lake, recorded on 13 August 2019 11:20 (Supplementary Video). We summarized all available evidence concerning the establishment of a thermokarst channel connecting the thermokarst lake to the debris flow initiation zone, highlighted that the vertical position of this channel is not known precisely in the text, and provided additional photographs in the new Supplementary Figures S1, S3, and S4 (please see also RC and AR addressing page 5, line 105, as well as RC and AR addressing page 7, line 126).*

**RC: Page 7, Line 126:** it is interesting that local knowledge is taken into account. Still, I suggest to perform a systematic analysis with modern remote sensing data (Sentinel 2 for example) to quantify the evolution of the lake in recent times.

*AR: We agree with this valuable suggestion and performed a systematic analysis with modern remote sensing data, carefully scanning available archives for indications of thermokarst lakes appearing on Hüttekar Rock*

*Glacier before June 2019. In addition, we included historical and modern maps in our analysis. We suggest communicating our results as outlined in the completely revised section 3 (please see our response to the referee comment addressing page 7, line 123).*

**RC: Page 7, Line 139:** Are the authors talking about dynamic forces here? Impact from external mass movements? Earthquakes?

*AR: We acknowledge that the term "dynamic forces" as used in our original manuscript is misleading. Throughout the manuscript, we replaced the term "dynamic forces" by "dynamic factors", strictly following the nomenclature suggested by Glade and Crozier (2005). In this way, we highlight the consistent application of the factor mapping methodology for landslide susceptibility assessment provided by these authors, in accordance with our responses to the referee comments addressing page 1, line 6 (please see above) and page 9, section 4.1 (please see below), respectively.*

**RC: Page 8, from Figure 4:** Very interesting observations of the rock glacier front. What's the height of the shear horizon depth here? I would be very happy to see in the published version of the manuscript a detailed description of this area (possibly before and after the event?), which is highly relevant to the debris flow initiation. A figure showing a simple comparison before-after the event (maybe in appendix?) would be very useful. What is the thickness of the debris around the lake? And around the front? Panel g and h seem to indicate a debris covered glacier. Would It be possible to provide a more detailed geomorphological assessment of the landform and a map of it?

*AR: We thank Anonymous Referee #1 for appreciating the observations of the rock glacier front and his or her suggestions to present them in more detail. To communicate these observations as objectively as possible, we followed these suggestions by preparing a new figure that shows a comparison before-after the event from two different perspectives (Supplementary Figure S4). In addition, we extended the former Supplementary Figure S2 (now Supplementary Figure S3) showing ortho-images of Hüttekar Rock Glacier before the 2019 debris flow event (1970, 2003, 2007, 2010, and 2015) as well as after the event (2020).*

*The depth of the shear horizon is highly variable, as the debris flow initiation zone is characterized by an irregularly shaped rupture in ice-cemented debris, adjusting its shape and widening by erosion of unfrozen debris in its vicinity. These issues are indicated in the revised description of the rock glacier in Section 2, the revised description of the debris flow event in Section 3, as well as the reported predisposing factors in Section 4.1 in the revised manuscript. We summarized what is known as precisely and objectively as possible, and clearly stated what is unknown or can only be roughly estimated.*

**RC: Page 9, Section 4.1:** some of the text presented here might be more suited for the introduction. Also, the analysis of the presented factors on slope stability is not clear to me. I don't see any stability assessment, or somehow I missed it. Maybe a clearer description of the methods might help.

*AR: As correctly stated by Anonymous Referee #1, our study does not provide a full stability assessment of the slope, but rather an analysis of landslide susceptibility employing factor mapping as outlined by Crozier and Glade (2005). We revised the entire Section 4 accordingly, removing general statements that are misplaced in this section and providing a more precise method description.*

*We recognize that the open-source based nature of the entire study was not adequately highlighted in the original manuscript. The analyzed datasets are publicly available, implying that the analysis is reproducible. Currently, the only exception is the employed snow model (SNOWGRID). However, full access to this dataset will be provided in the near future via GeoSphere Austria (former Central Institute for Meteorology and Geodynamics). The grain size analysis data collected by the authors as well as surface velocity data obtained are now included in 'Supplementary Material' (Supplementary Tables S1 and S2, respectively). The public availability of the analyzed data sets and employed software is highlighted at the beginning of the revised section, and associated weblinks are provided in the 'Data availability' section.*

**RC: Page 9, Line 165 and following:** This is in my opinion an interesting approach, but cannot be introduced so lightly. Please provide more details, assumptions, rationale, and potential limitations. As known, the internal structure of rock glacier is very different from the active layer (also not homogeneous in the active layer!).

Assuming the the latter is representative of the first is not valid in my opinion, especially given the photographs presented in figure 4. Were the boulders only collected at the surface? Only evaluating the boulders at the surface creates a bias towards boulders. How were the samples collected at the front? Who went there? Where exactly were they collected? How representative should this be of the internal part of the rock glacier, especially the shear horizon if you suggest that water was flowing there (erosion/deposition at the surface)?

*AR: We agree with Anonymous Refereee #1 that the internal structure of rock glaciers is highly heterogeneous and that the differentiation between active layer, rock glacier core, and rock glacier front is essential. To clarify this point and to restrict the analysis to the most essential features of the studied rock glacier, we suggest cancelling the discussion of the coarse-grained surface layer and focus on the composition of the rock glacier front, which is where the debris flow initiated. We revised Fig. 3 accordingly, and restricted Fig. 6 to the grain size distribution of the rock glacier front. We deleted line 284–290 and revised line 165–172, addressing the issues outlined by Anonymous Referee #1. In addition, we included the grain size analysis results in the new Supplementary Table S1.*

**RC: Page 9, line 177:** this is not a sufficient description for the data used.

*AR: We revised the entire paragraph, specifying the procedure and assessing associated uncertainties.*

**RC: Page 10, line 183 and following:** is the use of climate relevant here? Are the authors not limiting the analysis to meteorological forcing? Also, what is meant by "progressive change of climatic factors"? In general, the description of the method is not sufficient for reproduction, and in this sense, it should be improved and extended.

*AR: We addressed these issues carefully by revising and extending the method description to ensure reproducibility and demonstrate the relevance of climate in the context of the study. We consider the long-term influence of climate relevant for two reasons: (1) The limiting factor governing the rate and extent of thermokarst evolution at Hüttekar Rock Glacier is the available energy input, which is outstandingly high in summer 2019 compared to earlier years (as indicated in Table 3; please see also line 188–190 on page 10 and line 315–316 on page 16). In this sense, the analysis of climate allows putting the conditions during summer 2019 in a long-term context. (2) The strong increase in air temperature and positive degree day sum, outpacing corresponding trends at the global and European Alps scale (please see line 317–320 on page 16), suggests that permafrost degradation and upward shifting of the lower permafrost boundary affected Hüttekar-cirque strongly during the years preceding 2019. We agree with Catherine Bertrand (Referee #2), who outlined that the discussion of our study results "poses the problem in the long term, especially the effect of climate change which accelerates the degradation of permafrost and favors the creation of a thermokarst and the possible consequences." (please see comment on egusphere 2022-567 by Catherine Bertrand, https://doi.org/10.5194/egusphere-2022-567-RC2). Accordingly, we believe that the results are relevant in the context of our study and revised the manuscript to explain this in detail.*

*To ensure the reproducibility of our results, we highlighted the public availability of the analyzed datasets and the employed software. Weblinks are specified in the 'Data availability section', and data collected by the authors provided in the 'Supplementary Material'. Please note that the only exception is currently the employed snow model (SNOWGRID), that will be made publicly available in the near future by GeoSphere Austria (please see also RC and AR addressing page 9, Section 4.1).*

**RC: Page 10, Line 210:** are the calculations about snow cover validated in any way?

*AR: We recognize that the original manuscript is lacking information regarding the validation of the snow model. The analyzed dataset is a subset of a nationwide dataset that is continuously updated through the distributed SNOWGRID model (Olefs et al., 2013). The resulting snow cover data are routinely evaluated and validated by GeoSphere Austria (former Central Institute for Meteorology and Geodynamics), as outlined in the revised manuscript. Additional verification stems from feedback provided by a large range of SNOWGRID applications including research projects as well as commercial applications (for an overview, please see https://www.zamg.ac.at/cms/de/forschung/ klima/klimatografien/snowgrid).*

**RC: Page 11, Line 218:** The statements about the hydrology seems to be results. What observations are available? Please expand the description of the data and methods here. E.g. why until the 13th of August?

*AR: We agree with Anonymous Referee #1 that these observations are misplaced here. We suggest reporting the role of glacial meltwater in the last paragraph of section 2 (study site), summarizing the hydrology and hydrogeology of Hüttekar-cirque (please see also RC and AR addressing page 5, line 105). The employed methods aim at maximizing comparability with a similar study performed by Kofler et al. (2021), who analyzed the initiation of debris flows from two rock glacier fronts in nearby mountain ranges. To estimate the total water input received by Hüttekar-cirque in 2019 prior to the debris flow initiation, we calculated the cumulative rainfall, snow melt and ice melt from 1 January 2019 to 13 August 2019 (failure date) and compared it to the respective water volumes in previous years (consistently totalized from 1 January to 13 August). We revised the description of the data and the methods accordingly, explaining the employed methods in more detail and strictly separating methods from results.*

**RC: Page 14, line 308:** Data of surface velocities would be very interesting. Is it possible to present them? Especially relevant if the authors speak about destabilization. Possibly you could also see some signal in ice melt (vertical component – once subtracted subsidence due to dynamics).

*AR: We thank Anonymous Referee #1 for this valuable suggestion and prepared a new figure illustrating our results (Supplementary Figure 5). In addition, we included data for individual blocks, including coordinates, displacement, and displacement rate, along with summary statistics for every epoch in Supplementary Table S2. The revised manuscript clarifies that our study addresses local failure in a specific part of the rock glacier front, not destabilization of the entire landform (please see also RC and AR addressing page 1, line 1, as well as RC and AR addressing page 2, line 55). Regarding this important distinction, we revised the entire manuscript, ensuring consistent and precise language. Therefore, in our opinion calculating the subsidence due to rock glacier dynamics and extracting some signal in ice melt is beyond the scope of the current study.*

**RC: Page 17, Figure 8:** great figure. I suggest to add the equations to the lines and expand the discussion with regards to failure processes (Lou and Similaun different from this case, no?)

*AR: We thank Anonymous Referee #1 for appreciating the figure and for providing valuable suggestions that improved the figure as well as the main text of the manuscript. We agree with these suggestions and updated Figure 8 accordingly, now including the equations. As correctly noted by Anonymous Referee #1, the documented debris flows at Murfreit Rock Glacier, Lou Rock Glacier, Similaungrube Rock Glacier, Hintergrat Rock Glacier, and Ritigraben Rock Glacier occurred in response to heavy rainfall, in contrast to the debris flow at Hüttekar Rock Glacier. To highlight this key difference, we revised line 336–340 of the manuscript, emphasizing the specific situation found in our study.*

**RC: Discussion and Conclusions:** I would welcome a more detailed discussion of the original results of the manuscript, limiting long review of previous literature. Same for the conclusions. Maybe some of the comments above can help in the process.

*AR: We thank Anonymous Referee #1 for his or her valuable comments and suggestions. In our opinion, these comments were of great value during the process of manuscript revision. Specifically, we restricted the review of previous literature and focused on the original results of the manuscript, including (1) the unfavorable predisposition of the front of Hüttekar Rock Glacier, (2) the rapid development and sudden drainage of the thermokarst lake on Hüttekar Rock Glacier, driven by extraordinarily high atmospheric energy input during summer 2019 following a long-lasting general warming trend promoting permafrost degradation, (3) the quantification of the energy available for thermokarst evolution at Hüttekar Rock Glacier, critically linked to the development of the thermokarst lake and channel considering the considerable distance (about 350 m) between thermokarst lake and rock glacier front, and (4) implications for future considerations regarding debris flow initiation in permafrost-affected terrain.*

**Author's response to Catherine Bertrand**

We thank Catherine Bertrand for her time and effort in reviewing our manuscript. We sincerely appreciate the constructive feedback and helpful comments that will improve the quality of the manuscript. Below we have addressed all the points by stating the referee comments (RC) and outlining the corresponding author's response (AR) in *italics*.

**RC:** This study analyses the possibility of a gravitational event occurring as a result of a cascade of rapid events, namely the rupture of a thermokarst lake, the rupture of a rock glacier front, the development of a debris flow and the blockage of a river impacting the Radurschl valley (Ötztal Alps, Tyrol) on 13 August 2019. The analysis of the overflow risk of an artificial lake created during such events is a topical issue with important societal challenges.

This article proposes an interesting methodology based on multi-criteria analysis that is of interest to the scientific community. It is an interesting article well-argued and can be published in your journal.

The bibliographical references are numerous. These citations support the scientific demonstration but sometimes make the reading more difficult. For example, is it necessary to cite references from 1984 to 2010 (see line 231) or between 1990 and 2010 in line 235. It is not a question of ignoring previous work but of finding the right balance between citations.

Also, there are many examples where authors are quoted twice in two consecutive sentences (line 137 and 138 for Glade and Crozier et al., lines 183 and 185 for Patton et al. to name just two examples). Is this necessary? Why not choose the most striking points of the work without wanting to refer to everything that is said in the article. I think the readability would be improved by reworking the citations. But this is a side remark that does not detract from the quality of the work.

The article is well constructed, and the argument is well presented.

*AR: We welcome the constructive feedback provided by Catherine Bertrand and thank her for regarding our contribution as relevant to the scientific community, for highlighting the associated societal challenges, and for appreciating the article construction and our presented line of argument. We recognize that the numerous bibliographical references made the manuscript difficult to read and followed her recommendations regarding the right balance between citations. Accordingly, we revised the entire manuscript, reworking the citations by focusing on the most striking points, avoiding quotes in consecutive sentences as well as excessive quotations, and restricting recapitulation of previous literature to highlight the original analysis and results of our study.*

**RC:** In the introduction why not mention the study site in the last paragraph?

*AR: We agree with Catherine Bertrand and revised the last paragraph of the introduction (line 65–69) accordingly to describe the goals of the paper more precisely.*

**RC:** Line 63 the authors underline the fact that the assessment of the hazard potential of rock glaciers requires an integrated approach combining hydrogeological, meteorological, thermal, geomorphological, and mechanical aspects in a coherent framework. As these aspects are well identified in the introduction, why not go into more detail later?

*AR: We agree with Catherine Bertrand and suggest deleting the sentence in line 62–64. The revised manuscript includes a more detailed description of the applied methodology, as presented in the revised Section 4.*

**RC:** The hydrogeological descriptions are described succinctly from line 104 onwards. What arguments do you have to say that the cirque is drained only by subsurface water? Do you have any idea of the mineralization of the water from your various sources? Is it possible to indicate them on your figure 2 or 3. What arguments do you have for saying that the water flows along distinct channels eroded into the ice core?

*AR: We follow the suggestion of extending the description of our observations, including more relevant details and presentation of the available material. The slope downstream from Hüttekar-cirque, representing its orographic catchment outlet and connecting it to Radurschl Valley, does not show any indications of surface water. A group of springs emerging at the toe of the slope is the only observable outflow from Hüttekar-cirque.*

*We mapped these springs and measured the electrical conductivity of spring water during several field surveys between 2019 and 2022 (results included in the revised study site description). We followed the suggestion of Catherine Bertrand and included the position of these springs in the geological-geomorphological map (revised Fig. 3). In addition, we present a new figure (Supplementary Figure S1) documenting the absence of surface water bodies in Hüttekar-cirque as well as along the downstream slope connecting it to Radurschl Valley. Some of the springs at the toe of the slope are visible in two of the photographs and are outlined accordingly.*

*Regarding the observed channels along the rock glacier permafrost table, we suggest deleting the term 'eroded' to avoid hypotheses about the generating process at this point. We revised our description to include all available details, given the difficulties in precisely identifying flow paths in active rock glaciers. The updated description is complemented by the new Supplementary Figure S1, including a photograph of an observed water current along the permafrost table.*

**RC:** Line 105 you refer to a considerable catchment area when the only indication given of the size of the catchment area is 2.8km$^2$. Can you provide additional information to clarify this point?

*AR: We suggest avoiding the term "considerable" as it could be misleading here. While Hüttekar-cirque is the most prominent cirque in Upper Radurschl Valley, we acknowledge that it cannot be regarded "considerable" in a general sense. Accordingly, we deleted the term in the revised manuscript.*

**RC:** Line 88 can you clarify what you mean by "moderate annual precipitation"?

*AR: We suggest clarifying this issue by comparing it to the areal average across Austria (1077 mm), as outlined in the revised manuscript.*

**RC:** Table 1 the legend of your table shows a chronicle of data from 1976 to 2019 which is based on references from 2016 and 2018, there is a problem in your dates.

*AR: We acknowledge that our original table caption was misleading. The values reported in Table 1 are based on the continuously updated SPARTACUS dataset, with the underlying methodology described by Hiebl and Frei (2016, 2018). We updated the table caption accordingly.*

**RC:** 2019 does seem to be a special year from a meteorological point of view, with more snow and precipitation than the previous 4 years. Since 1976, have there been similar rain and snow conditions as those that caused the disaster, i.e. have you analysed all your weather data by year with the same finesse to see if there were similar weather conditions without necessarily reaching the 2019 disaster?

*AR: We agree with Catherine Bertrand that 2019 was a special year from a meteorological point of view. Specifically, air temperature, available melting energy as well as the timing and rate of snowmelt distinguished it from the previous years, suggesting that these are the most promising climate and weather factors for further analysis. In contrast, rainfall and ice melt volume were moderate in 2019 compared to the previous years. We revised and extended the associated paragraph accordingly and discussed implications regarding climate change in the revised 'Discussion' section (please see also RC and AR regarding the impact of climate change on hazard potential given below).*

**RC:** The particle size analysis was done on 8 samples. Is this enough to cover the whole surface?

*AR: We agree with Catherine Bertrand that this is a critical issue. Since the internal structure of rock glaciers is highly heterogeneous, the 8 samples cannot be regarded representative for the entire rock glacier. To clarify this point and to restrict the analysis to the most essential features of the studied rock glacier with respect to aims of our study, we suggest cancelling the discussion of the coarse-grained surface layer and focusing on the composition of the rock glacier front, which is where the debris flow initiated. This is in line with comments provided by Anonymous Referee #1. We revised Fig. 3 accordingly and restricted Fig. 6 to the grain size distribution of the rock glacier front. We suggest deleting line 284–290 and adapting line 165–172 as outlined in the revised manuscript. To enhance the reproducibility of our results, we included the grain size analysis data in the 'Supplementary Material' (Supplementary Table S1).*

**RC:** The discussion is really interesting and poses the problem in the long term, especially the effect of climate change which accelerates the degradation of permafrost and favours the creation of a thermokarst and the possible consequences.

*AR: We thank Catherine Bertrand for her appreciation and her precise summary of the implications of permafrost degradation and thermokarst development in a changing mountain environment. To highlight this key result of our study more clearly, the revised manuscript puts more emphasis on this topic, in turn shortening the literature review provided in the original manuscript.*

**RC:** Finally, the authors point out that debris flows are initiated by the destabilisation of rocky glacier fronts and most often occur in response to heavy rainfall. They also state that intense snowmelt or rain-on-snow events and exceptionally warm periods have also been identified as triggering factors. My question is, does climate change mean less snow and less rain? If so, what is the real impact of climate change on the risk? Can the authors elaborate on how they pro climate change: more snowmelt because more heat with less rain? less snow so a lower snowline and fewer glaciers? What is their scenario?

*AR: We thank Catherine Bertrand for this valuable suggestion and included an additional paragraph at the end of the 'Discussion' section addressing this point.*

**RC:** In conclusion, this article sheds important light on the risks involved weather conditions. The proposed retrospective analysis is interesting and deserves to be published in your journal.

*AR: We thank Catherine Bertrand for her valuable and constructive feedback and hope that the revised manuscript addresses the outlined issues appropriately.*

**Author's changes in the manuscript**

**Page 1, line 1:** "rock glacier front failure" changed to "local rock glacier front failure".

**Page 1, line 3:** "multivariate" deleted.

**Page 1, line 3:** "drainage network development" replaced by "drainage network development within the rock glacier".

**Page 1, line 5:** "Since the environmental forces inducing the debris flow evolved under ambiguous conditions, potentially destabilizing factors were analyzed systematically to deduce the failure mechanism and establish a basis for multi hazard assessment in similar settings." replaced by "We present a systematic analysis of destabilizing factors to deduce the failure mechanism."

**Page 1, line 6:** "revealed" replaced by "reveals".

**Page 1, line 8-9:** "thermokarst and debris flow" replaced by "a thermokarst network".

**Page 1, line 9:** "characterizing the hydraulic configuration" deleted.

**Page 1, line 10-12**: "The large amount of mobilizable sediment, dynamically changing internal structure, and substantial water flow along a rapidly evolving channel network eroded into the permafrost body, render active rock glaciers complex multi hazard elements in periglacial, mountainous environments." replaced by "Our results demonstrate the hazard potential of active rock glaciers due to their large amount of mobilizable 10 sediment, dynamically changing internal structure, thermokarst lake development, and substantial water flow along a rapidly evolving channel network."

**Page 1, line 14:** redundant reference deleted.

**Page 1, line 21-22:** redundant references deleted.

**Page 2, line 26-27:** redundant references deleted.

**Page 2, line 28-29** redundant references deleted.

**Page 2, line 33-34:** redundant references deleted.

**Page 2, line 37-41:** "In terms of landslide initiation, the steep slopes and large sediment accumulations provided by active rock glacier fronts pose a serious hazard. Active rock glaciers, i. e. distinct sediment accumulations consisting of ice-debris mixtures slowly creeping downhill, constitute important periglacial landforms regarding sediment dynamics and hydrology due to their large sediment transfer capability, pronounced water storage capacity, and wide-spread occurrence in mountainous terrain (Kummert and Delaloye, 2018; Winkler et al., 2018; Jones et al., 2019b; Hayashi, 2020; Wagner et al., 2020, 2021a)." replaced by "The large sediment accumulations provided by active rock glaciers, i. e. distinct ice-debris accumulations slowly creeping downhill, pose a serious hazard in terms of landslide initiation."

**Page 2, line 46-47:** redundant references deleted.

**Page 2, line 48-52:** "Active rock glaciers exhibit complex drainage patterns indicating a dual groundwater flow system, where large amounts of water are rapidly transported along a network of convoluted meltwater channels eroded into the frozen rock glacier core (Wahrhaftig and Cox, 1959; Potter, 1972; White, 1971; Johnson, 1978; Giardino et al., 1991; Burger et al., 1999; Krainer and Mostler, 2002; Vonder Mühll et al., 2003; Arenson et al., 2010; Springman et al., 2012; Winkler et al., 2018; Jones et al., 2019b; Wagner et al., 2021b; Kainz, 2022)." Replaced by "Common deformation rates are on the order of decimeters to meters per year, accelerating across the European Alps in response to climate change (Hock et al., 2019; Marcer et al., 2021). Previous research demonstrated that hydrological processes fundamentally control the style and rate of rock glacier kinematics (Ikeda et al., 2008; Buchli et al., 2013, 2018; Cicoira et al., 2019). However, while the impact of thermokarst evolution on the storage and release of water in these landforms is well documented, its role in debris flow initiation at rock glacier fronts remains largely unknown (Krainer and Mostler, 2002; Winkler et al., 2018; Jones et al., 2019; Kainz, 2022)."

**Page 2, line 55-57:** "Destabilizing rock glaciers frequently experience significant acceleration, often accompanied by the appearance of morphological discontinuities such as cracks and crevasses (Kääb et al., 2007; Roer et al., 2008; Stoffel and Huggel, 2012; Delaloye et al., 2013; Marcer et al., 2019)." deleted.

**Page 3, line 58:** "(e.g., Krainer et al. (2012); Marcer et al. (2020); Kofler et al. (2021))" added.

**Page 3, line 59-61:** redundant references deleted.

**Page 3, line 62-64:** "Thus, assessing the hazard potential of rock glaciers requires an integrated approach combining hydrogeological, meteorological, thermal, geomorphological, and mechanical aspects in a coherent framework." deleted.

**Page 3, line 65-69:** "The aim of this paper is to explore the destabilizing factors leading to failure of an active rock glacier front in a high mountain cirque, and the actuated cascading processes involving thermokarst lake outburst flood, debris flow and river blockage. We evaluate a set of potentially contributing factors, assess critical combinations, and develop a consistent conception explaining debris flow initiation and evolution. Similarities and differences with respect to documented rock glacier front failures and glacial lake outburst floods (GLOFs) are analyzed, and conclusions drawn regarding hazard potential." replaced by "The aim of this paper is to explore the destabilizing factors leading to local failure of an active rock glacier front in the high mountain cirque Hüttekar in the Austrian Alps. We analyze the cascading processes involving thermokarst lake outburst, debris flow, and river blockage. By evaluating a set of potentially contributing factors and assessing critical factor combinations, we develop a consistent conception explaining debris flow initiation at the rock glacier front. Similarities and differences with respect to documented debris flows at other rock glacier fronts are analyzed, and conclusions drawn regarding hazard potential."

**Page 3, line 72:** "of glacial and periglacial origin" deleted.

**Page 3, line 73:** "areas" replaced by "domains".

**Page 3, line 75:** redundant references deleted.

**Page 3, line 76:** "the" deleted.

**Page 3, line 77-79:** "The altitudinal range in the study area varies from 2,387 to 3,353 m a. s. l. (mean altitude 2,870 m a. s. l.), including a relatively flat valley bottom between 2,600 and 2,700 m a. s. l." replaced by "Its altitude ranges from 2,387 to 3,353 m a. s. l., including a relatively flat valley bottom between 2,600 and 2,700 m a. s. l. (mean catchment altitude 2,870 m a. s. l.)."

**Page 3, line 79-83:** "The bedrock lithology is composed of metamorphic rocks of the Ötztal–Stubai–Complex (Ötztal–Bundschuh nappe system; Hoinkes and Thöni (1993); Schmid et al. (2004)), as illustrated in Fig. 3. The mountain ridges bordering the cirque to the south and to the east are dominated by orthogneiss (augen- and flasergneiss), the ridge to the north and northwest exhibits muscovite-granite-orthogneiss. At Rotschragenjoch and south of Bruchkopf, paragneiss and micaschist with thin intercalations of amphibolite are exposed." deleted (bedrock geology is now described in a separate paragraph, please see below)

**Page 3, line 83-86:** "The terrain is composed of bedrock (38 %) mainly exposed in the highest parts, talus, and debris slopes (17 %) along the valley sides, while the lower parts are covered by moraine deposits 85 (12 %) and four rock glaciers (27 %). Two small, north facing cirque glaciers (Glockturmferner, Hüttekarferner; 6 %) are situated at 2,853 and 3,029 m a. s. l., respectively." replaced by "The terrain is composed of bedrock (38 %), mainly exposed in the highest parts. Talus and debris slopes (17 %) dominate along the valley sides, while the lower parts are covered by moraine deposits (12 %) and four rock glaciers (27 %). Two small, north facing cirque glaciers (Glockturmferner, Hüttekarferner; 6 %) are situated at mean altitudes of 2,853 and 3,029 m a. s. l., respectively (Fig. 3)."

**Page 3, line 86:** A separate paragraph describing bedrock geology is introduced in the revised manuscript. The new paragraph reads "The bedrock lithology is composed of metamorphic rocks of the Ötztal–Stubai–Complex (Ötztal–Bundschuh nappe system; Hoinkes and Thöni (1993); Schmid et al. (2004)). The mountain ridges bordering the cirque to the south and to the east are dominated by orthogneiss (augen- and flasergneiss), the ridge to the north and northwest exhibits muscovite-granite-orthogneiss (Fig. 3). At Rotschragenjoch and south of Bruchkopf, paragneiss and micaschist with thin intercalations of amphibolite are exposed." (please see also changes on page 3, line 79-83).

**Page 3, line 87:** redundant reference deleted.

**Page 3, line 88-89:** "Moderate annual precipitation (1042 mm) and low mean annual air temperature (-2.5 °C) reflect the high altitude of Hüttekar-cirque and its central position close to the main chain of the Alps." Replaced by "With respect to long-term (1976-2019) averages across Austria, annual precipitation is moderate (1042 mm) and mean annual air temperature is low (-2.5°C ), reflecting the high altitude of Hüttekar-cirque and its central position close to the main chain of the Alps (corresponding areal averages across Austria are 1077 mm and 6.6°C, respectively)."

**Page 4, Figure 1:** "Hüttekar" replaced by "Hüttekar-cirque" in the figure caption.

**Page 4, Table 1:** Table caption changed from "Long-term (1976–2019) mean monthly air temperature and precipitation in Hüttekar-cirque (based on Hiebl and Frei (2016, 2018))." to "Long-term (1976–2019) mean monthly air temperature and precipitation in Hüttekar-cirque. Values are calculated based on the continuously updated SPARTACUS dataset (Spatiotemporal Reanalysis Dataset for Climate in Austria), as described in Hiebl and Frei (2016, 2018)."

**Page 4, line 91:** "The local altitude distribution and dry climatic conditions promote the development of rock glaciers." replaced by "The dry climatic conditions and the local elevation distribution promote the development of rock glaciers."

**Page 4, line 94-95:** redundant reference deleted.

**Page 4, line 96-99:** "The coevolution of cirque glaciers and rock glaciers is highlighted by comparing a set of glacier inventories ranging from the little ice age (about 1850 CE) to 2015 CE (Fig. S1; Fischer et al. (2015); Buckel et al. (2018)), documenting the development of Glockturmferner from a pure ice glacier to a largely debris covered glacier, presumably transitioning into the ice-cored rock glacier below (Anderson et al., 2018; Jones et al., 2019a; Knight et al., 2019)." deleted.

**Page 4-6, line 99-104:** "Along the surface of Hüttekar Rock Glacier, distinct ridges and furrows are visible in the northernmost part, while the central and southern parts show a smooth and flat surface morphology. Rock glacier debris is composed of orthogneiss derived from the Glockturm massif. Poorly sorted boulders, arranged in a loose, clast-supported structure form a heterogeneous surface layer of variable thickness. The blocky surface layer covers a finer-grained, frozen layer dominated by well-graded gravel and sand that is exposed along the rock glacier front." replaced by "The largest rock glacier in this cirque, Hüttekar Rock Glacier, is 1408 m long and up to 493 m wide, covering an area of 0.5 km$^2$ in the lowermost parts of the cirque (Fig. 3, Supplementary Fig. S1). Its gently sloping surface is characterized by distinct furrows and ridges, while the steep front rests on top of a slope above Radurschl Valley. Rock glacier debris is composed of orthogneiss derived from the Glockturm massif. In the southeast, the lower, debris covered parts of Glockturmferner transition into the rock glacier rooting zone. The exact boundary between debris covered glacier and rock glacier is not known. Massive ice is frequently visible beneath a ~1–2 m thick debris layer in a shallow depression at the southeastern edge of the rock glacier, 90 close to the suspected transition zone. The debris layer consists of poorly sorted boulders with individual blocks measuring up to 4 m, arranged in a loose, clast-supported structure. The blocky surface layer covers a finer-grained layer dominated by poorly sorted gravel and sand that is exposed along the rock glacier front. The unfrozen domain of this heterogeneous debris layer increases irregularly in thickness towards the rock glacier front, reaching a maximal thickness of ~5–10 m."

**Page 6, Figure 3:** Figure 3 revised. In addition, the following terms and statements have been deleted from the figure caption: "provisional" (line 1), "preliminary" (line 3), and "Numbers indicate grain size analysis sampling locations." (line 4).

**Page 6, line 104-107:** "Water flowing along the permafrost table is visible and audible between boulders at several places in the southern part of the rock glacier, following distinct channels eroded into the ice core. Despite its considerable catchment area, Hüttekarcirque is drained exclusively by subsurface flow, emerging as a group of small springs at the toe of the slope descending to Radurschltal." replaced by "A small meltwater current from Glockturmferner infiltrates into the rock glacier rooting zone. Water flowing along the permafrost table is visible and audible between boulders at several places in the southern part of the rock glacier. These water currents follow distinct channels that can be traced below the boulders covering the rock glacier surface. Where visible between the boulders, the channels are up to 1 m wide and 20 cm deep (Supplementary Fig. S1). Surface water bodies in Hüttekar-cirque are restricted to small meltwater lakes and currents in immediate vicinity of the two cirque glaciers. The absence of surface creeks in the lower parts of Hüttekar-cirque indicates that it is drained exclusively by subsurface flow, emerging as a group of small springs at the toe of the slope descending to Radurschl Valley (Fig. 3, Supplementary Fig. S1). The exact number and position of these springs varies during the year, but they are constrained to the sedimentary cone covering the toe of the slope, as evident from 23 field surveys between 2019 and 2022 covering all months. Spring discharge of individual outlets is < 1 l/s and electrical conductivity ranges from 60 to 75 µS/cm, as indicated by repeated measurements during these surveys." The new Supplementary Figure is provided in the revised 'Supplementary Material'.

**Page 6, line 109-110:** "An active, retrogressive debris flow erodes the steep slope bordering Hüttekar-cirque to the west (Fig. 2; classification according to Varnes (1978); Cruden and Varnes (1996); Hungr et al. (2014))."

replaced by "A debris flow eroded the steep slope bordering Hüttekar-cirque to the west (Fig. 2; classification according to Hungr et al. (2014))."

**Page 6, line 110:** "The event description given below is based on witness reports by JosefWaldner (staff of the nearby hut Hohenzollernhaus) and Gerhard Schaffenrath (local shepherd). In addition, the debris flow was documented during several field surveys as well as two helicopter flights by Roman Außerlechner, Thomas Figl and Werner Thöny (Geological Survey of Tyrol) on 14 and 26 August 2019, respectively." added (please see also changes on page 6, line 111-112).

**Page 6, line 111:** "volume" replaced by "debris volume".

**Page 6, line 111-112:** "The event description is based on in situ observations of Josef Waldner, staff of the nearby hut Hohenzollernhaus (pers. comm.)." deleted (please see also changes on page 6, line 110).

**Page 7, line 116:** "Supplementary Fig. S2" added. The new Supplementary Figure is provided in the revised 'Supplementary Material'.

**Page 7, line 117:** "deposition" deleted.

**Page 7, line 117-118:** "and an estimated volume of 40,000-50,000 m³" deleted (please see also changes on page 7, line 122).

**Page 7, line 118:** "at 2,200 m a. s. l. (Fig. 4c, Fig. 4d)" replaced by "at an elevation of 2,200 m a. s. l. (Fig. 4c and 4d, Supplementary Fig. S1 and S2)."

**Page 7, line 120:** "Supplementary Fig. S1 and S2" added.

**Page 7, line 122:** A new paragraph is introduced in the revised manuscript: "Based on in situ observations, aerial photographs, and several surveys conducted before as well as after the debris flow event, a total volume of 40,000 – 50,000 $m^3$ of mobilized sediment is estimated, providing a rough indication of the event magnitude. The quoted water depths are reconstructed from in situ observations and reviewed by mapping the maximum lake before and after channel excavation, and corresponding volume calculations employing a high-resolution (1 × 1 m) digital terrain model (DTM) based on airborne laserscanning data acquired one year before debris flow occurrence (Government of the Province of Tyrol, 2021a)." (please see also changes on page 7, line 117-118 as well as changes on page 9, line 151-153).

**Page 7, line 123-134:** "Concurrently with the debris flow initiation, a thermokarst lake on top of Hüttekar Rock Glacier (∼350 m behind the initiation zone; Fig. 3, Fig. 4e, Fig. 4f) drained almost completely within one day. The thermokarst lake had started to develop coincidently with the onset of snowmelt in early June 2019 within a shallow depression where massive ice within the rock glacier prevented drainage (Fig. 4g). During the last decades, a comparable feature had never been observed before in Hüttekarcirque, despite frequent visits by hikers, hunters, shepherds, and staff of Hohenzollernhaus. In the stage of its largest extent, the thermokarst lake was approximately 300 m long, up to about 150 m wide and 4–5 m deep, comprising an estimated water volume of ∼150,000 $m^3$. Effective drainage through a large crevasse (width ∼1.5 m, height ∼2 m) caused initiation of the debris flow at the front of the rock glacier (Fig. 4e, Fig. 4f, Fig. 4h). Large amounts of water were rapidly transferred to the debris flow initiation zone and torrent beneath. Retrogressive linear erosion visible in the initiation zone indicates that concentrated water flow emerged at the main scarp (Fig. 4b), in good agreement with earlier observations of rock glacier front failures (Kummert et al., 2018; Marcer et al., 2020; Kofler et al., 2021). Since then, the depression never filled again but still shows distinct morphology." replaced by:

"During the same night as the debris flow initiation, a thermokarst lake on top of Hüttekar Rock Glacier started draining ∼350 m behind the debris flow initiation zone and emptied almost completely during the following day, as indicated by local observations (Fig. 3, Fig. 4e, Fig. 4f, Supplementary Video). The thermokarst lake had started to develop coincidently with the sudden onset of intense snowmelt in early June 2019 within a shallow depression where massive ice within the rock glacier prevented drainage (Fig. 4g). In the stage of its largest extent, the thermokarst lake was approximately 300 m long, up to about 150 m wide and 4–5 m deep, comprising an estimated water volume of ∼150,000 $m^3$.

Effective drainage occurred through a large crevasse (width ∼1.5 m, height ∼2 m) that formed in association with the debris flow initiation (Fig. 4h). The crevasse was part of a newly formed channel system connecting the thermokarst lake to that part of the rock glacier front that failed, constituting the debris flow initiation zone. While the exact vertical position of this channel system is not known, clearly discernible and precisely confined collapse structures indicate the trace of this channel network along the rock glacier surface (Supplementary Fig. S1 and S3). Large amounts of water were rapidly transferred to the debris flow initiation zone where

retrogressive linear erosion indicates that concentrated water flow emerged at the main scarp (Fig. 4b). During the following months, erosion of loose sediment in the vicinity of the debris flow initiation zone persistently modified its shape and continues to widen it until today. The depression storing the former thermokarst lake never filled again but still shows distinct morphology (Supplementary Fig. S1 and S3).

During the decades preceding the debris flow event, thermokarst lakes had never been observed before on Hüttekar Rock Glacier, despite frequent visits by hikers, hunters, shepherds, and staff of Hohenzollernhaus (Josef Waldner, pers. comm.). In line with these observations, publicly available remote sensing data and historical maps of Hüttekar-cirque do not exhibit any indications of thermokarst lake development on Hüttekar Rock Glacier before June 2019 (evident from Sentinel-2 satellite imagery provided by Copernicus and processed by Sentinel Hub (https://www.sentinel-hub.com/), historical ortho-images and laserscans provided by the Government of the Province of Tyrol, (https://www.data.gv.at/; https://lba.tirol.gv.at/), and historical maps provided by the Government of the Province of Tyrol (https://hik.tirol.gv.at))."

**Page 7, line 136:** "under consideration" deleted.

**Page 7, line 137:** redundant reference deleted.

**Page 7, line 139-141:** "(1) predisposing factors (inducing a static setting capable of enhancing the destabilizing impact of dynamic forces), (2) preparatory factors (dynamic forces shifting the slope towards a state susceptible to failure), and (3) triggering factors (dynamic forces initiating failure)." replaced by "(1) predisposing factors (inducing a static setting that enhances the destabilizing impact of dynamic factors), (2) preparatory factors (dynamic factors shifting the slope towards a state susceptible to failure), and (3) triggering factors (dynamic factors initiating failure)."

**Page 7, line 142-144:** "Acknowledging that debris flow hazard analysis requires consideration of multiple destabilizing factors, their impact on slope stability is characterized individually to establish a basis for reconstructing the failure mechanism. The chosen methods aim at maximizing comparability to earlier studies on rock glacier front failures and debris flows in periglacial regions." replaced by "We evaluate the impact of individual factors on slope stability and identify critical factor combinations to reconstruct the failure mechanism. The chosen methods aim at maximizing comparability to earlier studies on debris flow initiation at active rock glacier fronts. All analyzed data and employed software are freely available, the respective links are provided in the 'Data availability' section."

**Page 8, figure 4:** The following error occurring in the original manuscript was corrected in the revised manuscript: "Photographs were taken on 14 August 2019 (c–h) and 26 August 2019 (a–b), respectively, by Roman Außerlechner, Thomas Figl, Werner Thöny, and Josef Waldner." correctly reads "Photographs were taken on 13 August 2019 (g–h) by Josef Waldner, as well as on 14 August 2019 (c–f) and 26 August 2019 (a–b), respectively, by Roman Außerlechner, Thomas Figl, Werner Thöny."

**Page 9, line 146-153:** "A topographical setting promoting rock glacier destabilization is a necessary (but not sufficient) precondition for rock glacier front failure. Attributes significantly affecting slope stability include morphometric parameters such as length, slope angle, and curvature (Chowdhury et al., 2010; Reichenbach et al., 2018; Marcer et al., 2019). Sediment erosion, transport, and deposition depend on local energy gradients, available transport vectors, and material properties (Bracken et al., 2015). Collectively, these variables control local loading stresses and the availability of material for subsequent mobilization (Kummert et al., 2018; Kofler et al., 2021). The respective impact of these factors on slope stability is evaluated using a high-resolution (1 × 1 m) DTM based on airborne laserscanning data acquired two years prior to debris flow occurrence (Government of the Province of Tyrol, 2021a), applying SAGA GIS 2.1.4 for morphometric analyses (Conrad et al., 2015)." replaced by "An unfavorable topographical setting is a necessary precondition for rock glacier front failure. The impact of morphometric factors including slope angle, length, and curvature is evaluated using the high-resolution DTM mentioned above (Government of the Province of Tyrol, 2021a). Morphometric analyses are conducted applying SAGA GIS 2.1.4 (Conrad et al., 2015)."

**Page 9, line 156-157:** "Thermal ground conditions and the presence of subsurface ice affect the shear strength of involved materials, alter groundwater flow patterns, and impact the storage and release of water by thermokarst evolution." deleted.

**Page 9, line 161:** redundant references deleted.

**Page 9, line 162:** redundant references deleted.

**Page 9, line 165-172:** "The mechanics of debris flows are largely determined by solid-fluid interactions, rendering grain size distribution and water content major variables driving flow behavior (Iverson, 1997). The rate of water supply is determined by the permeability of sediments above the initiation zone. Geotechnical and hydrogeological characteristics of the materials involved are inspected by analyzing the grain size distribution of the rock glacier surface layer at one coarse-grained and three relatively fine-grained domains (sampling locations indicated in Fig. 3). At each location the maximum diameter of 200 clasts lying side by side is measured in an area of approximately 2 × 2 m (fine grained domains) to 5 × 5 m (coarse grained domain). The grain size distribution of the poorly sorted sediment layer below the blocky surface is analyzed by manual wet sieving of a sample taken at the steep rock glacier front." replaced by "The grain size distribution of the rock glacier front material is inspected by wet sieving of four samples taken in 2009, 50 m southwest of the later debris flow initiation zone (position indicated in Fig. 3). Due to the heterogeneous structure of the rock glacier, these samples are not representative for the grain size composition of the entire rock glacier. However, considering the proximity of the sampling locations to the subsequent debris flow initiation zone, and the dangerous sampling conditions at active rock glacier fronts in general, we consider these samples reasonable approximations to the grain size composition of the debris flow source material."

**Page 9, line 174-175:** "The geometry of the active rock glacier front depends on the dynamic balance between rock glacier kinematics and erosion rates (Kummert and Delaloye, 2018)." deleted.

**Page 9-10, line 177-180:** "ortho-images provided by the Government of the Province of Tyrol (2021b)). In the observation period 1970–2020, 200 prominent blocks, geometrically well distributed on the rock glacier surface, are visually identified at each ortho-image epoch to approximate the horizontal surface displacement rate at the respective location." replaced by "The analyzed ortho-images exhibit a spatial resolution of 20 cm and are provided as Supplementary Fig. S3 (Government of the Province of Tyrol, 2021b). In the observation period 1970–2020, 200 prominent blocks, geometrically well distributed on the rock glacier surface, are visually identified at every ortho-image epoch to approximate the horizontal surface displacement rates at the respective location within single periods (1970-2003-2007-2010-2020, Avian et al. (2009); Kummert and Delaloye (2018))."

**Page 10, line 181-182:** "Both analyses provide the basis for assessing the kinetic patterns on the rock glacier surface (Avian et al., 2009; Kummert and Delaloye, 2018)." replaced by "Visual detectability within optical imagery is approximately half of a pixel size (in this case 10 cm). Assessed surface displacement rates cover at least 3 years, thus minimum rates of 3–4 cm are detectable."

**Page 10, line 183-185:** "Climatic factors and their progressive change exert a key control on slope stability (Gariano and Guzzetti, 2016; Patton et al., 2019). Alterations in volume and intensity of rainfall, snowmelt and ice melt affect subsurface water content, pore-water pressure and seepage forces (Ikeda et al., 2008; Cicoira et al., 2019; Patton et al., 2019)." deleted.

**Page 10, line 187:** "The dataset is continuously updated by GeoSphere Austria." added.

**Page 10, line 188-194:** "Specific aspects of the local climate are explored by calculating the annual positive degree day sum (daily mean air temperature > 0 °C) as proxy for available melting energy, as well as precipitation due to very wet days (> 95th percentile) reflecting the annual magnitude of heavy precipitation events (Klein Tank et al., 2009; Cuffey and Paterson, 2010). Considering that the hydrometeorological conditions preceding the debris flow determine the critical amount of water necessary to initiate failure (Crozier, 2010), the impact of these climate indices is assessed by comparing their respective values during summer 2019 (01 June–13 August) to previous years (1976–2019)." replaced by "The averaging domain corresponds to the Hüttekar Rock Glacier catchment as specified by Wagner et al. (2020). We calculate the annual positive degree day sum (daily mean air temperature > 0 °C) as a proxy for available melting energy, and the precipitation due to very wet days (> 95th percentile) which reflects the annual magnitude of heavy precipitation events (Klein Tank et al., 2009; Cuffey and Paterson, 2010). The impact of these climate indices is assessed by comparing their respective values during summer 2019 (01 June–13 August (failure date)) to previous years (1976–2019)."

**Page 10, line 195:** redundant references deleted.

**Page 10, line 198-206:** "In the Alps, individual storms exhibit strong intensity gradients at length scales < 5 km (i. e. below the basic length-to-crest scale; Haiden et al. (2011); Nikolopoulos et al. (2014, 2015a); Marra et al. (2016); Destro et al. (2017)). The combination of weather station data and remote sensing information allows the detection of individual event characteristics (Haiden et al., 2011; Borga et al., 2014; Marra et al., 2014, 2016; Destro et al., 2017). Detailed (~hourly) temporal resolution of precipitation data is necessary to avoid biased estimates of rainfall intensity and duration (Marra, 2019). Meteorological analyses at time intervals from hours

to months are thus based on the INCA system (Integrated Nowcasting through Comprehensive Analysis; Haiden et al. (2011)), providing gridded (1 × 1 km) data sets at hourly temporal resolution. Inclusion of 12 weather stations at a distance < 25 km from Hüttekar-cirque, in combination with C-band radar measurements and Meteosat Second Generation Satellite Products, allows assessing the spatial and temporal patterns of meteorological conditions at the study site." deleted (please see also changes on page 11, line 238).

**Page 10-11, line 207-217:** "Snow cover development is assessed using the spatially distributed, physically based snow cover model SNOWGRID (Olefs et al., 2013). It employs a simple two-layer scheme, considering settling, the heat and liquid water content of the snow cover, snowline depression effects and the energy added by rain. For every time step and layer, the state variables snow density, snow water equivalent, snow temperature, liquid water content, bottom liquid water flux, and surface albedo are calculated. The primary focus of the model is to obtain a high-resolution representation of their spatial distribution and to provide fast calculations on a large grid. The model employs a 100 × 100 m bilinear interpolation of the INCA data set in combination with schemes for radiation and cloudiness developed at the Central Institute for Meteorology and Geodynamics (Austria). These are based on ground measurements, satellite products, and high quality solar and terrestrial radiation data (Olefs et al., 2013, 2016). For each winter season, the cumulated snowmelt volume in Hüttekar-cirque is calculated and compared to the 2018/19 season. The rate of snowmelt is approximated by the respective seasonal average during the time span between maximal snow volume and complete ablation of the winter snow cover (Kofler et al., 2021)." replaced by "For each winter season, the cumulated snowmelt volume in Hüttekar-cirque is calculated and compared to the 2018/19 season. The rate of snowmelt is approximated by the respective seasonal average during the time span between maximal snow volume and complete ablation of the winter snow cover, in analogy to Kofler et al. (2021). Snow cover development is assessed using the spatially distributed, physically based snow cover model SNOWGRID (Olefs et al., 2013). The model employs a simple two-layer scheme, considering settling, the heat and liquid water content of the snow cover, snowline depression effects and the energy added by rain. Snow density, snow water equivalent, snow temperature, liquid water content, bottom liquid water flux, and surface albedo are calculated at hourly resolution. The model employs a 100 × 100 m bilinear interpolation of the INCA data set (Integrated Nowcasting through Comprehensive Analysis; Haiden et al. (2011)) in combination with schemes for radiation and cloudiness based on ground measurements, satellite products, and high quality solar and terrestrial radiation data (Olefs et al., 2013, 2016). The resulting snow cover data are routinely evaluated and validated by GeoSphere Austria using more than 50 station measurements of snow depth, additional measurements regarding snow depth and snow water equivalent (5 stations along with more than 200 individual measurements), snow depth measurements employing laser sensors, winter mass balance measurements of glaciers within the model domain, spatial validation of snow cover extent using satellite-based fractional snow cover area provided by MODIS, as well as cumulative runoff data (Olefs et al., 2020)."

**Page 11, line 218-220:** "Glacial meltwater infiltrates directly from Glockturmferner and indirectly from Hüttekarferner into the rock glacier (Fig. 3), thus the total volume and average rate of ice melt is calculated for each year between ablation of the snow cover and 13 August, based on the surface energy balance for the respective glacier (Hock, 2005; Cuffey and Paterson, 2010)." replaced by "The total volume and average rate of ice melt are calculated for each year between ablation of the snow cover and 13 August (failure date), based on the surface energy balance for the respective glacier (Hock, 2005; Cuffey and Paterson, 2010)."

**Page 11, line 230-235:** "Rainfall-induced debris flows are triggered by infiltration rates exceeding subsurface drainage rates and the corresponding alteration of pore-water pressures (Sidle, 1984; Johnson and Sitar, 1990; Anderson and Sitar, 1995; Wieczorek, 1996; Wieczorek and Glade, 2005; Crozier, 2010). Infiltrating water adversely affects slope stability by adding additional weight while simultaneously decreasing shear strength, either by lowering effective stresses in response to increasing positive pore-water pressures or by eliminating suction due to declining negative pore-water pressures (Johnson and Sitar, 1990; Anderson and Sitar, 1995; Chowdhury et al., 2010; Crozier, 2010)." deleted.

**Page 11, line 235:** "storm" replaced by "rainfall event."

**Page 11, line 236:** "at hourly resolution" added.

**Page 11, line 238:** "Inclusion of 12 weather stations at a distance < 25 km from Hüttekar-cirque, in combination with C-band radar measurements and Meteosat Second Generation Satellite Products, allows reliable identification of event characteristics at the study site." added (please see also changes on page 10, line 198-206).

**Page 11, line 242:** "earlier events hitting Hüttekar-cirque as well as to" added.

**Page 11-12, line 243-252:** "Glacial lake outbursts evolving into debris flows are commonly initiated by mechanical failure of the dam or by rapid expansion of the lake drainage system through thermal erosion of ground ice. The volume and geometry of the impounded reservoir, dam characteristics, failure mechanism, downstream topography and sediment availability control the hazard potential (Clague and Evans, 1994). The development of thermokarst features, including meltwater lakes and channels, is highly sensitive to thermal ground conditions and their response to climate change (Kääb and Haeberli, 2001). The spatiotemporal evolution of the thermokarst lake on Hüttekar Rock Glacier is deduced by combining Sentinel-2 multi-spectral satellite data and the latest (2017/18) available DTM, characterizing the lake surface area dynamics and corresponding water volume development (modified Copernicus data 2020 processed by Sentinel Hub; Sentinel Hub (2020); Government of the Province of Tyrol (2021a)). Due to the dynamic nature of the rock glacier surface, volume estimates are considered rough estimates, failing to account for water stored in the pore space and potential intra-permafrost channels." deleted (please see also changes on page 12, line 264).

**Page 12, line 252:** "within the rock glacier" added.

**Page 12, line 254-255:** redundant references deleted.

**Page 12, line 258:** "meltwater" replaced by "thermokarst"

**Page 12, line 259:** "provides" replaced by "represents".

**Page 12, line 264:** "The spatiotemporal evolution of the thermokarst lake on Hüttekar Rock Glacier is deduced by combining Sentinel-2 multi-spectral satellite data and the latest (2018) available DTM, characterizing the lake surface area dynamics and corresponding water volume development (modified Copernicus data 2020 processed by Sentinel Hub (2020); Government of the Province of Tyrol (2021a)). Due to the dynamic nature of the rock glacier surface, volume estimates are considered rough estimates, failing to account for water stored in the pore space and potential intra-permafrost channels." added (please see also changes on page 11-12, line 243-252).

**Page 12, page 266-267:** "A systematic evaluation of destabilizing factors and adverse combinations thereof requires an individual evaluation of every potentially contributing factor." deleted.

**Page 12, page 272-277:** "The channel below the rock glacier front was less steep and concave, while the debris fan was already comparatively flat before the 2019 debris flow (Table 2). Irregular micromorphology, depositional levees, and boulder trains document earlier debris flows (Fig. 2, Fig. 5; Soeters and van Westen (1996); Pack (2005); Hungr 275 et al. (2014)). Material eroded at the rock glacier front is available for subsequent mobilization further downslope, inhibiting the formation of a stabilizing debris accumulation at the toe of the rock glacier front (Kummert and Delaloye, 2018; Kummert et al., 2018; Kofler et al., 2021)." Replaced by "It formed a part of the 400 m wide and 100 m high rock glacier front, bounded to the top by a distinct erosional edge separating it from the flat rock glacier surface. The steep rock glacier front (35° on average) exhibited roughly homogeneous appearance, except for two bedrock outcrops in its southern part. A direct comparison of this area before and after the debris flow event is provided in Supplementary Fig. S4." The new Supplementary Figure is provided in the revised 'Supplementary Material'.

**Page 13, Figure 5:** In the figure caption, the phrase "acquired in the period 2017/2018" is replaced by "one year before debris flow occurrence". The term "color map" is replaced by the term "color ramp".

**Page 14, Table 2:** In the table caption, the phrase "acquired in the period 2017/2018" is replaced by "one year before debris flow occurrence".

**Page 14, line 278-283:** "Assessment of thermal ground conditions indicates that the debris flow initiation site is subjected to a high energy environment. Potential incoming solar radiation reaches 2,135 kWh m$^{-2}$ a$^{-1}$ (Fig. 5c). The Alpine Permafrost Index Map indicates that the initiation zone is located at the lower permafrost boundary, with permafrost preserved only in very favorable conditions (Fig. 5d). Collectively, the convex morphology and exposed position, strong potential radiation and unfavorable permafrost index are conducive to an advanced stage of permafrost degradation, increased water contents, and large amounts of unfrozen, loose sediment susceptible to mobilization (Marcer et al., 2019)." deleted (please see also changes on

**Page 14, line 284-290:** "Grain size analyses indicate that the blocky surface layer of Hüttekar Rock Glacier is composed of poorly sorted, coarse-grained sediment. The finer-grained parts, measured on three locations (1–3; positions given in Fig. 3), show an average grain size of 11.6, 12.8, and 16.0 cm, respectively. The coarse-grained site (4) exhibits an average grain size of 65.6 cm, documenting the heterogeneous structure of the blocky surface layer. Figure 6a displays the respective grain-size distributions. Due to the subordinate presence of

components smaller than gravel (largely matrix-free surface layer), the rock glacier classifies as bouldery rock glacier sensu Ikeda and Matsuoka (2006). These characteristics indicate a permeable surface layer exhibiting high infiltration capacity and low hydraulic resistance to water flowing along the permafrost table." deleted.

**Page 14, line 291-292:** "Sieve analyses of samples 5–8 taken at the rock glacier front (Fig. 6b, positions indicated in Fig. 3) show that gravel (64 %) and sand (25 %) are the dominating grain sizes, while the amount of clay and silt is very low (< 1 %)." replaced by "Sieve analyses of the rock glacier front material (Fig. 6) show that gravel (64 %) and sand (25 %) are the dominating grain sizes, while the amount of clay and silt is very low (< 1 %). Detailed results are provided in Supplementary Table S1."

**Page 14, line 296-297:** "(dashed lines in Fig. 6b)" moved to the end of the sentence.

**Page 14, line 297-301:** "Provided sufficient water is available to keep the rock glacier debris saturated, these characteristics promote the development of undrained loading conditions, high sensitivity of effective stresses to sediment compaction and perpetually high excess pore-water pressures in response to deformation (Major, 1996; Iverson, 1997; Major et al., 1997). These features document the compositional propensity of the rock glacier front to debris flow mobilization." deleted.

**Page 14, line 301:** "The 80 m wide and 200 m long debris flow initiation zone eroded into the rock glacier front exposes its internal structure. The irregular, concave niche exhibits steep flanks that are up to 30 m high and composed of loose, poorly-sorted sediment (Fig. 4a and 4b; Supplementary Fig. S1–S4). Its top represents the terminus of a collapse structure network connecting it to the former position of the thermokarst lake (Supplementary Fig. S1, S3, and S4), with linear erosion features indicating concentrated emergence of water. After the debris flow initiation, frozen material was exposed immediately below the top (Fig. 4a and 4b; Supplementary Fig. S2). Assessment of thermal ground conditions indicates that the debris flow initiation zone is subjected to a high energy environment. Potential incoming solar radiation reaches 2,135 kWh m$^{-2}$ a$^{-1}$ (Fig. 5c). The Alpine Permafrost Index Map indicates that the initiation zone is located at the lower permafrost boundary, with permafrost preserved only in very favorable conditions (Fig. 5d).

The channel below the rock glacier front is less steep and concave (Table 2). The debris fan was already comparatively flat before the 2019 debris flow (Table 2). Irregular micromorphology, depositional levees, and boulder trains document earlier debris flows (Fig. 2, Fig. 5; Supplementary Fig. S2; Soeters and van Westen (1996); Pack (2005); Hungr et al. (2014)). Collectively, the convex morphology and exposed position, strong potential radiation and unfavorable permafrost index are conducive to an advanced stage of permafrost degradation, increased water contents, and large amounts of unfrozen, loose sediment susceptible to mobilization (Marcer et al., 2019)." added (please see also changes on page 12, page 272-277).

**Page 15, Figure 6:** Figure revised. In the figure caption, the phrase "Grain-size distributions of Hüttekar Rock Glacier surface layer (a) and front material (b). The corresponding sampling locations are indicated in Fig. 3." is replaced by "Grain-size distributions of material composing the rock glacier front. The corresponding sampling locations are 50 m southwest of the debris flow initiation zone, as indicated in Fig. 3."

**Page 15-16, line 303-310:** "Analysis of multi-temporal ortho-images indicates constant, moderate surface displacement rates. The kinematics of the rock glacier surface show mean annual surface displacement rates of the 200 prominent blocks ranging from 1–50 cm a$^{-1}$ (mean: 14 cm a$^{-1}$) in the observation period of 1970–2020. The position of the rock glacier front fluctuates non-directionally (amplitude 3–7 m), indicating that steady erosion of the front approximately balances rock glacier movement, thus ensuring an invariant long-term front geometry (Kummert and Delaloye, 2018). While destabilization of rock glaciers is frequently characterized by accelerating surface deformation rates and the development of surface discontinuities such as cracks and crevasses, the absence of these features on Hüttekar Rock Glacier does not point towards general destabilization of the rock glacier before the 2019 debris flow." replaced by "Analysis of multi-temporal ortho-images indicates roughly constant, moderate surface displacement rates (Supplementary Fig. S5). The kinematics of the rock glacier surface show mean annual displacement rates of the 200 prominent blocks ranging from 1–50 cm a$^{-1}$ (mean: 15 cm a$^{-1}$) in the observation period of 1970–2020. Data for individual blocks and summary statistics are provided in Supplementary Table S2. The position of the rock glacier front fluctuates non-directionally (amplitude 3–7 m), indicating that steady erosion of the front approximately balances rock glacier movement. The absence of surface discontinuities and the constant surface displacement rates do not indicate general destabilization of the rock glacier before the 2019 debris flow."

**Page 16, line 311-312:** "Fig. S2; Government of the Province of Tyrol (2021b)" replaced by "Supplementary Fig. S3". The updated Supplementary Figure is provided in the revised 'Supplementary Material'.

**Page 16, line 313-314:** "Analyzing the long-term (1976–2019) climate signal highlights the distinct hydrometeorological characteristics of the months preceding the failure and the respective long-term trends in Hüttekar-cirque (Table 3)." replaced by "The months preceding the debris flow were characterized by warm and dry conditions, on top of an exceptionally strong longterm trend of increasing air temperature in Hüttekar-cirque (Table 3)."

**Page 16, line 318:** "record" deleted.

**Page 16, line 319:** "(±0.01)" and "(±0.02)" deleted.

**Page 16, line 320:** "increased" replaced by "increases"

**Page 16, line 321-322:** "Recapitulating, the months preceding the debris flow are characterized by warm and dry conditions, on top of an exceptionally strong long-term trend of increasing air temperature in Hüttekar-cirque." deleted.

**Page 16, line 323-330:** "indicating moderately wet conditions anteceding the debris flow. Intense snowmelt and ice melt (11.2 and 2.7 mm d$^{-1}$, respectively) reflect the high energy available in 2019, in agreement with the high positive degree day sum reported above. The winter season preceding the debris flow is characterized by a prominent snow cover, almost monotonically gaining volume until 29 May 2019, roughly restricting meltwater production to June. The long-lasting snow cover and rapid snowmelt (Fig. 7b) is in excellent agreement with observations throughout Tyrol (Hydrological Service of Tyrol, 2019). Comparing total volumes indicates inconspicuous overall conditions in 2019 (Fig. 7c), acknowledging that the relatively short record precludes statistically substantiated conclusions." replaced by:

"The winter season preceding the debris flow was characterized by a prominent snow cover, almost monotonically gaining volume until 305 29 May 2019. The late onset of snowmelt is responsible for the unusually high peak, followed by high solar exposure and extraordinarily warm air temperatures driving rapid ablation in June. The high rates of meltwater production caused widespread flooding in Tyrol, despite the fact that June 2019 was a particularly dry month (Hydrological Service of Tyrol, 2019; Hübl and Beck, 2020). Intense snowmelt and ice melt rates in Hüttekar-cirque reflect the strong atmospheric energy input in 2019 (11.2 and 2.7 mm d$^{-1}$, respectively; Fig. 7b). In contrast to the high intensity of the melting processes, comparing total volumes of snowmelt, ice melt and rainfall indicates inconspicuous overall conditions in 2019 (Fig. 7c). As a first-order approximation, we compare cumulative volumes between 1 January 2019 and 13 August 2019 (failure date) to earlier years (consistently totalized from 1 January to 13 August), acknowledging that the relatively short record precludes statistically substantiated conclusions. Compared to the four years before 2019 (for which continuous SNOWGRID data are available), the total snow melt volume exceeds the 5-year-average by 19 %, while both the total ice melt volume and rainfall volume are slightly below this average (-3 % and -2 %, respectively). The rapid snowmelt efficiently eliminated the snow cover in Hüttekar-cirque, so that snowmelt ceased on 22 July 2019, i.e. 22 days before the debris flow initiated.

The combination of large amounts of snow available for melting and abnormally warm and dry weather conditions resulted in rapid and extensive meltwater production, thus favoring the ponding of meltwater on the surface of Hüttekar Rock Glacier. The continuously high atmospheric energy input promoted the subsequent development of the thermokarst lake from 3 June 2019 to 13 August 2019 described in Sect. 5.3. Summarizing, major hydrometeorological factors distinguishing 2019 from earlier years are the rapid and late snowmelt, the high energy environment during the summer months, and the storage of water in the newly formed thermokarst lake."

**Page 17, Figure 7:** "(linear approximation)" deleted in the figure caption.

**Page 17, line 332:** "∼" removed.

**Page 17, line 334:** "does not exceed" replaced by "did not cross", "during the summer season" deleted.

**Page 18, Figure 8:** In the figure caption "Rainfall event frequency analysis with respect to (a) rainfall intensity-duration and (b) rainfall volume-duration based on INCA (Haiden et al., 2011). The rainfall event preceding the Hüttekar debris flow (red dot) was neither especially intense nor especially persistent with respect to earlier rainfall events hitting Hüttekar-cirque (grey dots), critical rainfall thresholds (black lines) and rainfall events triggering documented rock glacier front failures in the European Alps (black dots) (note logarithmic scale). Regional thresholds describing minimum conditions for rainfall-induced debris flows are based on Nikolopoulos et al. (2015b) and Marra et al. (2016)." is replaced by "Rainfall event frequency analysis with respect to (a) rainfall intensity I vs. rainfall event duration D and (b) rainfall event volume E vs. rainfall event duration D based on INCA

(note logarithmic scale). Regional thresholds describing minimum conditions for rainfallinduced debris flows are based on Nikolopoulos et al. (2015) and Marra et al. (2016), with equations given below the corresponding lines (thresholds are representative for the summer season)."

**Page 18, line 337:** "rock glacier front failures" replaced by "debris flows from other rock glacier fronts"

**Page 18, line 340:** "In contrast, the rainfall event preceding the debris flow at Hüttekar Rock Glacier was neither especially intense, nor especially persistent." added.

**Page 18, line 342:** "was" deleted, "before the debris flow" added.

**Page 18, line 344-346:** "The thermokarst lake evolved over a period of 10 weeks, but drained within hours through a newly formed channel, indicating that the characteristic time scale of thermokarst breakthrough is shorter than the summer season (Fig. 4e-h, Fig. S2)." deleted (please see also changes on page 19, line 349).

**Page 18, line 346:** "Comparing snow cover and lake development shows that the latter" replaced by "The lake".

**Page 18, line 347:** "Fig. 8a" replaced by "Fig. 7a".

**Page 19, line 348-349:** "While the area gain slowed down at the beginning of July, the energy input was considerable thereafter, allowing for extensive ice melt beneath the surface." replaced by "While the lake surface area stayed constant after the beginning of July, the energy input was considerable thereafter, suggesting that large amounts were transferred to the subsurface, allowing for extensive ice melt beneath the surface."

**Page 19, line 349:** "The thermokarst lake evolved over a period of 10 weeks and drained within one day through a newly formed channel, indicating that the time scale of thermokarst breakthrough is shorter than the summer season." added (please see also changes on page 18, line 344-346).

**Page 19, Table 4:** In the table caption, "2017" is replaced by "2018".

**Page 19, line 353:** "is" replaced by "was".

**Page 19, line 356:** "∼" replaced by "a few".

**Page 19, line 356-358:** "In contrast to many hazardous glacial lakes evolving over time scales of years to decades (Haeberli, 1983; Mölg et al., 2021), the thermokarst lake in Hüttekar-cirque unfolded its destructive power already two months after its formation." deleted (please see also changes on page

**Page 19, line 361-362:** "While the absence of instrumentation in Hüttekar-cirque during the failure inhibits a detailed reconstruction of the debris flow initiation mechanism, the relative importance of destabilizing factors can confidently be evaluated." deleted.

**Page 20, line 363-364:** redundant references deleted.

**Page 20, line 364:** "Convex topography generally" replaced by "The convex top"

**Page 20, line 365:** redundant references deleted.

**Page 20, line 365-368:** "Since the rock glacier did not show discernible signs of destabilization prior to debris flow initiation, external drivers including the strong increase in air temperature and positive degree day sum, along with available snow and ice meltwater and their pronounced impact on permafrost during the months preceding the debris flow event are considered crucial preparatory factors (Table 3)." deleted.

**Page 20, line 369-371:** "Rock glacier creep balancing erosion rates in combination with a high energy environment favoring permafrost degradation at the rock glacier front provide a large accumulation of loose, unfrozen sediment that is susceptible to mobilization down the adjacent steep slope (Fig. 5, Fig. S2)." replaced by "Rock glacier creep balancing erosion rates at the rock glacier front, in combination with a high energy environment favoring permafrost degradation, provides a large accumulation of loose, unfrozen sediment. The eroded material is available for subsequent mobilization down the adjacent steep slope, inhibiting the formation of a stabilizing debris accumulation at the toe of the rock glacier front (Kummert and Delaloye, 2018)."

**Page 20, line 373:** "drives" replaced by "drive".

**Page 20, line 374-375:** redundant references deleted.

**Page 20, line 377:** redundant references deleted.

**Page 20, line 377-384:** "Poorly sorted sand and gravel, such as the Hüttekar Rock Glacier material (Fig. 6b), are susceptible to undrained deformation and attendant pore-water pressure coevolution, demonstrating the susceptibility of the rock glacier to debris flow mobilization (Iverson, 1997, 2005; Iverson 380 et al., 1997)." replaced by: "Once saturated, poorly sorted sand and gravel such as the Hüttekar Rock Glacier material (Fig. 6) is susceptible to undrained deformation and attendant pore-water pressure coevolution. Since these features

document the compositional propensity of the rock glacier front to debris flow mobilization, the remaining issues requiring clarification concern potential sources of water capable of saturating the rock glacier front material at the initiation zone."

**Page 20, line 381:** redundant reference deleted.

**Page 20, line 383:** "Fig. 6b" replaced by "Fig. 6".

**Page 20, line 385:** "The hydrometeorological conditions preceding the debris flow determine the critical amount of water necessary to initiate failure (Crozier, 2010)." added.

**Page 20, line 387:** "(Table 3)" deleted.

**Page 20, line 388:** "fails to" replaced by "could not"; "and sustain" deleted; "for several hours" deleted.

**Page 20, line 394:** "Fig. 4e, Fig. 4f, Fig. 4h, Fig. S2" replaced by "Fig. 4e, 4f, and 4h; Supplementary Fig. S1, S3, and S4".

**Page 21, line 398:** "provided" replaced by "had".

**Page 21, line 401-402:** "Consequently, rapid drainage emptied the lake within hours, without apparent prior indications of a lake outburst." deleted (please see also changes on page 21, line 403-411).

**Page 21, line 403-411:** "With respect to GLOF hazard evaluation, the decisive factor is that thermokarst development and breakthrough happened within an extremely short time scale (on the order of weeks), driven by the high energy input during summer 2019. In terms of mechanical properties, the ice-debris mixture composing the rock glacier represents a transitional form between the common glacier dammed and moraine dammed lakes (Clague and Evans, 1994; Huggel et al., 2004; Schaub, 2015). The Hüttekar Rock Glacier failure mechanism involved drainage channel enlargement as well as collapse of the rock glacier front, demonstrating that thermokarst development and slope failure operated synergistically. Since their contrasting time scales (~weeks for channel enlargement, ~seconds to hours for rock glacier front collapse) are commonly associated with different materials (ice and debris, respectively), the integration of both mechanisms distinguishes rock glaciers from glaciers and moraines in terms of GLOF hazard assessment." replaced by "In contrast to many hazardous glacial lakes evolving over time scales of years to decades (Haeberli, 1983; Mölg et al., 2021),

the thermokarst lake in Hüttekar-cirque unfolded its destructive power already 10 weeks after its formation. Rapid drainage emptied it without apparent prior indications of an outburst flood. The decisive factor in terms of hazard assessment is that thermokarst development and channel breakthrough happened within an extremely short time scale (on the order of weeks), driven by the high energy input during summer 2019. The failure mechanism involved drainage channel enlargement as well as collapse of the rock glacier front, demonstrating that thermokarst development and slope failure operated synergistically."

**Page 21, line 416:** "including overtopping, piping and mechanical collapse" deleted.

**Page 21, line 417:** "and reduction" added.

**Page 21, line 418:** "This issue will likely gain importance as climate change alters the conditions in prospective debris flow initation zones. With respect to thermokarst evolution, the key issue is the expected increase in available melting energy that is closely linked to rising air temperatures. Previous research showed that the development of thermokarst features, including lakes and channel networks, is highly sensitive to thermal ground conditions and their response to climate change (Kääb and Haeberli, 2001). Consequently, we expect the establishment of these features in areas currently characterized by permafrost. Our study shows that such channel systems might develop within weeks, and subsequently are able to transfer large amounts of water within a short time. It is this hardly predictable short-term development of thermokarst features that challenges the evaluation of slope stability under a warming mountain climate. Once established, these channel networks are capable of rapidly concentrating water at points of converging channels. This process threatens slope stability irrespective of the specific trigger mechanisms, by adding additional weight while simultaneously decreasing shear strength (Johnson and Sitar, 1990; Chowdhury et al., 2010). Its hazardous role is underpinned by concentrated water outflow in debris flow initiation zones documented at several other rock glacier fronts (Kummert et al., 2018; Marcer et al., 2020; Kofler et al., 2021). While this study is the first one to explicitly assess its rate and magnitude in this specific context, we argue that similar processes might support localized failure mechanisms of active rock glacier fronts in a wide range of settings." added.

**Page 21, line 421:** "The" replaced by "Local".

**Page 21, line 422:** "sediment" added.

**Page 21, line 423:** "multi" deleted.

**Page 21, line 424:** "a" replaced by "this".

**Page 21-22, line 426-430:** "In combination with challenges regarding hazard detection and prediction in remote areas, this complicates integrated multi hazard assessment (Kappes et al., 2012; Gallina et al., 2016). In this context, the combination of several raster data sets comprising terrain models, satellite imagery, and gridded climate and weather variables proofed a valuable tool for assessing the individual impact of destabilizing factors in complex mountainous terrain." deleted (please see also changes on page 22, line 440).

**Page 22, line 431-432:** "Not only the amount of rainfall, but its combination with the" replaced by "Rapid".

**Page 22, line 432:** "primarily" added.

**Page 22, line 434:** "two months" replaced by "10 weeks".

**Page 22, line 439:** "multi" deleted.

**Page 22, line 440:** "In combination with challenges regarding hazard detection and prediction in remote areas, this complicates integrated multi hazard assessment (Kappes et al., 2012; Gallina et al., 2016). In this context, the combination of several raster data sets comprising terrain models, satellite imagery, and gridded climate and weather variables proofed a valuable tool for assessing the individual impact of destabilizing factors in complex mountainous terrain." added (please see also changes on page 21-22, line 426-430).

**Page 22, line 444:** "gains" replaced by "will likely gain".

**Page 22, line 447:** "an expedient tool" replaced by "a benchmark".

**Page 22, line 448-452:** "Input data for this study are freely available. Meteorological data sets (SPARTACUS, INCA) are provided by the Central Institute for Meteorology and Geodynamics (https://data.hub.zamg.ac.at/, last access: 28 June 2022). The Government of the Province of Tyrol provides the digital terrain model as well as current and historical ortho-images (https://www.data.gv.at/, last access: 28 June 2022). Sentinel-2 multi-spectral satellite data are provided by Copernicus and processed by Sentinel Hub (https://www.sentinel-hub.com/, last access: 28 June 2022)." replaced by "Meteorological data sets, including the Spatiotemporal Reanalysis Dataset for Climate in Austria (SPARTACUS) and the Integrated Nowcasting through Comprehensive Analysis Dataset (INCA), are continuously updated by GeoSphere Austria and freely available from https://data.hub.zamg.ac.at/ (last access: 15 February 2023). The Government of the Province of Tyrol provides the digital terrain model, current and historical ortho-images, as well as historical maps and laserscans (freely available from https://www.data.gv.at/, https://hik.tirol.gv.at/, and https://lba.tirol.gv.at/, respectively; last access: 15 February 2023). Sentinel-2 multi-spectral satellite data are provided by Copernicus and processed by Sentinel Hub (https://www.sentinel-hub.com/, last access: 15 February 2023). The Alpine Permafrost Index Map is freely available from https://doi.pangaea.de/10.1594/PANGAEA.784450 (last access 15 February 2023). The rock glacier inventory and corresponding rock glacier catchment inventory are freely available from https://doi.pangaea.de/10.1594/PANGAEA.921629 (last access 15 February 2023). The employed software is freely available from https://sourceforge.net/projects/saga-gis/ (SAGA GIS), https://grass.osgeo.org/ (GRASS GIS), and https://cran.r-project.org/ (R). Grain size analysis results, surface velocity data, and additional photographs are provided in the Supplementary Material." The new Supplementary Figures, the grain size analysis results (Supplementary Table S1) and the surface velocity data (Supplementary Table S2) are included in the revised 'Supplementary Material'.

**Page 22, line 452:** "Video supplement. Video documentation of the rapidly draining thermokarst lake, recorded on 13 August 2019 11:20 by Josef Waldner, is available in the Supplementary Material." added. The new Supplementary Video is included in the revised 'Supplementary Material'.

**Page 23, line 484:** "Glacial meltwater infiltrates directly from Glockturmferner and indirectly from Hüttekarferner into Hüttekar Rock Glacier (Fig. 3)."as well as "for both glaciers" added.

**Page 23, line 492-496:** "This research was funded by the Austrian Federal Ministry of Agriculture, Regions and Tourism and the Federal Provinces of Vorarlberg, Tyrol, Salzburg, Styria and Carinthia within the DaFNE project RG-AlpCatch. We thank Josef Waldner and Gerhard Schaffenrath for witness reports, local information, discussion, and photographs. Additional photographs and information were kindly provided by Roman Außerlechner, Thomas Figl, Werner Thöny (Geological Survey of Tyrol), Rudolf Philippitsch, Danny F. Minahan, Ainur Kokimova, and Irène Apra. The authors acknowledge the financial support by the University of Graz." replaced by "We thank Catherine Bertrand and one anonymous referee for their constructive feedback and helpful comments. We also thank Josef Waldner and Gerhard Schaffenrath for witness reports, local information,

discussion, and photographs. Additional photographs and information were kindly provided by Roman Außerlechner, Thomas Figl, Werner Thöny (Geological Survey of Tyrol), Rudolf Philippitsch, Magdalena Seelig, Danny F. Minahan, Ainur Kokimova, Samuel Kainz, and Irène Apra. This research was funded by the Austrian Federal Ministry of Agriculture, Forestry, Regions, and Water Management and the Federal Provinces of Vorarlberg, Tyrol, Salzburg, Styria and Carinthia within the DaFNE project RG-AlpCatch. The authors acknowledge the financial support by the University of Graz."